# Score Matching with Missing Data

**Josh Givens** [1]  **Song Liu** [1]  **Henry W J Reeve** [2]

## Abstract

Score matching is a vital tool for learning the distribution of data with applications across many areas including diffusion processes, energy based modelling, and graphical model estimation. Despite all these applications, little work explores its use when data is incomplete. We address this by adapting score matching (and its major extensions) to work with missing data in a flexible setting where data can be partially missing over any subset of the coordinates. We provide two separate score matching variations for general use, an importance weighting (IW) approach, and a variational approach. We provide finite sample bounds for our IW approach in finite domain settings and show it to have especially strong performance in small sample lower dimensional cases. Complementing this, we show our variational approach to be strongest in more complex high-dimensional settings which we demonstrate on graphical model estimation tasks on both real and simulated data.

## 1. Introduction

Over the last decade, score matching has established itself as a powerful tool with downstream use in many areas of machine learning. Examples include: energy based modelling (Swersky et al., 2011; Bao et al., 2020; Li et al., 2019b), mode-seeking clustering (Sasaki et al., 2014), and perhaps most prominently of all Diffusion processes (Song & Ermon, 2019; Song et al., 2021b; Tashiro et al., 2021; Song et al., 2021a; Huang et al., 2021). Score matching aims to learn the score of a distribution which is the gradient of the log of the probability density function (PDF) ($s(x) = \nabla_x \log p(x)$). In contrast to modelling the density directly, the score does not need to integrate to one meaning there is no need to calculate a normalising constant. This allows it to be much more more flexibly modelled than the

density itself. Furthermore, the validity of the score matching objective itself requires only very mild assumptions of the family of proposed scores further ensuring this flexibility. Alongside the classical method (Hyvärinen, 2005), various adaptations of score matching have arisen to improve performance, decrease computational cost, and extend the approach to a wider range of settings (Hyvärinen, 2007; Vincent, 2011; Song et al., 2020; Liu et al., 2022).

In this work, we extend the score matching framework to handle missing data at training time. Specifically, we learn the full score function from partially missing multidimensional input data, a paradigm we term *missing score matching*. Crucially, our approach is compatible with any parameterised score model, enabling its application to both explicit score formulations and more general approaches such as neural networks (NNs). We propose two methods to adapt the original score matching method as well as its popular adaptations, truncated, sliced, and denoising score matching (Hyvärinen, 2007; Vincent, 2011; Liu & Wang, 2017; Song et al., 2020). These two distinct but closely related methods complement each other allowing for a wide range of problems to be tackled. The first method is a simpler importance weighting (IW) approach which we refer to as marginal IW score matching. For this method we obtain finite sample bounds in the bounded domain setting under certain conditions. We also provide experimental results demonstrating its efficacy in lower dimensional settings and where less data is available. Our second approach is a more computationally sophisticated variational approach which we refer to as marginal variational score matching. We demonstrate the efficacy of this approach in more complex, high dimensional settings by applying it to the problem of graphical model estimation with both real and synthetic datasets.

In section 2 we discuss relevant works for score matching and related fields. In section 3 we will introduce our problem more formally including score matching and any notation used. Section 4 will be used to introduce our methods. Section 5 will present results on some real and simulated datasets. In Section 6 we give our conclusion.

## 2. Related Works

While there has been some work which utilises score matching with missing data, these approaches mostly do so

---

[1]School of Mathematics, University of Bristol, Bristol, UK [2]School of Artificial Intelligence, Nanjing University, China. Correspondence to: Josh Givens <josh.givens@bristol.ac.uk>.

*Proceedings of the 42nd International Conference on Machine Learning*, Vancouver, Canada. PMLR 267, 2025. Copyright 2025 by the author(s).

exclusively through the lens of diffusion models. Specifically works such as MissDiff (Ouyang et al., 2023) and Ambient Diffusion (Daras et al., 2023) require the score function itself to take the form of a neural network (NN) which learns the scores of the fully-observed and corrupted scores simultaneously. This prohibits their use in situations where our model for the score is some explicit parameterisation whose parameters we want to learn as is the case in settings such as energy based modelling Li et al. (2023); Bao et al. (2020); Salimans & Ho (2021) and Gaussian graphical models (Lin et al., 2016; Yu et al., 2018). Ambient Diffusion also requires the data to be further artificially corrupted in order to create a pseudo-supervised learning paradigm making both Ambient Diffusion and MissDiff subject to various levels of out of sample learning without specific adjustments for this phenomenon.

Looking more generally at distribution estimation with missing data, multiple works in the field of generative modelling have looked to tackle the problem of providing a generative model for a distribution given corrupted samples from it. Prominent among these are MisGAN (Li et al., 2019a), which presents a marginalised GAN framework and MCFlow (Richardson et al., 2020) , which presents a EM like normalising flow framework. Neither of these approaches allow for flexible specification of a parametric density estimate however with MCFlow requiring the density to be a normalising flow and MisGAN having no model for the density whatsoever.

To our knowledge, the only approach which seems to adapt score matching to missing data in a parameter preserving manner is presented in (Uehara et al., 2020) using an iterative EM-like procedure. However they themselves admit that there is little intuitive understanding of when this approach will converge. Additionally, due to the nature of the score matching objective, the expectation step cannot be directly approximated using Monte Carlo estimation and instead requires fractional importance weighting, a method which employs nested Monte Carlo estimates introducing bias into the training objective.

Parallel to this, some papers have looked to extend score matching to the latent variable setting, an area with much commonality to missing data (Vértes & Sahani, 2016; Bao et al., 2020; 2021). Latent variable modelling differs in two crucial aspects from missing score matching. Firstly the components which are unobserved (the latent variables) remain constant between samples, and secondly there is not necessarily a notion of a ground truth for the unobserved components in when data is corrupted. Additionally each of these works has limitations; Vértes & Sahani (2016) only applies to exponential families, Bao et al. (2020) requires a gradient unrolling step in its optimisation which is computationally expensive and can lead to errors in the op-

timisation procedure (as acknowledged in their follow on work), and Bao et al. (2021) is only given for denoising score matching, not for classical or sliced score matching.

## 3. Setting

### 3.1. Notation

For $n \in \mathbb{N}$ let $[n] \coloneqq \{1, \ldots, n\}$. For a random variable $Z$ we use $\operatorname{supp}(Z)$ for the support of $Z$. For $f : \mathbb{R}^d \to \mathbb{R}$ we write $\partial_j f(\boldsymbol{x}) \coloneqq \frac{\partial f}{\partial x_j}$ where $\boldsymbol{x} = (x_1, \ldots, x_d)^\top$ and $\nabla_{\boldsymbol{x}} f(\boldsymbol{x}) \coloneqq (\partial_j f(\boldsymbol{x}), \ldots, \partial_d f(\boldsymbol{x}))^\top$, the gradient of $f$. For $\boldsymbol{f} : \mathbb{R}^d \to \mathbb{R}^d$ take $\boldsymbol{f}(\boldsymbol{x})_j$ as the $j^{\text{th}}$ component of $\boldsymbol{f}(\boldsymbol{x})$ and write $\nabla_{\boldsymbol{x}} \cdot \boldsymbol{f}(\boldsymbol{x}) \coloneqq \partial_1 \boldsymbol{f}(\boldsymbol{x})_1 + \cdots + \partial_d \boldsymbol{f}(\boldsymbol{x})_d$. Finally for $\boldsymbol{a}, \boldsymbol{b} \in \mathbb{R}^d$, take $\boldsymbol{a} \circ \boldsymbol{b}$ to be the Hadamard product.

We now introduce some indexing notation which we will be using for RVs and functions throughout. This will prove useful when identifying the missing non-missing components of our data. Let $Z$ be a random variable taking values in $\mathbb{R}^d$. We use $Z_j$ to refer to the $j^{\text{th}}$ component $Z$ and for $\lambda \subseteq [d]$ take $Z_\lambda = \{Z_j\}_{j \in \lambda}$. We use negation in indexing to mean the complementing coordinates. More precisely we let $-j$ denote $[d] \setminus \{j\}$ and let $-\lambda$ denote $[d] \setminus \lambda$. We typically use $Z^{(i)}$ to denote an independent copy of $Z$. For a function $f : \mathcal{X} \to \mathcal{Y}$ and $\boldsymbol{x}_\lambda \in \mathcal{X}_\lambda, \boldsymbol{x}'_{-\lambda} \in \mathcal{X}_{-\lambda}$, we take $f(\boldsymbol{x}_\lambda, \boldsymbol{x}'_{-\lambda})$ to be $f(\boldsymbol{z})$ where

$$z_j \coloneqq \begin{cases} x_j & \text{if } j \in \lambda \\ x'_j & \text{if } j \in -\lambda \end{cases} .$$

We will take $X$ to be a RV taking values in $\mathcal{X} \subseteq \mathbb{R}^d$ representing our original dataset and $X'$ to be a RV representing some generative/variational/importance weighting distribution. i.e., the "artificial distributions" we will utilise in our method. Similarly, we take $\mathbb{E}, \mathbb{E}'$ to be expectations with respect to (w.r.t.) $X, X'$ respectively.

Throughout we take $p$ to be the pdf of the RV, $X$, and $p_\theta$ to be a model therein. We let $q$ represent an unnormalised density (i.e. $N^{-1} \cdot q = p$ for some normalising constant $N > 0$.) We will write marginalisations/conditionings for both true and model densities implicitly with $p(\boldsymbol{x}_\lambda) \coloneqq \int_{\mathcal{X}} p(\boldsymbol{x}) \mathrm{d}\boldsymbol{x}_{-\lambda}$ and $p(\boldsymbol{x}_\lambda | \boldsymbol{x}_{-\lambda})$ being the conditional density of $X_\lambda | X_{-\lambda} = \boldsymbol{x}_{-\lambda}$ for example.

Now that we have introduced our notation we can move onto the key area of focus for our work, score matching.

### 3.2. Score Matching

First proposed by (Hyvärinen, 2005), score matching aims to learn the gradient of the log-density (score). The advantage of this framework over full density approaches such as maximum likelihood estimation (MLE) is that we are not restricted to parametric models which integrate to 1. This allows us to be much more flexible in how we param-

eterise in turn making high dimensional distribution modelling more feasible. We now introduce the approach.

Let $X$ be a RV over $\mathbb{R}^d$ with PDF $p$. We say that $q$ is the unnormalised density of $X$ if $N^{-1} \cdot q(\boldsymbol{x}) = p(\boldsymbol{x})$ where $p$ is the PDF of $X$ and $N$ is the normalising constant of $q$. Define the score, of $X$ to be

$$\boldsymbol{s}(\boldsymbol{x}) := \nabla_{\boldsymbol{x}} \log p(\boldsymbol{x}) = \nabla_{\boldsymbol{x}} \log q(x).$$

The aim of score matching is to learn $\boldsymbol{s}$ from a collection of IID copies of $X$ which we denote $\mathcal{D} := \{X^{(i)}\}_{i=1}^n$. Following Hyvärinen (2005), we introduce a generic parameterised proposal score $\boldsymbol{s}_\theta$ for $\theta \in \Theta \subseteq \mathbb{R}^p$ and aim to minimise the *Fisher Divergence* between the true distribution and our proposal distribution which is given by

$$F(\theta) := \mathbb{E}[\|\boldsymbol{s}(X) - \boldsymbol{s}_\theta(X)\|^2].$$

The key result from Hyvärinen (2005) which enables us to practically implement score matching is that under certain (fairly minimal) regularity conditions, which we provide in Appendix D.1, we have

$$L(\theta) := \mathbb{E}\left[2\nabla_X \cdot \boldsymbol{s}_\theta(X) + \|\boldsymbol{s}_\theta(X)\|^2\right] = F(\theta) - C \tag{1}$$

where here and throughout, we take $C$ to represent any constant which does not depend upon $\theta$. Crucially, $L(\theta)$ is now an expectation of observable random variables. Hence we can now approximate this with our data and take $\hat{\theta}$ as

$$\hat{\theta} := \operatorname*{argmin}_\theta \frac{1}{n} \sum_{i=1}^n \left[2\nabla_{X^{(i)}} \cdot \boldsymbol{s}_\theta(X^{(i)}) + \|\boldsymbol{s}_\theta(X^{(i)})\|^2\right].$$

TRUNCATED SCORE MATCHING

A limitation of standard score matching is that it requires $\lim_{x_i \to \infty} p(\boldsymbol{x}) = 0$ for all $x_i \in \mathbb{R}$. Thus it cannot be used for many distributions with compact support if the density does not converge to zero at the (topological) boundary. Initial work to adapt score matching to truncated distributions was presented in (Hyvärinen, 2007) for distributions on $[0, \infty)$ then further expanded in (Liu et al., 2022; Yu et al., 2022) to general compact spaces $\mathcal{X}$. For our compact space $\mathcal{X} \subseteq \mathbb{R}^d$ we use $\partial \mathcal{X}$ to denote the (topological) boundary. We now minimise some weighted version of the Fisher divergence whose weights go to zero at the boundary. Specifically let $\boldsymbol{g} : \mathcal{X} \to \mathbb{R}$ be a function satisfying $\lim_{\boldsymbol{x} \to \boldsymbol{x}'} \boldsymbol{g}(\boldsymbol{x})_j = 0$ for any $\boldsymbol{x}' \in \partial \mathcal{X}$, $j \in [d]$. Our objective is then

$$F_{\mathrm{T}}(\theta) := \mathbb{E}\left[\left\|\boldsymbol{g}^{\frac{1}{2}}(X) \circ (\boldsymbol{s}_\theta(X) - \boldsymbol{s}(X))\right\|^2\right].$$

Just as in classical score matching we obtain an equivalence (though this time via Green's theorem rather than simple

integration by parts) giving us that under certain regularity conditions on $\boldsymbol{g}$, $\boldsymbol{s}$, and $\mathcal{X}$,

$$L_{\mathrm{T}}(\theta) := \mathbb{E}\left[\sum_{j \in d} \boldsymbol{g}(X)_j \left(2\partial_j \boldsymbol{s}_\theta(X)_j + \boldsymbol{s}_\theta(X)_j^2\right)\right]$$
$$+ \mathbb{E}\left[\sum_{j \in d} \partial_j \boldsymbol{g}(X)_j \boldsymbol{s}_\theta(X)_j\right] = F_{\mathrm{T}}(\theta) - C.$$

This can again be approximated via data using standard Monte Carlo approximation. Full details on the conditions required for this approach alongside the proof can be found in (Liu et al., 2022). Two other key extensions of score matching are sliced score matching (Song et al., 2020) and denoising score matching (Vincent, 2011). We introduce these extensions in Appendix D with our corresponding adaptations to missing data given in Appendix A.1. Now, we give our missing data scenario.

### 3.3. Missing Data Scenario
Instead of observing samples from $X$ we assume that we observe samples from the corrupted version of the RV given by $\tilde{X}$. To define $\tilde{X}$ we introduce a mask RV $M$ over $\{0, 1\}^d$ and then define $\tilde{X}$ by

$$\tilde{X}_j = \begin{cases} X_j & \text{if } M_j = 1 \\ \varnothing & \text{if } M_j = 0 \end{cases}$$

where $\tilde{X}_j = \varnothing$ represents that coordinate being missing. We will be focussing on the missing completely at random scenario where $M \perp X$. However, we do provide an extension to missing not at random data in Appendix A.1.4. We introduce the RV $\Lambda$ on $\mathcal{P}([d])$ defined by $\Lambda := \{i \in [d] | M_i = 1\}$ so that $\Lambda$ gives the non-corrupted coordinates of $\tilde{X}$ and take $\lambda$ to be a sample of $\Lambda$. Crucially given samples from $\tilde{X}$, we also have samples from $X_\Lambda$.

Our aim is to adapt the score matching objective to estimate the full score $\boldsymbol{s}$ by a parameterised score $\boldsymbol{s}_\theta$ using samples from the corrupted data $\tilde{\mathcal{D}} := \{\tilde{X}^{(i)}\}_{i=1}^n \equiv \{X_{\Lambda_i}^{(i)}\}_{i=1}^n$.

## 4. Marginal Score Matching
To motivate our approach we look at how we might use MLE in the case where the normalising constant and conditional normalising constants were calculable. For $p_\theta$ our parametric model of the density, we would choose $\hat{\theta}$ to be

$$\hat{\theta} := \operatorname*{argmax}_\theta \sum_{i=1}^n \log \tilde{p}_\theta(\tilde{X}^{(i)})$$

where $\tilde{p}_\theta$ is the associated corrupted data density when $X \sim p_\theta$. As our data is missing completely at random

this is actually equivalent to maximising

$$\sum_{i=1}^{n} \log p_{\theta;\Lambda_i}(X_{\Lambda_i}^{(i)}), \text{ where } p_{\theta;\lambda}(\boldsymbol{x}_\lambda) \coloneqq \int_{\mathcal{X}_{-\lambda}} p_\theta(\boldsymbol{x}) \mathrm{d}\boldsymbol{x}_{-\lambda}.$$

For notational simplicity we will thus reframe our problem as working with marginal samples $\{X_{\Lambda_i}^{(i)}\}_{i=1}^{n}$.

### 4.1. Marginal Score Matching
Our approach is to directly alter the score matching objective similarly. Just as densities have associated *marginal densities* so do scores have associated *marginal scores*.

**Definition 4.1** (Marginal Score function). Let $\boldsymbol{s}$ be a score function with $\boldsymbol{s}(\boldsymbol{x}) = \nabla_{\boldsymbol{x}} \log q(\boldsymbol{x})$ for $q$ an unnormalised PDF. Then the associated *marginal score* function is

$$\boldsymbol{s}_\lambda(\boldsymbol{x}_\lambda) \coloneqq \nabla_{\boldsymbol{x}_\lambda} \log \int_{\mathbb{R}^{d-|\lambda|}} q(\boldsymbol{x}) \mathrm{d}\boldsymbol{x}_{-\lambda}. \qquad (2)$$

This definition of marginal scores restricts $\boldsymbol{s}$ to a genuine score function. For this reason we will also want $\boldsymbol{s}_\theta$ to always be a genuine score function or at least to have an anti-derivative. The simplest way to achieve this is to work with $q_\theta : \mathcal{X} \to (0, \infty)$ as our baseline and define $\boldsymbol{s}_\theta(\boldsymbol{x}) \coloneqq \nabla_{\boldsymbol{x}} \log q_\theta(\boldsymbol{x})$. We will also take $p_\theta(\boldsymbol{x}) \coloneqq \left(\int_\mathcal{X} q_\theta(\boldsymbol{x}) \mathrm{d}\boldsymbol{x}\right)^{-1} q_\theta(\boldsymbol{x})$ which we assume to be unknown.

With this notion of a marginal score we can define our marginal Fisher divergence to be

$$F_\mathrm{M}(\theta) \coloneqq \mathbb{E}[\|\boldsymbol{s}_\Lambda(X_\Lambda) - \boldsymbol{s}_{\Lambda;\theta}(X_\Lambda)\|^2] \qquad (3)$$

where $\boldsymbol{s}_{\lambda;\theta}$ is defined analogously to $\boldsymbol{s}_\lambda$. As with normal score matching can relate this objective to one involving no terms of $\boldsymbol{s}_\lambda$. We first need the following assumptions.

**Assumption 4.2.** For any $\theta > 0, \lambda \in \mathrm{supp}(\Lambda)$:

(a) $p_\theta$ is well defined, i.e. $\int_\mathcal{X} q_\theta(x) \mathrm{d}x < \infty$;

(b) $\mathbb{E}[\|\boldsymbol{s}_\lambda(X_\lambda)\|^2], \mathbb{E}[\|\boldsymbol{s}_{\lambda;\theta}(X_\lambda)\|^2] < \infty$;

(c) $p_\lambda(\boldsymbol{x})$ is differentiable and $q_{\lambda;\theta}$ is twice differentiable;

(d) $p_\lambda(\boldsymbol{x}_\lambda)\boldsymbol{s}_{\lambda;\theta}(\boldsymbol{x}_\lambda) \longrightarrow 0$ as $\|\boldsymbol{x}\| \longrightarrow \infty$;

(e) $p_{\lambda;\theta}(X_\lambda) = p_\lambda(X_\lambda)$ almost surely (a.s.) for all $\lambda \in \mathrm{supp}(\Lambda)$, implies that $p_\theta(X) = p(X)$ a.s..

Assumption (a) ensures that our proposal unnormalised density is always a genuine unnormalised density. Assumptions (b)-(d) are similar to the standard assumptions given for standard score matching. Assumption (e) is an identifiability assumption which is required to be feasibly able to learn the true data distribution from our corrupted data.

**Proposition 4.3.** *Given Assumptions 4.2(a)-(d) hold*

$$L_\mathrm{M}(\theta) \coloneqq \mathbb{E}[2\nabla_{X_\Lambda} \cdot \boldsymbol{s}_{\Lambda;\theta}(X_\Lambda) + \|\boldsymbol{s}_{\Lambda;\theta}(X_\Lambda)\|^2] \qquad (4)$$
$$= F_\mathrm{M}(\theta) - C.$$

*If (e) also holds and there exists some $\theta^*$ such that $\boldsymbol{s}_{\theta^*}(X) = \boldsymbol{s}(X)$ a.s.. Then if $\tilde\theta$ is a minimiser of $L_\mathrm{M}(\theta)$ we have that $q_{\tilde\theta}(X) = Np(X)$ a.s. for some constant $N$, i.e. the minimiser is the true unnormalised density.*

Through this result we have shown, much like with standard score matching, that under certain regularity conditions our objective is uniquely minimised by the true unnormalised density. We then approximate this objective by

$$\hat{L}_{\mathrm{M};n}(\theta) \coloneqq \frac{1}{n} \sum_{i=1}^{n} \nabla_{X_{\Lambda_i}^{(i)}} \cdot \boldsymbol{s}_{\Lambda_i,\theta}(X_{\Lambda_i}^{(i)}) + \|\boldsymbol{s}_{\Lambda_i;\theta}(X_{\Lambda_i}^{(i)})\|^2$$

and choose $\hat\theta = \mathrm{argmin}_\theta \hat{L}_{\mathrm{M};n}(\theta)$.

Unfortunately this approach in its current state is practically infeasible as the integrals involved in deriving the marginal scores for any non-trivial problem will be intractable. Hence, we must devise a way to estimate the marginal scores without having to compute the integrals. We tackle this issue in Section 4.2, but first we provide a similar result for the case of truncated score matching.

#### 4.1.1. TRUNCATED SCORE MATCHING
Truncated score matching can be adapted similarly to standard score matching by simply having marginal weighting functions $\boldsymbol{g}_\lambda : \mathcal{X}_\lambda \to [0, \infty)$ for each subset $\lambda \in \mathrm{supp}(\Lambda)$ and taking the marginal truncated Fisher divergence to be

$$F_\mathrm{TM}(\theta) \coloneqq \mathbb{E}\left[\left\|\boldsymbol{g}_\Lambda(X_\Lambda)^{\frac{1}{2}} \circ (\boldsymbol{s}_\Lambda(X_\Lambda) - \boldsymbol{s}_{\Lambda;\theta}(X_\Lambda))\right\|^2\right].$$

using integration by parts gives the following equivalence

$$L_\mathrm{TM}(\theta) \coloneqq \mathbb{E}\left[\sum_{j \in \Lambda} \boldsymbol{g}_\Lambda(X_\Lambda)_j \left(\boldsymbol{s}_{\Lambda;\theta}(X_\Lambda)_j^2 + 2\partial_j \boldsymbol{s}_{\Lambda;\theta}(X_\Lambda)_j\right)\right]$$
$$+ \mathbb{E}\left[\sum_{j \in \Lambda} 2\partial_j \boldsymbol{g}_\Lambda(X_\Lambda)_j \boldsymbol{s}_{\Lambda;\theta}(X_\Lambda)_j\right] \qquad (5)$$
$$= F_\mathrm{TM}(\theta) - C.$$

Proof given in Appendix C.1.1. We then take $\hat{L}_{\mathrm{TM};n}$ as the Monte-Carlo estimate of $L_\mathrm{TM}$. We also construct similar objectives from sliced and denoising score matching as well as a similar result for missing not at random data in Appendix A.1. We now move to the task of estimating the marginal scores in these objectives.

### 4.2. Importance Weighting
Our first proposal is an importance weighting approach. Let $p'$ be a density over $R^{d-|\lambda|}$ which we can both evaluate and sample from then

$$\int_{\mathbb{R}^{d-|\lambda|}} q_\theta(\boldsymbol{x}) \mathrm{d}\boldsymbol{x}_{-\lambda} = \mathbb{E}_{X'_\lambda \sim p'}\left[\frac{q_\theta(\boldsymbol{x}_\lambda, X'_{-\lambda})}{p'(X'_{-\lambda})}\right]. \qquad (6)$$

**Algorithm 1** Marginal IW Score Matching

---
**Input:** $\{X_{\Lambda_i}^{(i)}\}_{i\in[n]}, q_\theta, p', \theta_0, r \in \mathbb{N}$.
Set $\theta = \theta_0$.
**repeat**
  **for** i=1 **to** n **do**
    Sample $\{X'^{(i,k)}\}_{k\in r}$ from $p'(.|X_{\Lambda_i}^{(i)})$.
    Use $X_{\Lambda_i}^{(i)}, \{X_{-\Lambda_i}'^{(i,k)}\}_{k\in[r]}$ to get Monte-Carlo estimates, $\hat{s}_{\Lambda_i,r;\theta}(X_{\Lambda_i}^{(i)})$, of the marginal scores by (7).
  **end for**
  Use $\hat{s}_{\Lambda_i,r;\theta}(X_{\Lambda_i}^{(i)})$ to obtain $\hat{L}_{\mathrm{M/TM};n,r}(\theta)$ by (4).
  Compute $\nabla_\theta \hat{L}_{\mathrm{M/TM};n,r}(\theta)$ and update the value of $\theta$.
**until** Maximum iteration reached.

---

This allows us to define our *marginal score estimate*.

**Definition 4.4** (Marginal Score Estimate). For a given $\lambda \in \mathrm{supp}(\Lambda)$, $\boldsymbol{x}_\lambda \in \mathcal{X}_\lambda$, score model, $\boldsymbol{s}_\theta$, and $r \in \mathbb{N}$ we take our estimate of $\boldsymbol{s}_{\theta;\lambda,r}(\boldsymbol{x}_\lambda)$ to be

$$\hat{\boldsymbol{s}}_{\lambda,r;\theta} := \nabla_{\boldsymbol{x}_\lambda} \log\left(\frac{1}{r}\sum_{k=1}^r \frac{q_\theta(\boldsymbol{x}_\lambda, X_{-\lambda}'^{(k)})}{p'(X_{-\lambda}'^{(k)})}\right) \quad (7)$$

where $X_{-\lambda}'^{(1)}, \ldots, X_{-\lambda}'^{(r)}$ are IID copies of $X_{-\lambda}' \sim p'$.

### 4.2.1. IW SAMPLE OBJECTIVE

We can now plug these marginal score estimates into our sample objective for either normal or truncated score matching. We use M/TM to denote analogous definitions and results for both marginal and truncated marginal score matching. Let $\{X_{\Lambda_i}^{(i)}\}_{i=1}^n$ be our samples from $X_\Lambda$. We then take our IW sample objective to be as $\hat{L}_{\mathrm{M/TM};n}(\theta)$ but with $\hat{s}_{\Lambda_i,r;\theta}(X_{\Lambda_i}^{(i)})$ replacing $s_{\Lambda_i;\theta}(X_{\Lambda_i}^{(i)})$. The full objective is given in Appendix E.1.1 We refer to this sample objective as $\hat{L}_{\mathrm{M/TM};n,r}(\theta)$ and take our estimate to be

$$\hat{\theta} := \underset{\theta}{\mathrm{argmin}}\, \hat{L}_{\mathrm{M/TM};n,r}(\theta).$$

Algorithm 1 gives our high level estimation algorithm.

*Remark* 4.5. Algorithm 1 can directly be applied to both sliced and denoised score matching by replacing equation (4) by equations (13) and (15) respectively.

### 4.2.2. FINITE SAMPLE BOUNDS

A benefit of truncated score matching is that it allows us to work on distributions with densities bounded below which enables us to give finite sample bounds for the error of our estimated score w.r.t. our marginal objective. We briefly present these now with more detail given in Appendix A.2.

**Theorem 4.6.** *Suppose assumption 4.2 alongside assumptions A.1, A.11, A.13 from the Appendix hold and let*

$\theta_{n,r} \in \Theta$ *be the minimiser of* $\hat{L}_{\mathrm{TM};n,r}(\theta)$. *If* $\Theta \subseteq \mathbb{R}^p$ *with* $\mathrm{diam}(\Theta) = A$ *then for sufficiently large* $n, r$

$$\mathbb{P}\left(F_{\mathrm{TM}}(\theta_{n,r}) \geq \beta_1 \sqrt{\frac{p\log(dnrA/\delta)}{\min\{r,n\}}}\right) < \delta.$$

Note that $r$ is the number of importance weighting samples for each data sample and therefore is something we can choose ourself. This means that with this approach we can achieve approximately $\sqrt{n}$ convergence rates. A downside however is that to achieve this we need $r$ to be of order at least $n$ which would lead to an $O(n^2)$ computational cost. In practice we find relatively strong performance choosing $r$ small. Setting it at $r = 10$ in our experiments.

*Remark* 4.7. The error presented is measured with respect to our Marginal Fisher Divergence, rather than the full Fisher Divergence (which would be the preferred accuracy metric). Relating these two quantities requires connecting the fully observed distribution to its marginals, a task that depends on the specific form of the distribution. Investigating the assumptions and conditions under which this connection can be made offers an interesting and valuable direction for future research.

### 4.3. Gradient First Approach

A key limitation with an IW approach is that it will struggle in higher dimensional scenarios. Additionally the importance weighting is embedded inside other functions which leads to the same nested expectation issue as the EM approach of Uehara et al. (2020), causing bias in our estimator. As an alternative to this we build upon a variational approach initially discussed in the context of latent variable models in Vértes & Sahani (2016); Bao et al. (2020; 2021).

The core idea is to start with $L_\mathrm{M}$ as before and then take gradients w.r.t. our parameters before then writing our objective in terms of expectations over $X_{-\lambda|\lambda;\theta}$. As we don't then need to take gradients of these expectations w.r.t. $\theta$, we can estimate them with any black-box method we desire, opening the door for variational approximation to be used. This approach has been explored for exponential family distributions (Vértes & Sahani, 2016) and for denoising score matching (Bao et al., 2021) however we provide the most general version of this result which can be applied to any of the score matching methods and any model class. We first introduce the following key Lemma.

**Lemma 4.8.** *Fix* $\lambda \subseteq [d], \boldsymbol{x}_\lambda \in \mathcal{X}_\lambda$. *We have that for any function* $h_\theta : \mathcal{X} \to \mathbb{R}$.

$$\boldsymbol{s}_{\theta;\lambda}(\boldsymbol{x}_\lambda) = \mathbb{E}'[\boldsymbol{s}_\theta(\boldsymbol{x}_\lambda, X_{-\lambda}')_\lambda] \quad (8)$$
$$\nabla\mathbb{E}'[h_\theta(\boldsymbol{x}_\lambda, X_{-\lambda}')] = \mathbb{E}'[\nabla h_\theta(\boldsymbol{x}_\lambda, X_{-\lambda}')] \quad (9)$$
$$+ \mathrm{Cov}'(\boldsymbol{s}_\theta(\boldsymbol{x}_\lambda, X_{-\lambda}'), h_\theta(\boldsymbol{x}_\lambda, X_{-\lambda}'))$$

*where $\nabla$ represents the gradient w.r.t. either $\boldsymbol{x}_\lambda$ or $\theta$ and here $\mathbb{E}'$, $\mathrm{Cov}'$ are w.r.t. $X'_{-\lambda}|X_\lambda = \boldsymbol{x}_\lambda \sim p_\theta(.|\boldsymbol{x}_\lambda)$.*

This results allows us to obtain our alternative objective.

**Corollary 4.9.** *Let $L_{\mathrm{M}}$ be defined as in (4). We have that*

$$\nabla_\theta L_{\mathrm{M}}(\theta) = \mathbb{E}\Bigg[2\sum_{j\in\Lambda}\Big(\Psi_\Lambda(\boldsymbol{s}_\theta(.)_j^2 + \partial_j\boldsymbol{s}_\theta(.)_j) \quad (10)$$
$$- \mathbb{E}'[\boldsymbol{s}_\theta(X_\Lambda, X'_{-\Lambda})_j]\Psi_\Lambda(\boldsymbol{s}_\theta(.)_j)\Big)\Bigg]$$

*where for any function $h_\theta : \mathbb{R}^d \to \mathbb{R}$, $\lambda \subseteq [d]$,*

$$\Psi_\Lambda(h_\theta) = \mathbb{E}'[\nabla_\theta h_\theta(X_\Lambda, X'_{-\Lambda})]$$
$$+ \mathrm{Cov}'\big(\nabla_\theta \log q_\theta(X_\Lambda, X'_{-\Lambda}), h_\theta(X_\Lambda, X'_{-\Lambda})\big)$$

*and $\mathbb{E}'$, $\mathrm{Cov}'$ are w.r.t. $X'_{-\Lambda}|X_\Lambda \sim p_\theta(.|X_\Lambda)$ with $\mathbb{E}$ being w.r.t. $X_\Lambda \sim p$.*

Proofs for both results are given in Appendix C.3.

Crucially $\mathbb{E}'$, $\mathrm{Cov}'$ can be estimated freely. This allows us to use variational inference to approximate $p_\theta(\boldsymbol{x}_{-\lambda}|\boldsymbol{x}_\lambda)$ and in turn the expectations and covariances in (10).

*Remark* 4.10. We provide additional implementation details for computing this gradient estimate in Appendix A.5. We also discuss equivalences between this objective and our marginal IW objective in A.3.

We explore estimation of $\mathbb{E}'$, $\mathrm{Cov}'$ in Section 4.3.2 but first we provide a similar result for truncated score matching.

### 4.3.1. TRUNCATED SCORE MATCHING
We define a similar objective for truncated score matching.

**Corollary 4.11.** *With $L_{\mathrm{TM}}$ defined as in (5) we have that*

$$\nabla_\theta L_{\mathrm{TM}}(\theta) = \mathbb{E}\Bigg[2\sum_{j\in\Lambda}\Big(\boldsymbol{g}_\Lambda(X_\Lambda)_j\big\{\Psi_\Lambda(\boldsymbol{s}_\theta(.)_j^2 + \partial_j\boldsymbol{s}_\theta(.)_j)$$
$$- \mathbb{E}'[\boldsymbol{s}_\theta(X_\Lambda, X'_{-\Lambda})_j]\Psi_\Lambda(\boldsymbol{s}_\theta(.)_j)\big\}$$
$$+ \partial_j\boldsymbol{g}_\Lambda(X_\Lambda)_j\Psi_\Lambda(\boldsymbol{s}_\theta(.)_j)\Big)\Bigg] \quad (11)$$

*with $\Psi_\Lambda$ and $\mathbb{E}'$ defined as in Corollary 4.9.*

Proof given in Appendix C.1.1. Similar results for sliced and denoising score matching are given in Appendix A.1.

### 4.3.2. VARIATIONAL APPROXIMATION
We can now use variational approximation to estimate the expectations and covariances in Corollaries 4.9 & 4.11. Specifically, let $p'_\phi(\boldsymbol{x}_{-\lambda}|\boldsymbol{x}_\lambda)$ be some generative conditional distribution dependent upon parameter $\phi$. We want to train $p'_\phi$ to approximate $p_\theta$. We may write $\phi(\theta)$ to highlight

the dependence on our current parameter estimate however we will omit this for brevities sake. The following proposition from Bao et al. (2020) shows us how to train $\phi$.

**Proposition 4.12** (Bao et al. (2020)). *For distributions $p'$, $p$ let $F(p'|p)$ and $\mathrm{KL}(p'|p)$ be the Fisher and KL divergences between $p'$ and $p$. We have that for any $\lambda \subseteq [d]$, $\boldsymbol{x}_\lambda \in \mathcal{X}_\lambda$*

$$\mathrm{KL}(p'_\phi(.|\boldsymbol{x}_\lambda)|p_\theta(.|\boldsymbol{x}_\lambda)) = \mathbb{E}'\left[\log\left(\frac{p'_\phi(X'_{-\lambda}|\boldsymbol{x}_\lambda)}{q_\theta(\boldsymbol{x}_\lambda, X'_{-\lambda})}\right)\right] + B$$

$$F(p'_\phi(.|\boldsymbol{x}_\lambda)|p_\theta(.|\boldsymbol{x}_\lambda)) = \mathbb{E}'\bigg[\Big\|\nabla_{X'_{-\lambda}}\log\big(p'_\phi(X'_{-\lambda}|\boldsymbol{x}_\lambda)\big)$$
$$- \boldsymbol{s}_\theta(\boldsymbol{x}_\lambda, X'_{-\lambda})_{-\lambda}\Big\|^2\bigg]$$

*where expectations are w.r.t. $X'_{-\lambda} \sim p'_\phi(.|\boldsymbol{x}_\lambda)$ and $B$ is a constant not depending upon $\phi$ (but will depend on $\theta$.) In other words we can fit to the conditional density $p_\theta(.|\boldsymbol{x}_\lambda)$ given only the unconditional unnormalised density $q_\theta(\boldsymbol{x}_\lambda, .)$ or full score $\boldsymbol{s}_\theta(\boldsymbol{x}_\lambda, .)$.*

This allows us to train $p'_\phi(.|\boldsymbol{x}_\lambda)$ to approximate the conditional density, $p_\theta(.|\boldsymbol{x}_\lambda)$. In our case we won't be learning this variational model for a fixed $\boldsymbol{x}_\lambda$ or even fixed observed coordinates $\lambda$. Hence we take our objective to be one of

$$J_{KL}(\phi, \theta) \coloneqq \mathbb{E}\left[\log\left(\frac{p'_\phi(X'_{-\Lambda}|X_\Lambda)}{q_\theta(X_\Lambda, X'_{-\Lambda})}\right)\right]$$

$$J_F(\phi, \theta) \coloneqq \mathbb{E}\left[\left\|\nabla_{X'_{-\Lambda}}\log\left(\frac{p'_\phi(X'_{-\Lambda}|X_\Lambda)}{q_\theta(X_\Lambda, X'_{-\Lambda})}\right)\right\|^2\right]$$

with $(X_\Lambda, X'_{-\Lambda}) \sim p'_\phi(X'_{-\Lambda}|X_\Lambda)p(X_\Lambda)$. We then take and $\hat{J}_F, \hat{J}_F$ to be the Monte-Carlo approximations with samples $(X_\Lambda, X'_{-\Lambda})$ from the same distribution.

*Remark* 4.13. $J_F$ has the advantage of not needing to know the normalising constant of $q'_\phi = N_\phi \cdot p'_\phi$ either.

*Remark* 4.14. As $\phi$ depends upon $\theta$, we need to update it each time we update $\theta$. In practice we find taking 10 gradient steps of $\phi$ for each gradient step of $\theta$ to work well.

With this, we define $\widehat{\nabla_\theta L}_{\mathrm{M/TM}}(\theta)$ to be the Monte-Carlo estimate of (10)/(11) with samples $\{(X^{(i)}_{\Lambda_i}, X'^{(i,k)}_{-\Lambda_i})\}_{(i,k)\in[n]\times[r]}$ where $X^{(i)}_\Lambda$ are our original corrupted data samples from $p$ and $X'^{(i,k)}_{-\Lambda}$ are our variational samples from $p'_\phi(.|X^{(i)}_{\Lambda_i})$. We can now state our full variational approach which is given in Algorithm 2.

*Remark* 4.15. Algorithm 2 can directly be applied to both sliced and denoised score matching by replacing equation (10) by equations (14) and (16) respectively.

**Algorithm 2** Marginal Variational Score Matching

**Input:** $\{X_{\Lambda_i}^{(i)}\}_{i\in[n]}$, $q_\theta$, $p'_\phi$, $\theta_0$, $\phi_0$, $L \in \mathbb{N}$, $r \in \mathbb{N}$.
Set $\theta = \theta_0, \phi = \phi_0$.
**repeat**
  **for** $l = 1$ **to** $L$ **do**
    For $i \in [n]$ sample $X'^{(i)}$ from $p'_\phi(.|X_{\Lambda_i}^{(i)})$.
    Use $\{(X_{\Lambda_i}^{(i)}, X'^{(i)}_{-\Lambda_i})\}_{i\in[n]}$ to get Monte-Carlo approximates of $J_{KL/F}(\phi, \theta)$ given by $\hat{J_{KL/F}}(\phi, \theta)$.
    Compute $\nabla_\phi \hat{J_{KL/F}}(\phi, \theta)$ and update $\phi$.
  **end for**
  For $i \in [n]$ sample $\{X'^{(i,k)}\}_{k\in r}$ from $p'_\phi(.|X_{\Lambda_i}^{(i)})$.
  Use $\{(X_{\Lambda_i}^{(i)}, X'^{(i,k)}_{-\Lambda_i})\}_{(i,k)\in[n]\times[r]}$ to get our Monte-Carlo estimate, $\widehat{\nabla_\theta L}_{\mathrm{M/TM}}(\theta)$ using equation (10)/(11).
  Use this gradient estimate to update $\theta$.
**until** Maximum iterations reached.

# 5. Results

Here we go through simulated results comparing our IW approach (Marg-IW) in Algorithm 1 and our variational approach (Marg-Var) in Algorithm 2 to the EM approach of Uehara et al. (2020). We also compare to a naive marginalisation approach involving zeroing out the missing dimensions and only taking the observed output dimensions of the score, which we call Zeroed Score Matching. This approach is the natural adaptation of MissDiff from Ouyang et al. (2023) away from NN to explicitly parameterised models. We describe Zeroed Score Matching and its relation to MissDiff in Appendix D.2. In our experiments, we highlight a unique strength of our methods by applying them to explicitly parameterised score models. We could however, equally apply them to more complex, noninterpretable models such as NNs. More implementation details can be found in Appendix E.3. [1]

## 5.1. Parameter Estimation

### 5.1.1. TRUNCATED GAUSSIAN MODEL

In this experiment a 10-dim normal distribution is set up with fixed mean and random covariance before being truncated on the first 3 dimensions. 1000 samples are taken and corrupted independently on each coordinate with probability 0.2. For each of our methods a Gaussian score is fit and the Fisher divergence between this score and the truth computed. This is repeated 200 times with the mean Fisher divergence alongside 95% C.I.s then presented in figure 1. More details in Appendix E.3.1. Marg-IW and EM perform best with Marg-Var approaching asymptotically. We see the effect of Zeroed's naive marginalisation as it does

---
[1] All code and data for the experiments presented can also be found at `https://github.com/joshgivens/ScoreMatchingwithMissingData`

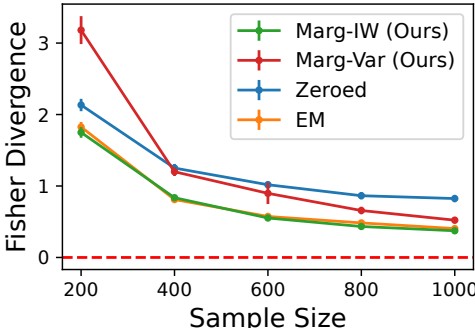

Figure 1: Average Fisher Divergence for Gaussian score estimates alongside 95% C.I.s Lower is better.

not converge, a phenomenon we discuss more in Appendix D.2. In Appendix B.1.1 we present the average mean and precision estimation error for this experiment. In Appendix B.1.2 we present the untruncated results and illustrate how the naive marginalisation poorly models strong relationship between dimensions 1 and 10.

### 5.1.2. NON-GAUSSIAN MODEL

For this experiment we tested our parameter estimation for a an ICA inspired unnormalisable model of the form

$$p(\boldsymbol{x}) \propto \exp \sum_{i,j} \theta_{i,j}^* x_i^2 x_j^2.$$

Here we parameterise our model identically with the aim of estimating $\theta^*$. We vary the dimension of $X$ and plot the estimation error with a sample size of 1,000 and each coordinate missing independently with probability 0.5. The results are presented in Figure 2.

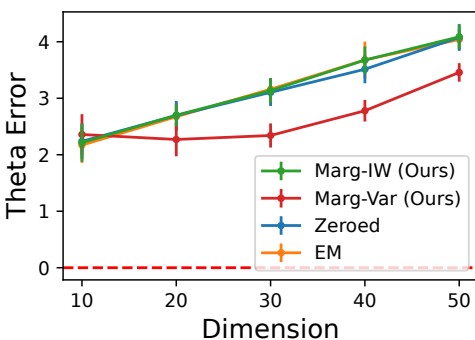

Figure 2: Average Fisher Divergence for Gaussian score estimates alongside 95% C.I.s. Lower is better.

Our variational method (Marg-Var) consistently yields the lowest error. Moreover, as the dimensionality increases, the performance gap between Marg-Var and the other methods widens. This supports the notion that our approach is

more accurately able to capture complex marginalisations than the competing approaches which fail as the dimension grows. We note that all other methods perform comparably with the performance of EM and Marg-IW being indistinguishable, a pattern we observe throughout our experiments. This similarity is unsurprising both approaches use self normalised importance weighting to approximate conditional expectations with respect to our current score estimate while being broadly motivated by fitting to the marginal scores. Nevertheless, the precise mechanism for this similarity remains unclear and warrants further exploration. Additional experiments exploring the effect of sample size and missingness probability on estimation accuracy are given in appendix B.1.3.

### 5.2. Gaussian Graphical Model Estimation

Gaussian graphical models (GGM) are a popular way of modelling dependence between dimensions of data. Let us assume that the underlying data follows a Gaussian distribution with mean $\mu \in \mathbb{R}^d$ and precision $P \in \mathbb{R}^{d \times d}$. In this setting, a Bayesian network (BN) can represent the dependencies between the dimensions of $X$ with the (undirected) edges of the BN exactly being the non-zero off-diagonal entries of the precision, $P$. Hence estimating the precision matrix P gives the BN. Score matching has been shown to be an effective way of achieving this with L1-regularisation on the off-diagonal of $P$ to push terms to 0 (Lin et al., 2016; Yu et al., 2018). Decreasing the level of L1-regularisation then gives a range of classifiers with increasing True and False positive rates (TPR/FPR) as the level of regularisation decreases. Score matching can also be applied to truncated GGMs where we aim to learn the original BN but only observe the samples inside some truncated region.

We apply our methods to learn GGMs and truncated GGMs with missing data as well. We use varying levels of L1 regularisation on our objective via proximal stochastic gradient descent in our optimisation (Beck, 2017).

#### 5.2.1. STAR SHAPED TRUNCATED GRAPHICAL MODEL

Here we create a star shaped GGM in which one node has a high probability of being connected with each other node independently and all other connections have probability 0. We truncate the data along a random hyperplane such that 20% of the distribution lies outside of the truncation boundary. Each coordinate is then MCAR independently with the same probability. We run multiple experiments with this probability ranging from 0.2 to 0.9 and present the results in figure 3. As we can see here Marg-Var performs best with all other approaches performing comparably. For illustrative purposes, we plot individual ROC curves from this experiment in Appendix B.2.3.

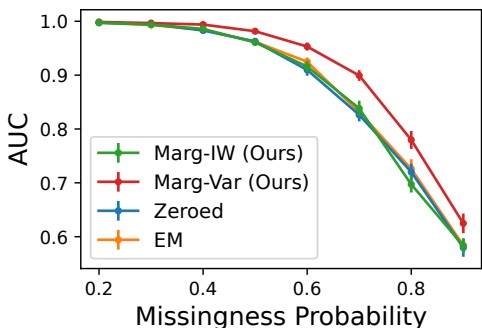

Figure 3: Mean AUC of star graph edge detection with varying missingness alongside 95% C.I.s. Higher is better.

#### 5.2.2. UNSTRUCTURED DENSE GRAPHICAL MODEL

Here we create a GGM by making each edge occur independently with probability 0.5. The rest of the experiment was constructed as before. Results are given in Figure 4. Again we can see that our variational approach performs

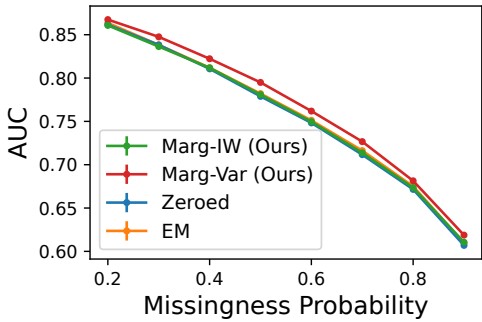

Figure 4: Mean AUC of dense graph edge detection with varying missingness alongside 95% C.I.s. Higher is better.

best though not as clearly as in the previous example. We believe this to be because for more unstructured problems, naive marginalisation performs moderately well.

#### 5.2.3. INCREASING NUMBER OF STARS

To explore this further, we construct and experiment where we vary the number of star centres (high degree nodes) while keeping the edge density constant. We present the results in Figure 5. As we increase the number of star centres, Marg-Var no longer noticeably outperforms the other approaches. This is because as the number of stars increases, (i.e. the structure of the graph decreases) naive marginalisation is a better approximation. This is illustrated on the marginal precisions themselves in Appendix B.2.1.

#### 5.2.4. S&P 100

Here we took closing price data over 5 years for the 100 stocks in the S&P 100 with each stock being a dimension

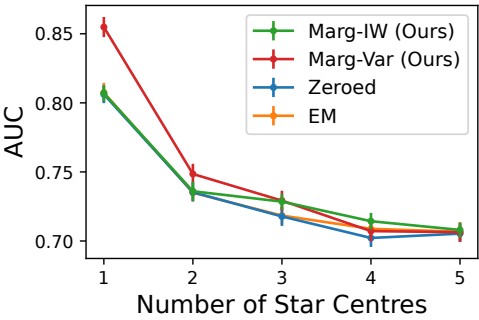

Figure 5: Mean AUC with 95% C.I.s for edge detection as number of star centres in graph varies. Higher is better.

and each day being a sample. Gaussian graphical models with various levels of connectivity were then constructed using standard score matching on the fully observed data. The data was then artificially corrupted and each missing score matching approach applied. The AUC was then calculated for each method taking the GGM from fully observed score matching as the ground truth. More details given in appendix E.3.3. The results are shown in figure 6.

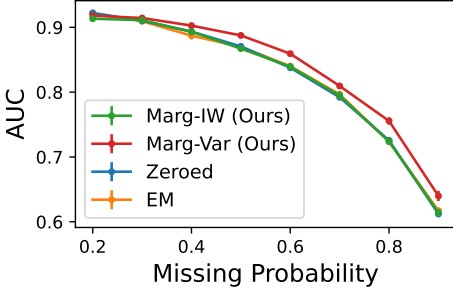

Figure 6: Mean AUC of various methods when compared to non-corrupted score matching with 95% confidence intervals on stocks in S&P 100. Higher is better.

As we can see Marg-Var clearly out performs all the other approaches which appear to perform equivalently.

### 5.2.5. YEAST DATA
Here data first introduced in Brem & Kruglyak (2005) is used consisting of readings of expression for 7086 genes/ORFs across 262 yeast segregants. Each gene represents a dimension with each segregant representing a sample. We subset the data to take the 106 genes present in at least 95% of the samples with the aim of learning the relationship between them. The same approach as the previous section is applied with the results shown in figure 7. Again Marg-Var clearly outperforms the other approaches which all perform comparably.

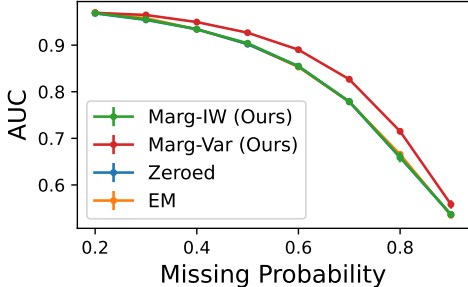

Figure 7: Mean AUC of various methods when compared to non-corrupted score matching with 95% confidence intervals on genetic yeast data. Higher is better.

## 6. Conclusion
To conclude, score matching is a versatile method whose applications at the heart of modern machine learning problems. In this work we have tackled the problem of adapting score matching to partially missing data. We have presented two separate but related approaches to this method, one using importance weighting and another using variational approximation. We have also provided extensions of these methods to truncated score matching, sliced and denoising score matching. For truncated score matching with our IW approach we have provided finite sample bounds on the accuracy of the estimated score in terms of the marginal truncated Fisher divergence.

We have provided several simulated and real world experiments demonstrating our methods' efficacy for both parameter estimation and downstream GGM edge detection. We have shown the benefits and drawbacks of each approach with IW performing best in lower dimensional settings with less data and the variational approach performing best in more complicated higher dimensional settings.

There is, however still much work to be done in this area. From a theoretical perspective, while we have finite sample bound on the error of our loss, marginal nature of the loss makes it unclear exactly how this translates to parameter or general score model accuracy, leaving room for further theoretical exploration. From an implementation perspective, variational inference in the presence of missing data requires accounting for the randomness of "latent" and "observed" variables. The standard variational inference technique can be further refined to accommodate this setting. Finally, since our method is compatible with denoised score matching, it can naturally be extended to diffusion-based model. This paves the way for future work on applying our approach to generative modelling with diffusion processes in the presence of missing data.

## Acknowledgements

Josh Givens was supported by a PhD studentship from the EPSRC Centre for Doctoral Training in Computational Statistics and Data Science (COMPASS).

## Impact Statement

This paper presents work whose goal is to advance the field of Machine Learning. There are many potential societal consequences of our work, none which we feel must be specifically highlighted here.

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

# A. Additional Theoretical Results

Here we present some interesting results which we feel help further build up the landscape of our method but were unable to fit within the main body of the paper.

## A.1. Additional Methods

Firstly some additional adaptations of score matching. Most of these are relatively immediate adaptations following our framework for missing score matching although there are some important aspects and caveats which make them worth officially documenting. Missing and sliced score matching are introduced in detail in Appendix D.

### A.1.1. TRUNCATED SCORE MATCHING

We have already presented truncated score matching in the paper however we present it in more details alongside its assumptions here.

**Assumption A.1.** For any $\lambda \in \text{supp}(\Lambda), \theta \in \Theta$:

- $\mathcal{X}_\lambda$ is connected, open and Lipschitz;

- $p_\lambda, g_\lambda, q_{\lambda;\theta} \in H^1(\mathcal{X}_\lambda)$;

- $p_\lambda, g_\lambda$ are continuously differentiable and $q_{\theta;\lambda}$ is twice continuously differentiable;

- for any $\boldsymbol{x}'_\lambda \in \partial\mathcal{X}_\lambda$, and $j \in \lambda$ we have

$$\lim_{\boldsymbol{x}_\lambda \longrightarrow \boldsymbol{x}'_\lambda} \boldsymbol{s}_{\lambda;\theta}(\boldsymbol{x}_\lambda)_j p_\lambda(\boldsymbol{x}_\lambda) g_\lambda(\boldsymbol{x}_\lambda) v_j(\boldsymbol{x}'_\lambda) = 0.$$

where $v(\boldsymbol{x}'_\lambda)$ is the normal vector to the boundary $\delta X_\lambda$.

This now leads us to our proposition on the validity of our population objective.

**Proposition A.2.** *Suppose that assumptions 4.2 & A.1 hold. Then we have*

$$J_{\text{TM}}(\theta) \coloneqq \mathbb{E}\left\{g_\Lambda(X)\|\boldsymbol{s}_{\Lambda;\theta}(X_\Lambda) - \boldsymbol{s}_\Lambda(X_\Lambda)\|^2\right\} = L_{\text{TM}}(\theta) - C \tag{12}$$

*where $C$ is does not depend upon $\theta$. As a direct result for $\tilde{\theta}$ a minimiser of $L_{\text{TM}}(\theta)$ we have that $\boldsymbol{s}_{\tilde{\theta}}(X) = \boldsymbol{s}(X)$ a.s..*

*Proof.* Proof given in Appendix C.1.1 $\qquad\qquad\square$

### A.1.2. MISSING SLICED SCORE MATCHING

For readers who are unfamiliar with sliced score matching we provide a brief introduction in Appendix D.3. For sliced score matching the only adaptations we need to make is to use our marginal scores and now alter our projection vectors to be over the appropriate subspace. Thus our objective becomes

$$\begin{aligned}L_{\text{SM}}(\theta) &\coloneqq \mathbb{E}[2\left\{\nabla_{X_\Lambda}(V_\Lambda^\top \boldsymbol{s}_{\Lambda;\theta}(X_\Lambda))\right\}^\top V_\Lambda + V_\Lambda^\top \boldsymbol{s}_{\Lambda;\theta}(X_\Lambda)] \\ &= F_{\text{SM}}(\theta) - C\end{aligned}$$

where for any $\lambda \in \text{supp}(\Lambda)$, $V_\lambda$ is a RV on $\mathbb{R}^{|\lambda|}$ satisfying $\mathbb{E}[V_\lambda C_\lambda^\top]$ positive definite and $\mathbb{E}[\|V_\lambda\|^2] < \infty$.

To write this and it's gradient in terms of the full score, $\boldsymbol{s}_\theta$ we can again use Lemma 4.8.

This gives the following results

**Proposition A.3.**

$$L_{\text{SM}}(\theta) = 2\mathbb{E}\left[\mathbb{E}'\left[\nabla_{X_\Lambda}\left(V_\Lambda^\top \boldsymbol{s}_\theta(X_\Lambda, X'_{-\Lambda})^\top V_\Lambda\right)\right] + \mathbb{E}'[(V^\top \boldsymbol{s}_\theta(X_\Lambda, X'_{-\Lambda}))^2] - \mathbb{E}'[(V^\top \boldsymbol{s}_\theta(X_\Lambda, X'_{-\Lambda}))]^2\right] \tag{13}$$

$$\nabla_\theta L_{\text{SM}}(\theta) = 2\mathbb{E}\left[\Psi_\Lambda\left(\nabla_{X_\Lambda}\left(V_\Lambda^\top \boldsymbol{s}_\theta(.)\right)^\top V_\Lambda\right) + \Psi_\Lambda\left(V^\top \boldsymbol{s}_\theta(.))^2\right) - \mathbb{E}'[V^\top \boldsymbol{s}_\theta(X_\Lambda, X'_{-\Lambda})]\Psi_\Lambda\left(V^\top \boldsymbol{s}_\theta(.)\right)\right] \tag{14}$$

*where for any function $h_\theta : \mathbb{R}^d \to \mathbb{R}$,*

$$\Psi_\Lambda(h_\theta) = \mathbb{E}'[\nabla_\theta h_\theta(X_\Lambda, X'_{-\Lambda})] + \mathrm{Cov}'\big(\nabla_\theta \log q_\theta(X_\Lambda, X'_{-\Lambda}), h_\theta(X_\Lambda, X'_{-\Lambda})\big)$$

*and $\mathbb{E}', \mathrm{Cov}'$ are w.r.t. $X'_{-\Lambda}|X_\Lambda \sim p_\theta(.|X_\Lambda)$ with $\mathbb{E}$ being w.r.t. $X_\Lambda \sim p$.*

*Proof.* We first have that

$$L_{\mathrm{SM}}(\theta) = \mathbb{E}\left[2\left\{\nabla_{X_\Lambda}(V_\Lambda^\top \mathbb{E}'[\boldsymbol{s}_\theta(X_\Lambda, X'_{-\Lambda})_\Lambda])\right\}^\top V_\Lambda + (V_\Lambda^\top \mathbb{E}'[\boldsymbol{s}_\theta(X_\Lambda, X'_{-\Lambda})_\Lambda])^2]\right]$$

$$= \mathbb{E}\left[\left(2\sum_{j\in\Lambda} V_j \nabla_{X_\Lambda} \mathbb{E}'[\boldsymbol{s}_\theta(X_\Lambda, X'_{-\Lambda})_j]^\top V_\Lambda\right) + (V_\Lambda^\top \mathbb{E}'[\boldsymbol{s}_\theta(X_\Lambda, X'_{-\Lambda})_\Lambda])^2\right]$$

$$= \mathbb{E}\left[\left(2\sum_{j\in\Lambda} V_j \left(\mathbb{E}'[\nabla_{X_\Lambda}\boldsymbol{s}_\theta(X_\Lambda, X'_{-\Lambda})_j] + \mathrm{Cov}(\boldsymbol{s}_\theta(X_\Lambda, X'_{-\Lambda}), \boldsymbol{s}_\theta(X_\Lambda, X'_{-\Lambda})_j))\right)^\top V_\Lambda\right)\right.$$

$$\left. + (V_\Lambda^\top \mathbb{E}'[\boldsymbol{s}_\theta(X_\Lambda, X'_{-\Lambda})_\Lambda])^2\right]$$

$$= \mathbb{E}\left[\left(2\mathbb{E}'\left[\nabla_{X_\Lambda}\left(V_\Lambda^\top \boldsymbol{s}_\theta(X_\Lambda, X'_{-\Lambda})\right)^\top V_\Lambda\right] + \mathbb{E}'[(V^\top \boldsymbol{s}_\theta(X_\Lambda, X'_{-\Lambda}))^2] - \mathbb{E}'[(V^\top \boldsymbol{s}_\theta(X_\Lambda, X'_{-\Lambda}))]^2\right)\right.$$

$$\left. + \mathbb{E}'[V_\Lambda^\top \boldsymbol{s}_\theta(X_\Lambda, X'_{-\Lambda})_\Lambda]^2\right]$$

$$= 2\mathbb{E}\left[\mathbb{E}'\left[\nabla_{X_\Lambda}\left(V_\Lambda^\top \boldsymbol{s}_\theta(X_\Lambda, X'_{-\Lambda})\right)^\top V_\Lambda\right]\right] + \mathbb{E}'[(V^\top \boldsymbol{s}_\theta(X_\Lambda, X'_{-\Lambda}))^2] - \mathbb{E}'[(V^\top \boldsymbol{s}_\theta(X_\Lambda, X'_{-\Lambda}))]^2\right]$$

The second results directly from applying Lemma 4.8 again alongside the chain rule.

$\square$

### A.1.3. MISSING DENOISED SCORE MATCHING

As with sliced score matching the adaptation is relatively immediate however we do first need to make some further restrictions on our noising process. Specifically we require that for any $t \in [0,1]$, and $j, j' \in [d]$ we have $X(t)_j \perp X(t)_{j'}|X(0)_j$.

In most practical implementations each coordinate is independently noised therefore satisfying this condition. We require this to allow us to easily write the marginal transition kernel for any $\lambda \in \mathrm{supp}(\Lambda)$ given by $p_\lambda(x_\lambda(t)|x_\lambda(0))$. We then make our population objective

$$L_{\mathrm{DM}}(\theta) := \mathbb{E}\left[\nu(t)\left\{\|\boldsymbol{s}_{\Lambda;\theta}(X_\Lambda(t), t)\|^2 + \nabla_{X_\Lambda(t)} \log p_\Lambda(X_\Lambda(t)|X_\Lambda(0))\right\}\right]$$

We can again write this in terms of $\boldsymbol{s}_\theta$ as we do in the following proposition

**Proposition A.4.**

$$L_{\mathrm{DM}}(\theta) = \mathbb{E}\left[\nu(t)\left\{\sum_{j\in\Lambda} \mathbb{E}'\left[\boldsymbol{s}_\theta(X_\Lambda, X'_{-\Lambda})_j\right]^2 + \nabla_{X_\Lambda(t)} \log p_\Lambda(X_\Lambda(t)|X_\Lambda(0))\right\}\right] \tag{15}$$

$$\nabla_\theta L_{\mathrm{DM}}(\theta) = \mathbb{E}\left[\nu(t)\left\{\sum_{j\in\Lambda} \mathbb{E}'\left[\boldsymbol{s}_\theta(X_\Lambda(t), X'_{-\Lambda}, t)_j\right]\left(\mathbb{E}'[\partial_j \boldsymbol{s}_\theta(X_\Lambda(t), X'_{-\Lambda})_j]\right.\right.\right. \tag{16}$$

$$\left.\left.\left. + \mathrm{Cov}'(\boldsymbol{s}_\theta(X_\Lambda(t), X'_{-\Lambda}, \boldsymbol{s}_\theta(X_\Lambda(t), X'_{-\Lambda}, t)_j)\right)\right.\right.$$

$$\left.\left. + \nabla_{X_\Lambda(t)} \log p_\Lambda(X_\Lambda(t)|X_\Lambda(0))\right\}\right]$$

*where for any function $h_\theta : \mathbb{R}^d \to \mathbb{R}$,*

$$\Psi_\Lambda(h_\theta) = \mathbb{E}'[\nabla_\theta h_\theta(X_\Lambda(t), X'_{-\Lambda})] + \mathrm{Cov}'\left(\nabla_\theta \log q_\theta(X_\Lambda(t), X'_{-\Lambda}), h_\theta(X_\Lambda(t), X'_{-\Lambda})\right)$$

*and $\mathbb{E}', \mathrm{Cov}'$ are w.r.t. $X'_{-\Lambda}|X_\Lambda(t) \sim p_\theta(.|X_\Lambda)$ with $\mathbb{E}$ being w.r.t. $X_\Lambda(t) \sim p_t$.*

*Proof.* Using Lemma 4.8, we have that

$$L_{\mathrm{DM}}(\theta) = \mathbb{E}\left[\nu(t)\left\{\sum_{j \in \Lambda} s_{\Lambda;\theta}(X(t)_\Lambda, t)_j^2 + \nabla_{X_\Lambda(t)} \log p_\Lambda(X_\Lambda(t)|X_\Lambda(0))\right\}\right]$$

$$= \mathbb{E}\left[\nu(t)\left\{\sum_{j \in \Lambda} \mathbb{E}'\left[s_\theta(X_\Lambda(t), X'_{-\Lambda}, t)_j\right]^2 + \nabla_{X_\Lambda(t)} \log p_\Lambda(X_\Lambda(t)|X_\Lambda(0))\right\}\right]$$

A second application of the lemma then gives,

$$\nabla_\theta L_{\mathrm{DM}}(\theta) = \mathbb{E}\left[\nu(t)\left\{\sum_{j \in \Lambda} \mathbb{E}'\left[s_\theta(X_\Lambda(t), X'_{-\Lambda}, t)_j\right]\left(\mathbb{E}'[\partial_j s_\theta(X_\Lambda(t), X'_{-\Lambda})_j]\right.\right.\right.$$

$$\left.\left.+ \mathrm{Cov}'(s_\theta(X_\Lambda(t), X'_{-\Lambda}, t), s_\theta(X_\Lambda(t), X'_{-\Lambda}, t)_j)\right\}\right)$$

$$\left.+ \nabla_{X_\Lambda(t)} \log p_\Lambda(X_\Lambda(t)|X_\Lambda(0))\right\}\right].$$

$\square$

### A.1.4. MISSING NOT AT RANDOM DATA

So far we have assumed that our data is missing completely at random so that $X_\Lambda|\Lambda = \lambda \sim X_\lambda$. In other words, we could treat corrupted samples as though they were simply marginal samples and still perform valid inference. Often however, such an assumption is unrealistic and the probability of parts of a sample begin missing depends upon the sample itself. Generally this is split into two cases. Missing at Random (MAR) and Missing not at Random (MNAR). MAR data occurs when the probability of a coordinate being missing depends only upon other coordinates of the sample. This means that $M_j \perp X_j|X_{-j}$. In MNAR data we allow $M_j$ to depend upon $X_j$ as well meaning that an observations value determines its own probability of being missing.

Here we will focus in the MNAR scenario and treat the MAR scenario as a special case of this. The core idea of this approach will be to work with a "joint" score rather than a marginal score. Before we do this we need to set-up our MNAR case. Specifically for $\lambda \in \mathrm{supp}(\Lambda)$ define the event

$$E_\lambda := \{X'_\lambda \neq \varnothing, X'_{-\lambda} = \varnothing\}$$

and define $\varphi_\lambda(X) := \mathbb{P}(E_\lambda|X)$. Throughout we will assume each $\varphi_\lambda$ to be *known*. This is often an unrealistic assumption however this allows us the flexibility of having a method which is independent of how the $\varphi_\lambda$ are learned.

To work with this MNAR data we need to define some adaptations of densities and score functions.

**Definition A.5.** $X$ with PDF $p$ and event $E$ define $p(\boldsymbol{x}; E)$ to be the "joint" density satisfying

$$\int_B p(\boldsymbol{x}; E)\mathrm{d}x = \mathbb{P}(\{X \in B\} \cup E)$$

for all $B \in \mathcal{B}_\mathcal{X}$.

From this and with our particular events we can redefine the missing score as,

$$s_\lambda(\boldsymbol{x}_\Lambda) = \nabla_{\boldsymbol{x}_\lambda} \log p_\lambda(\boldsymbol{x}_\lambda; E_\lambda) \tag{17}$$

$$= \nabla_{\boldsymbol{x}_\lambda} \log\left(\int p(\boldsymbol{x}; E)\mathrm{d}\boldsymbol{x}_{-\lambda}\right) \tag{18}$$

$$= \nabla_{\boldsymbol{x}_\lambda} \log\left(\int p(\boldsymbol{x})\varphi_\lambda(\boldsymbol{x})\mathrm{d}\boldsymbol{x}_{-\lambda}\right)$$

*Remark* A.6. this missing score is *not* the same as the marginal score. We slightly abuse notation here using the same notation as we did for the marginal score. This is however reasonable as for the MCAR case the marginal score and the missing score are identical.

With this newly defined score, we can proceed similarly to the MCAR case and use the objective $\hat{L}_M(\theta)$ defined in (4) or (5) but with our new defined score. We now show a provide a similar justification for this approach as in the MCAR case but first need to introduce an additional assumption.

**Assumption A.7.** For each $\lambda \in \text{supp}(\Lambda)$, $\mathbb{P}(E_\lambda | X_\lambda) > 0$ a.s..

*Remark* A.8. We do not require every missingness pattern to have positive probability just that if a missingness pattern does have positive probability, it has positive probability for every possible underlying sample.

This then leads us to our desired result.

**Proposition A.9.** *Suppose with are in our MNAR set-up and assume that assumptions 4.2 & A.7 hold and that there exists $\theta^*$ with $s_{\theta^*}(X) = s_\theta(X)$ a.s.. Then if $\tilde{\theta}$ is a minimiser of $L_M(\theta)$ where the missing scores are defined by (17 then $s_{\tilde{\theta}}(X) = s(X)$ a.s..*

The proof for this is similar to the MCAR case and is given in Appendix C.1.2.

Now we have our objective we need to see how we can derive $s_\lambda(x_\lambda)$. Again we can do this similarly to the MCAR case. Let $q_\theta$ be our estimate of the unnormalised density then

$$
\begin{aligned}
s_{\lambda;\theta}(x_\lambda) &= \nabla_{x_\lambda} \log p_{\lambda;\theta}(x_\lambda; E_\lambda) \\
&= \nabla_{x_\lambda} \log \int_{\mathcal{X}_{-\lambda}} p_\theta(x; E_\lambda) \mathrm{d}x_{-\Lambda} \\
&= \nabla_{x_\lambda} \log \int_{\mathcal{X}_{-\lambda}} p_\theta(x) \varphi_\lambda(x) \mathrm{d}x_{-\Lambda} \\
&= \nabla_{x_\lambda} \log \int_{\mathcal{X}_{-\lambda}} q_\theta(x) \varphi_\lambda(x) \mathrm{d}x_{-\Lambda} \\
&= \nabla_{x_\lambda} \log \mathbb{E}_{p'} \left[ \frac{q_\theta(x_\lambda, X'_{-\lambda}) \varphi_\lambda(x_\lambda, X'_{-\lambda})}{p'(X'_{-\lambda})} \right] \\
&\approx \nabla_{x_\lambda} \log \frac{1}{r} \sum_{k=1}^{r} \left[ \frac{q_\theta(x_\lambda, X'^{(k)}_{-\lambda}) \varphi_\lambda(x_\lambda, X'^{(k)}_{-\lambda})}{p'(X'^{(k)}_{-\lambda})} \right]
\end{aligned}
$$

As a result we can approximate our objective analogously to our approach for MCAR data.

## A.2. Finite Sample Bounds for Truncated Importance Weighted Score Matching

To be able to prove finite sample bound results we first need to present some key definitions.

**Definition A.10** (Approximate Truncated Marginal Score Matching Objective). For $n, r \in \mathbb{N}$, $\theta \in \Theta$ take our sample objective to be

$$
\hat{L}_{\text{TM};n,r}(\theta) := \frac{1}{n} \sum_{i=1}^{n} g_{\Lambda_i}(X^{(i)}_{\Lambda_i}) \| \hat{s}_{\Lambda_i,r;\theta}(X^{(i)}_{\Lambda_i}) \|^2 + 2 g_{\Lambda_i}(X^{(i)}_{\Lambda_i}) \nabla_{X^{(i)}_{\Lambda_i}} \cdot \hat{s}_{\Lambda_i,r;\theta}(X^{(i)}_{\Lambda_i}) + 2 \nabla_{X^{(i)}_{\Lambda_i}} g_{\Lambda_i}(X^{(i)}_{\Lambda_i})^\top \hat{s}_{\Lambda_i,r;\theta}(X^{(i)}_{\Lambda_i})
$$

with $\hat{s}_{\lambda,r;\theta}(x_\lambda)$ being our estimated marginal score from Definition 4.4.

Additionally we define

$$
f_{0,\lambda}(x, \theta) := \frac{q_\theta(x)}{p'(x_{-\Lambda})} \qquad f_{1,\lambda}(x, \theta) := \frac{\nabla_x q_\theta(x)}{p'(x_{-\Lambda})} \qquad f_{2,\lambda}(x, \theta) := \frac{\nabla_x cdot(\nabla_x q_\theta(x))}{p'(x_{-\lambda})}.
$$

We now set-up the following assumptions

**Assumption A.11.** There exists $a > 0$ s.t. for all $x \in \mathcal{X}$, $\lambda \in \text{supp}(\Lambda), k \in \{0, 1, 2\}$

- $\|f_{k,\lambda}(\boldsymbol{x},\theta)\|,\ g_\lambda(\boldsymbol{x}_\lambda),\ \|\nabla_{\boldsymbol{x}_\lambda}g_\lambda(\boldsymbol{x}_\lambda)\|< a,$

- $\frac{1}{a} < f_{0,\lambda}(\boldsymbol{x},\theta)$

*Remark* A.12. It is this assumptions which restrict us from obtaining a similar result in the non-truncated case as it is unrealistic to have both $\frac{1}{a} < f_{0,\lambda}(\boldsymbol{x})$ and $p(\boldsymbol{x}_\lambda) \to 0$ as $\|\boldsymbol{x}_\lambda\| \to \infty$.

**Assumption A.13.** For each $\lambda \in \mathrm{supp}(\Lambda),\ l \in \{0,1,2\}$ we have that for any $\theta, \theta' \in \Theta$:

$$\|f_{l,\lambda}(\boldsymbol{x},\theta) - f_{0,\lambda}(\boldsymbol{x},\theta')\| \le M_k(\boldsymbol{x})\rho(\theta,\theta'),$$

where $M_k(X_\lambda, \boldsymbol{x}_{-\lambda}), M_k(\boldsymbol{x}_\lambda, X'_{-\lambda})$ are sub-Gaussian with parameters $\sigma_{l,\lambda}, \sigma'_{l,-\lambda}$ respectively for all $\boldsymbol{x}_{-\lambda} \in \mathcal{X}_{-\lambda}$.

*Remark* A.14. This assumption is immediately satisfied if $\Theta$ is compact and $f_{l,\lambda}(\boldsymbol{x},\theta)$ are pointwise Lipschitz w.r.t. $\theta$. Hence this assumption is slightly weaker than a uniformly lipschitz assumption

We can now state our theorem

**Theorem A.15.** *Assume that assumptions 4.2, A.1, A.11, A.13 hold and let $\theta_{n,r} \in \Theta \subseteq \mathbb{R}^p$ be the minimisers of $\hat{L}_{\mathrm{TM};n,r}(\theta)$. If $\Theta \subseteq \mathbb{R}^p$ then for sufficiently large $n, r$*

$$\mathbb{P}\left( F_{\mathrm{TM}}(\theta_{n,r}) \ge \beta_1 \sqrt{\frac{p\log(dnr\,\mathrm{diam}(\Theta)/\delta)}{r}} + \beta_2 \sqrt{\frac{p\log(n\,\mathrm{diam}(\Theta)/\delta)}{n}} + \beta_3 \left(\frac{n+r}{nr}\right)\left(C + \sqrt{\frac{\log(n/\delta)}{n}}\right)\right) < \delta.$$

*where $\beta_1, \beta_2$ depend upon $a$, $\beta_3$ depends upon $a, \{\sigma_{\lambda,l}, \sigma'_{-\lambda,l}\}_{(l,\lambda)\in\{0,1,2\}\times\mathrm{supp}(\Lambda)}$ and $C$ depends upon $a$, $\{\mathbb{E}[M_k(X_\lambda, X'_{-\lambda})]\}_{(l,\lambda)\in\{0,1,2\}\times\mathrm{supp}(\Lambda)}$.*

*Proof.* The proof for this alongside multiple intermediary results can be found in C.2 $\qquad\square$

Here we have shown convergence of our sample/approximate objective to the population objective. This combined with proposition A.2 which states that our population objective is minimised by the true score suggests that our approach does give valid inference for learning the score. A key limitation of our result is that to obtain convergence, we require $r \longrightarrow \infty$. Furthermore, to obtain $\log(n)/n$ rate convergence we need $r$ to be of the same order as $n$. As the computational complexity of our algorithm in $O(nr)$, this suggests that to obtain our desired convergence to the population objective will have $O(n^2)$ computational complexity.

*Remark* A.16. Our dependency on our Lipschitz constants only enters into the $C$ term with the associate sub-Gaussian parameters entering only into the $\sigma$.

*Remark* A.17. Dependence upon $g$ simply requires $g$ and $\nabla g$ bounded. This is achieved on a compact $\mathcal{X}$ by $g(\boldsymbol{x}) = \min_{\boldsymbol{x}'\in\partial X} d(\boldsymbol{x}, \boldsymbol{x}')$ and on a non-compact space by $g(\boldsymbol{x}) = \min_{\boldsymbol{x}'\in\partial X} d(\boldsymbol{x}, \boldsymbol{x}') \bigwedge 1$.

**A.3. Relationship between IW and Variational objectives**
Despite being derived quite differently from the marginal score matching objective. We show below that the two objectives are actually identical in some cases. Specifically, when the IW density $p'$ doesn't depend upon the observed data $\boldsymbol{x}_\Lambda$, we can treat the importance weighted approach as an importance weighting approximation of the gradient estimate in (10). We demonstrate this through the two results below

**Lemma A.18.** *For some density $p'$ which generates IID samples $\{X_{-\lambda}'^{(k)}\}_{k \in r}$ let*

$$w_k := \frac{q_\theta(\boldsymbol{x}_\lambda, X_{-\lambda}'^{(k)})}{p'(X_{-\lambda}'^{(k)})}$$

$$\bar{w}_k := w_k \left( \sum_{k'=1}^{r} w_{k'} \right)^{-1}$$

$$\hat{\boldsymbol{s}}_{\theta,\lambda}(\boldsymbol{x}_\lambda) := \nabla_{\boldsymbol{x}_\lambda} \log \frac{1}{r} \sum_{k=1}^{r} w_k$$

$$\hat{\mathbb{E}}_{iw}[g_\theta(X)] := \frac{1}{r} \sum_{k=1}^{r} \bar{w}_k g_\theta(\boldsymbol{x}_\lambda, X_{-\lambda}'^{(k)})$$

$$\hat{\text{Cov}}_{iw}(f(X), g_\theta(X)) := \frac{1}{r} \sum_{k=1}^{r} \bar{w}_k g_\theta(\boldsymbol{x}_\lambda, X_{-\lambda}'^{(k)}) f(\boldsymbol{x}_\lambda, X_{-\lambda}'^{(k)}) - \left( \frac{1}{r} \sum_{k=1}^{r} \bar{w}_k g_\theta(\boldsymbol{x}_\lambda, X_{-\lambda}'^{(k)}) \right) \left( \frac{1}{r} \sum_{k=1}^{r} \bar{w}_k f(\boldsymbol{x}_\lambda, X_{-\lambda}'^{(k)}) \right).$$

*Then*

$$\hat{\boldsymbol{s}}_{\theta;\lambda}(\boldsymbol{x}_\lambda) = \hat{\mathbb{E}}_{iw}[\boldsymbol{s}_\theta(\boldsymbol{x}_\lambda, X_{-\lambda}')] \tag{19}$$

$$\nabla \hat{\mathbb{E}}_{iw}[g_\theta(\boldsymbol{x}_\lambda, X_{-\lambda}')] = \hat{\mathbb{E}}_{iw}[\nabla g_\theta(\boldsymbol{x}_\lambda, X_{-\lambda}')] + \hat{\text{Cov}}_{iw}(\boldsymbol{s}_\theta(\boldsymbol{x}_\lambda, X_{-\lambda}'), g_\theta(\boldsymbol{x}_\lambda, X_{-\lambda}')). \tag{20}$$

*where $\nabla$ represents the gradient w.r.t. $\boldsymbol{x}_\lambda$ or $\theta$. In other words, we can take importance weights first then gradients (LHS) or gradients and then importance weights (RHS).*

*Proof.* Proof given in Section C.4. $\square$

**Corollary A.19.** *We have that*

$$\nabla_\theta \hat{L}(\theta; \boldsymbol{x}_\lambda, X_{-\Lambda}') = -2\hat{\mathbb{E}}_{iw}[\boldsymbol{s}_\theta(\boldsymbol{x}_\lambda, X_{-\Lambda}')_i] \left\{ \hat{\mathbb{E}}_{iw}[\nabla_\theta \boldsymbol{s}_\theta(\boldsymbol{x}_\lambda, X_{-\Lambda}')_i] + \hat{\text{Cov}}_{iw} \left( \nabla_\theta \log q_\theta(\boldsymbol{x}_\lambda, X_{-\Lambda}'), \boldsymbol{s}_\theta(\boldsymbol{x}_\lambda, X_{-\Lambda}')_i \right) \right\}$$

$$+ 2(\hat{\mathbb{E}}_{iw}[\nabla_\theta \boldsymbol{s}_\theta(\boldsymbol{x}_\lambda, X_{-\Lambda}')_i^2] + \hat{\text{Cov}}_{iw} \left( \nabla_\theta \log q_\theta(\boldsymbol{x}_\lambda, X_{-\Lambda}'), \boldsymbol{s}_\theta(\boldsymbol{x}_\lambda, X_{-\Lambda}')_i^2 \right)) \tag{21}$$

$$+ 2(\hat{\mathbb{E}}_{iw}[\nabla_\theta \partial_i \boldsymbol{s}_\theta(\boldsymbol{x}_\lambda, X_{-\Lambda}')_i] + \hat{\text{Cov}}_{iw} \left( \nabla_\theta \log q_\theta(\boldsymbol{x}_\lambda, X_{-\Lambda}'), \partial_i \boldsymbol{s}_\theta(\boldsymbol{x}_\lambda, X_{-\Lambda}')_i \right))$$

*Proof.* Proof given in Section C.4. $\square$

### A.4. Exploring the Marginal Fisher Divergence for Normal Distributions

While intuitively the Fisher divergences of the marginal distributions should act as effective proxies for the Fisher divergence for the fully observed distributions, we would like to be able to examine the relationship between the two more explicitly. We do know that marginal Fisher divergences will be zero when then fully observed distributions equivalent however here we give a more detailed examination in the case of normal distributions.

Suppose that $X \sim N(\mu, P^{-1})$

$$p(x) = \exp\{-\frac{1}{2}(\boldsymbol{x} - \boldsymbol{\mu})^\top P(\boldsymbol{x} - \boldsymbol{\mu})\} + C$$

with $C$ a constant w.r.t. $\boldsymbol{x}$. We then have that

$$\boldsymbol{s}(\boldsymbol{x}) = -P(\boldsymbol{x} - \boldsymbol{\mu})$$

If we suppose that our unnormalised density/score model is of the form

$$q_\theta(\boldsymbol{x}) := \exp \left\{ -\frac{1}{2}(\boldsymbol{x} - \boldsymbol{\mu}_\theta)^\top P_\theta(\boldsymbol{x} - \boldsymbol{\mu}_\theta) \right\} \qquad \Rightarrow \boldsymbol{s}_\theta = P_\theta(\boldsymbol{x} - \boldsymbol{\mu}_\theta)$$

Then with the marginal Fisher taken to be

$$F_{\mathrm{M}}(\theta) = \mathbb{E}_{\Lambda, X_\Lambda} \left[ \| \boldsymbol{s}_\Lambda(X_\Lambda) \boldsymbol{s}_{\theta;\Lambda} \|^2 \right]$$

where here for each $\lambda \in \mathrm{supp}(\Lambda)$, $\boldsymbol{s}_\lambda, \boldsymbol{s}_{\theta;\lambda}$ are the true marginal scores. Using properties of the normal distribution and the Schur complement we know that the precision of $X_\lambda$ is given by

$$\left\{ \left( P^{-1} \right)_{\lambda,\lambda} \right\}^{-1} = P_{\lambda,\lambda} - P_{\lambda,-\lambda} P_{-\lambda,-\lambda}^{-1} P_{-\lambda,\lambda}.$$

Plugging this in we get

$$\begin{aligned}
F_{\mathrm{M}}(\theta) = \mathbb{E} \Bigg[ \Bigg\| & (P_{\Lambda,\Lambda} - P_{\theta;\Lambda,\Lambda}) X_\Lambda + (P_{\Lambda,-\Lambda} P_{-\Lambda,-\Lambda}^{-1} P_{-\Lambda,\Lambda} - P_{\theta;\Lambda,-\Lambda} P_{\theta;-\Lambda,-\Lambda}^{-1} P_{\theta;-\Lambda,\Lambda}) X_\Lambda \\
& - ((P_{\Lambda,\Lambda} - P_{\Lambda,-\Lambda} P_{-\Lambda,-\Lambda}^{-1} P_{-\Lambda,\Lambda}) \boldsymbol{\mu}_\Lambda - (P_{\theta;\Lambda,\Lambda} - P_{\theta;\Lambda,-\Lambda} P_{\theta;-\Lambda,-\Lambda}^{-1} P_{\theta;-\Lambda,\Lambda}) \boldsymbol{\mu}_{\theta;\Lambda}) \Bigg\|^2 \Bigg].
\end{aligned}$$

This shows why naive marginalisation by zeroing out missing coordinates of our score would not work as in this case the Fisher divergence would be given by

$$F_{\mathrm{M}}(\theta) = \mathbb{E} \left[ \left\| ((P_{\Lambda,\Lambda} - P_{\Lambda,-\Lambda} P_{-\Lambda,-\Lambda}^{-1} P_{-\Lambda,\Lambda}) - P_{\theta;\Lambda,\Lambda}) X_\Lambda - ((P_{\Lambda,\Lambda} - P_{\Lambda,-\Lambda} P_{-\Lambda,-\Lambda}^{-1} P_{-\Lambda,\Lambda}) \boldsymbol{\mu}_\Lambda - P_{\theta;\Lambda,\Lambda} \boldsymbol{\mu}_{\theta;\Lambda}) \right\|^2 \right]$$

which encourages $P_{\theta;\lambda,\lambda}$ to be close to $P_{\lambda,\lambda} - P_{\lambda,-\lambda} P_{-\lambda,-\lambda}^{-1} P_{-\lambda,\lambda}$ for all $\lambda \in \mathrm{supp}(\Lambda)$ meaning it will not give us the true density.

### A.5. Variational Pseudo-loss

When using (10) it is helpful to be able to view it as the gradient of some pseudo-loss allowing it to plug into a more standard ML framework where we calculate the loss, take the gradient w.r.t. our parameter using auto-differentiation, and update our parameter estimate based on this. The below result show how we can do this by creating a loss with certain instances of our parameter detached from the computational graph.

**Proposition A.20.** *Let*

$$\begin{aligned}
J(\theta, \theta', \boldsymbol{x}_\lambda, X'_{-\lambda}) := & -2\mathbb{E}'[\boldsymbol{s}_{\theta'}(\boldsymbol{x}_\lambda, X'_{-\lambda})_i] \left\{ \mathbb{E}'[\boldsymbol{s}_\theta(\boldsymbol{x}_\lambda, X'_{-\lambda})_i] + \mathrm{Cov}'\left( \log q_\theta(\boldsymbol{x}_\lambda, X'_{-\lambda}), \boldsymbol{s}_{\theta'}(\boldsymbol{x}_\lambda, X'_{-\lambda})_i \right) \right\} \\
& + 2(\mathbb{E}'[\boldsymbol{s}_\theta(\boldsymbol{x}_\lambda, X'_{-\lambda})_i^2] + \mathrm{Cov}'\left( \log q_\theta(\boldsymbol{x}_\lambda, X'_{-\lambda}), \boldsymbol{s}_{\theta'}(\boldsymbol{x}_\lambda, X'_{-\lambda})_i^2 \right)) \\
& + 2(\mathbb{E}'[\partial_i \boldsymbol{s}_\theta(\boldsymbol{x}_\lambda, X'_{-\lambda})_i] + \mathrm{Cov}'\left( \log q_\theta(\boldsymbol{x}_\lambda, X'_{-\lambda}), \partial_i \boldsymbol{s}_{\theta'}(\boldsymbol{x}_\lambda, X'_{-\lambda})_i \right))
\end{aligned}$$

*where $\mathbb{E}', \mathrm{Cov}'$ are w.r.t. $X'_{-\lambda}|X_\lambda = \boldsymbol{x}_\lambda; \theta'$ Then*

$$\nabla_{\theta'} L(\theta', \boldsymbol{x}_\lambda) = \left. \frac{\partial}{\partial \theta} J(\theta, \theta', \boldsymbol{x}_\lambda) \right|_{\theta=\theta'}$$

*Proof.* This just follows directly from the exchangeability of expectations and gradients (when the gradient is w.r.t. something independent of the expectation distribution.) $\qquad\square$

Hence we can use this loss (by replacing all instances of $\theta'$ with $\theta$ and then detaching them from the computation graph) to treat our problem as a standard gradient descent problem.

Note that while we can treat this like a loss for our optimisation, our intent is not actually to minimise it. The estimated form of the loss is given in the proof of Corollary 4.9 which is given in C.3 but we state it again explicitly here for convenience.

$$L_{\mathrm{M}}(\theta) = \mathbb{E} \left[ \sum_{j \in \Lambda} -\mathbb{E}'[s_\theta(X_\Lambda, X'_{-\Lambda})_j]^2 + 2\mathbb{E}'[s_\theta(X_\Lambda, X'_{-\Lambda})_j^2] + 2\mathbb{E}'[\partial_i s_\theta(X_\Lambda, X'_{-\Lambda})_j] \right]$$

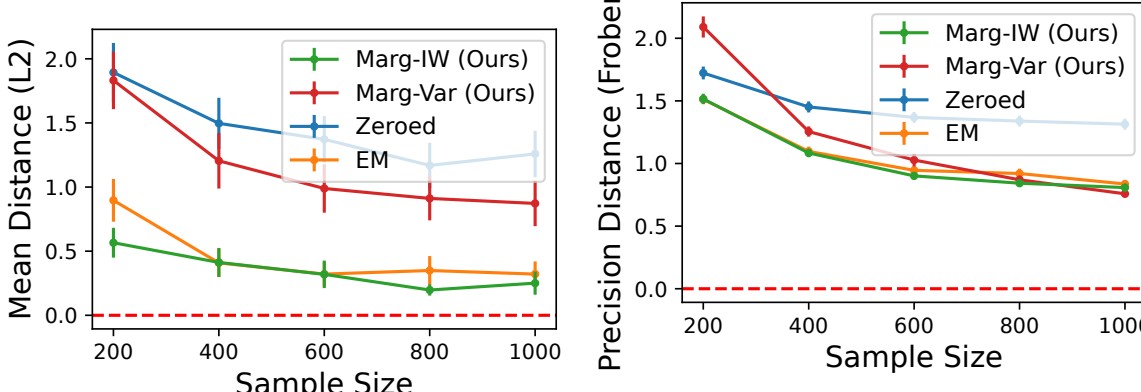

(a) Average error in estimation of the Mean (L2 norm).

(b) Average error in estimation of the Precision (Frobenius Norm).

Figure 8: Average parameter estimation error for truncated Gaussian score estimates alongside 95% confidence intervals under various methods.

## B. Additional Experimental Results

Here we present some additional experimental results not in the main body of the paper.

### B.1. Parameter Estimation

#### B.1.1. TRUNCATED GAUSSIAN MODEL

Here we present the accompanying mean and precision error results for Gaussian model estimation experiment presented in Section 5.1.1. These results are presented in Figure 8.

#### B.1.2. UNTRUNCATED GAUSSIAN MODEL

Here we present the untruncated version of the experiment presented in the main paper. Details of the distribution are the same as presented in Appendix E.3 but without the truncation.

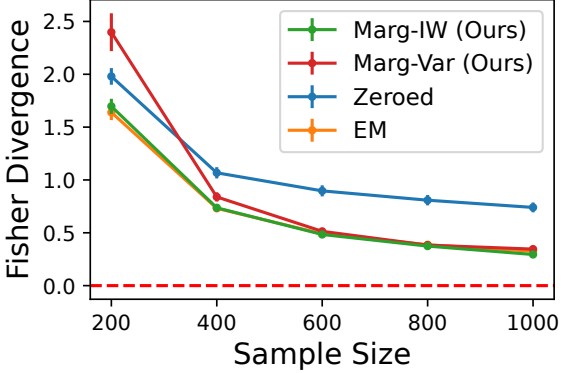

Figure 9: Average Fisher Divergence with 95% C.I.s for various approaches. Lower is better.

As we can see we obtains similar results here as in the truncated case.

We also illustrate what the true covariance and precision matrix look like for this example alongside the naive marginalisation in order to highlight where Zeroed Score Matching goes wrong.

In Figure 10 we can see the covariance and precision of a sample distribution where we can clearly see the strong dependence of dimensions 1 and 10 relative to the others.

In Figure 11 we can see the naive and true marginal precisions when dimension 1 is missing. For this plot the values have

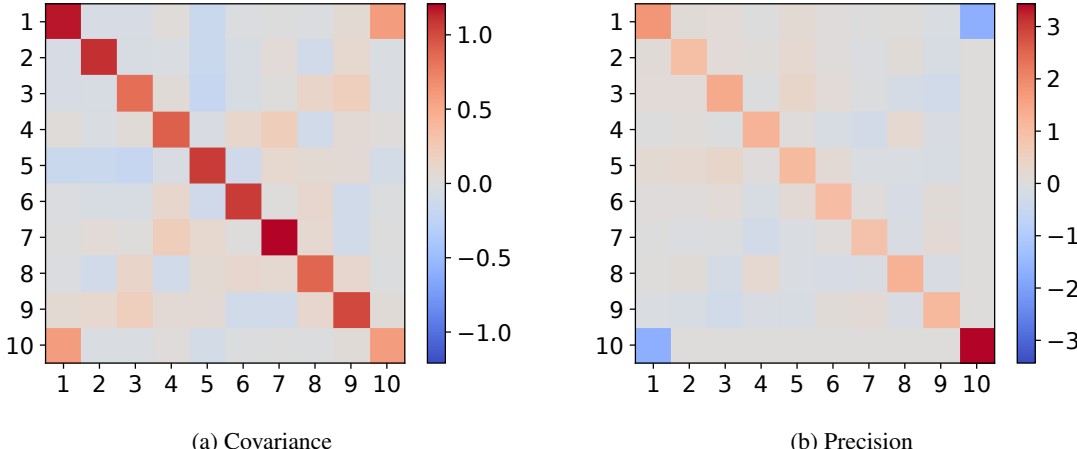

(a) Covariance

(b) Precision

Figure 10: Covariance and precision from a sample distribution from our normal experiment.

been cube-rooted in order to emphasize the difference between zero and non-zero entries. Here we can see that the naive marginalisation wouldn't capture the dependence between dimension 10 and the other dimensions that gets introduced when dimension 1 is removed. This means that a naive marginalisation would assume that dimension 10 must have a direct dependence on dimensions 2-9 even when that is not true. Interestingly, the rest of the marginalisation seems very similar suggesting that in some potentially less structured cases, naive marginalisation can provide a semi-reasonable approximation. This supports the results we see in our GGM estimation where highly structured graphs like star graphs are much more affected naive marginalisation than unstructured graphs.

### B.1.3. Non-Gaussian Estimation

Here we present further experiments exploring the non-Gaussian model presented in Section 5.1.2. Here we fix the dimension as 10. In Figure 12a we fix the missing probability as 0.5 and vary the sample size. In Figure 12b we fix the sample size as 1000 and vary the missing probability.

From Figure 12a we see that both EM and Marg-IW have the smallest estimation error for larger sample sizes. Zeroed Score Matching has the largest estimation error due to its inability to appropriately marginalise the distribution. In Figure 12b, we observe that Marg-Var has the smallest estimation error with its performance convergence to that of Marg-IW and EM as the missing probability increases.

### B.2. GGM Estimation

#### B.2.1. Varying Number of Star Centres

Here we present illustrations of the marginalisations for our star-shaped graphs with 1 node and then 5 nodes both with the same edge density.

In Figure 13 we show the covariance, precision, marginal covariance, and marginal precision for a star graph with 1 centre where the marginal terms are with dimension 1 removed. As we can see clearly the only meaningful structure left in the graph after marginalisation are in the negative precision terms which the model naive marginalisation fails to capture.

In Figure 14 we show the same thing for the case of a star graph with 5 centres. As we can see in the 5 centre case, the naive marginalisation picks up most of the structure as there are fewer negative terms which it ignores and also lots of additional positive terms which it does successfully pick up.

#### B.2.2. Varying Number of Dimensions

Here we use our same star-shaped GGM as in the main paper but with a varying number of dimensions. Throughout 1,000 samples are used and each coordinate is missing independently with probability 0.7. Results are presented in Figure 15.

As we can see, for higher dimensions the variational approach clearly performs best however at lower dimensions the other

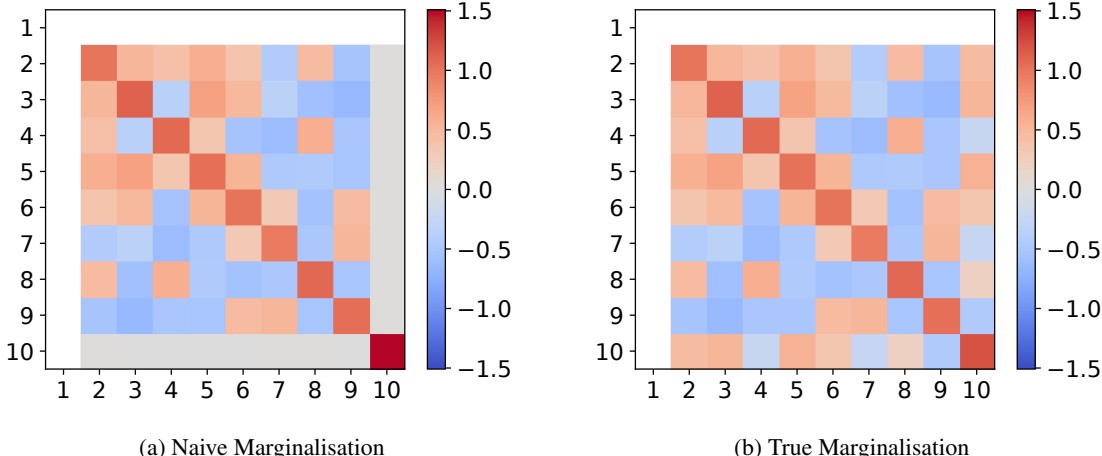

(a) Naive Marginalisation

(b) True Marginalisation

Figure 11: Marginalisation of the precision to remove dimension 1 by the naive approach (i.e. subsetting the precision) and the correct approach. All values cube-rooted for contrastive purposes.

approach catch up and even overtake it. This is because at lower dimensions, IW can effectively model the marginalisation of the score and so the more complicated variational approach is not required.

### B.2.3. INDIVIDUAL ROC CURVES
Here we present individual ROC curves from Section 5.2.1 with a missing probability of 0.5. Here we specifically present the ROC curves from the first 4 runs out of the 50 performed for the experiment. These ROC curves are displayed in Figure 16. We observe that Marg-Var has consistently the best ROC curves of any of the methods.

## C. Additional Proofs
### C.1. Marginal Score Matching Objectives
*proof of Proposition 4.3.* For the first claim we have that

$$\mathbb{E}[\|\boldsymbol{s}_{\Lambda;\theta}(X_\Lambda) - \boldsymbol{s}_\Lambda(X_\Lambda)\|^2] = \mathbb{E}[\|\boldsymbol{s}_{\Lambda;\theta}(X_\Lambda)\|^2] + \mathbb{E}[\|\boldsymbol{s}_\Lambda(X_\Lambda)\|^2] - 2\mathbb{E}[\boldsymbol{s}_{\Lambda;\theta}(X_\Lambda)^\top s_\Lambda(X_\Lambda)]$$

$$= C + \mathbb{E}[\|\boldsymbol{s}_{\Lambda;\theta}(X)\|^2] - 2\sum_{\lambda\in\mathrm{supp}(\Lambda)} \mathbb{P}(\Lambda = \lambda)\int_{X_\lambda} p_\lambda(\boldsymbol{x}_\lambda)\boldsymbol{s}_{\lambda;\theta}(\boldsymbol{x}_\lambda)^\top s_\lambda(\boldsymbol{x}_\lambda)\mathrm{d}\boldsymbol{x}_\lambda$$

$$= C + \mathbb{E}[\|\boldsymbol{s}_{\Lambda;\theta}(X_\Lambda)\|^2] - 2\sum_{\lambda\in\mathrm{supp}(\Lambda)} \mathbb{P}(\Lambda = \lambda)\int_{\mathcal{X}} \nabla_{\boldsymbol{x}_\lambda} p_\lambda(\boldsymbol{x}_\lambda)^T \boldsymbol{s}_{\lambda;\theta}(\boldsymbol{x}_\lambda)\mathrm{d}\boldsymbol{x}_\lambda$$

$$= C + \mathbb{E}[\|\boldsymbol{s}_{\Lambda;\theta}(X_\Lambda)\|^2] - 2\sum_{\lambda\in\mathrm{supp}(\Lambda)} \mathbb{P}(\Lambda = \lambda)\sum_{j=1}^d \int_{\mathcal{X}_\Lambda} \nabla_{\boldsymbol{x}_\lambda} p_\lambda(\boldsymbol{x}_\lambda)_j \boldsymbol{s}_{\lambda;\theta}(\boldsymbol{x}_\lambda)_j \mathrm{d}\boldsymbol{x}_\lambda$$

Hence by integration by parts we have

$$\mathbb{E}[\|\boldsymbol{s}_{\Lambda;\theta}(X_\Lambda) - \boldsymbol{s}_\Lambda(X_\Lambda)\|^2] = C + \mathbb{E}[\|\boldsymbol{s}_{\Lambda;\theta}(X_\Lambda)\|^2]$$

$$- 2\sum_{\lambda\in\mathrm{supp}(\Lambda)} \mathbb{P}(\Lambda = \lambda)\sum_{j\in\lambda}\left\{\lim_{\boldsymbol{x}_j \longrightarrow \partial\mathcal{X}} p_\lambda(\boldsymbol{x}_\lambda)\boldsymbol{s}_{\lambda;\theta}(\boldsymbol{x}_\lambda) - \int_{\mathcal{X}_\lambda} p_\lambda(\boldsymbol{x}_\lambda)_j \boldsymbol{s}_{\lambda;\theta}(\boldsymbol{x}_\lambda)_j \mathrm{d}\boldsymbol{x}_\lambda\right\}$$

$$= \mathbb{E}[\|\boldsymbol{s}_{\Lambda;\theta}(X_\Lambda)\|^2] + 2\mathbb{E}[\nabla_{X_\Lambda} \cdot \boldsymbol{s}_{\Lambda;\theta}(X_\Lambda))] + C$$

justifying our first claim. Hence if $\tilde{\theta}$ minimised our objective then it minimises

$$\mathbb{E}[\|\boldsymbol{s}_{\Lambda;\theta}(X_\Lambda) - \boldsymbol{s}_\Lambda(X_\Lambda)\|^2].$$

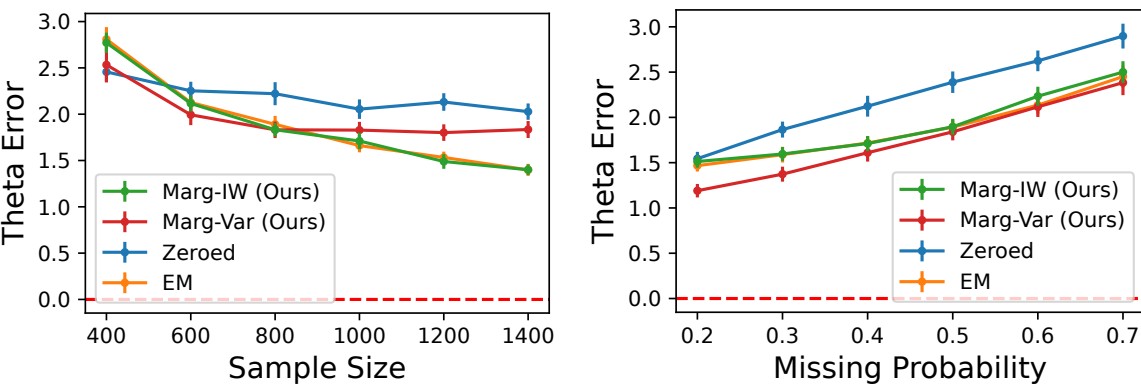

(a) Varying sample size with missing probability 0.5 and dimension 10.

(b) Varying missing probability with sample size 1000 and dimension 10.

Figure 12: Average parameter estimation error (L2 norm) for ICA inspired model with 95% C.I.s for various methods. Lower is better

As there exists a "true" $\theta$ we know that this objective is minimised at 0 and so we must have $\boldsymbol{s}_{\lambda;\tilde{\theta}}(X_\lambda) = \boldsymbol{s}_\lambda(X_\lambda)$ a.s. for all $\lambda \in \operatorname{supp}(\Lambda)$. By our assumption this then gives $p_{\tilde{\theta}}(X) = p(X)$ a.s. . $\qquad\square$

### C.1.1. TRUNCATED SCORE MATCHING

*Proof of Proposition A.2.* We mostly use (Liu et al., 2022). We firstly have that

$$\mathbb{E}[g_\Lambda(X_\Lambda)\|\boldsymbol{s}_{\Lambda;\theta}(X_\Lambda) - \boldsymbol{s}_\Lambda(X_\Lambda)\|^2]$$
$$= \mathbb{E}[g_\Lambda(X_\Lambda)\|\boldsymbol{s}_\Lambda(X_\Lambda)\|^2] + \mathbb{E}[g_\Lambda(X_\Lambda)\|\boldsymbol{s}_{\Lambda;\theta}(X_\Lambda)\|^2] - 2\mathbb{E}[g_\Lambda(X_\Lambda)\boldsymbol{s}_{\Lambda;\theta}(X_\Lambda)^\top \boldsymbol{s}_\Lambda(X_\Lambda)]$$

Now we have that

$$\mathbb{E}[g_\Lambda(X_\Lambda)\boldsymbol{s}_{\Lambda;\theta}(X_\Lambda)^\top \boldsymbol{s}_\Lambda(X_\Lambda)]$$
$$= \sum_{\lambda \in \operatorname{supp}(\Lambda)} \mathbb{P}(\Lambda = \lambda)\mathbb{E}[g_\lambda(X_\Lambda)\boldsymbol{s}_{\Lambda;\theta}(X_\Lambda)^\top \boldsymbol{s}_\Lambda(X_\Lambda)|\Lambda = \lambda]$$
$$= \sum_{\lambda \in \operatorname{supp}(\Lambda)} \mathbb{P}(\Lambda = \lambda) \sum_{j \in \lambda} \int_{\mathcal{X}} g_\lambda(\boldsymbol{x}_\lambda)\boldsymbol{s}_{\theta;\lambda}(\boldsymbol{x}_\lambda)\frac{\partial}{\partial \boldsymbol{x}_j}p_\lambda(\boldsymbol{x}_\lambda)\mathrm{d}\boldsymbol{x}_\lambda$$
$$= \sum_{\lambda \in \operatorname{supp}(\Lambda)} \mathbb{P}(\Lambda = \lambda) \sum_{j \in \lambda} \int_{\mathcal{X}} g_\lambda(\boldsymbol{x}_\lambda)\boldsymbol{s}_{\theta;\lambda}(\boldsymbol{x}_\lambda)_j\frac{\partial}{\partial \boldsymbol{x}_j}p_\lambda(\boldsymbol{x}_\lambda)\mathrm{d}\boldsymbol{x}_\lambda$$
$$\overset{(a)}{=} \sum_{\lambda \in \operatorname{supp}(\Lambda)} \mathbb{P}(\Lambda = \lambda) \sum_{j \in \lambda} \left\{ \int_{\partial X_\lambda} g_\lambda(\boldsymbol{x}_\lambda)\boldsymbol{s}_{\theta;\lambda}(\boldsymbol{x}_\lambda)_j p_\lambda(\boldsymbol{x}_\lambda)v_j(\boldsymbol{x}_\lambda)\mathrm{d}s - \int_{\mathcal{X}} \frac{\partial}{\partial \boldsymbol{x}_j}[g_\lambda(\boldsymbol{x}_\lambda)\boldsymbol{s}_{\theta;\lambda}(\boldsymbol{x}_\lambda)_j] p_\lambda(\boldsymbol{x}_\lambda)\mathrm{d}\boldsymbol{x}_\lambda \right\}$$
$$\overset{(b)}{=} -\sum_{\lambda \in \mathcal{P}(d)} \mathbb{P}(\Lambda = \lambda) \sum_{j \in \lambda} \int_{\mathcal{X}} \left[ g_\lambda(\boldsymbol{x}_\lambda)\frac{\partial}{\partial \boldsymbol{x}_j}\boldsymbol{s}_{\theta;\lambda}(\boldsymbol{x}_\lambda)_j + \frac{\partial}{\partial \boldsymbol{x}_k}g_\lambda(\boldsymbol{x}_\lambda)\boldsymbol{s}_{\lambda;\theta}(\boldsymbol{x}_\lambda) \right] p_\lambda(\boldsymbol{x}_\lambda)\mathrm{d}\boldsymbol{x}_\lambda$$
$$= -\mathbb{E}[\nabla_{X_\Lambda} g_\Lambda(X_\Lambda)^T \boldsymbol{s}_{\Lambda;\theta}(X_\Lambda) + g_\Lambda(X_\Lambda)\boldsymbol{s}_{\Lambda;\theta}(X_\Lambda)]$$

where $(a)$ is given by Green's Theorem and $(b)$ is given by our limiting condition. Plugging this back into our original result gives

$$\mathbb{E}[g_\Lambda(X_\Lambda)\|\boldsymbol{s}_{\Lambda;\theta}(X_\Lambda) - \boldsymbol{s}_\Lambda(X_\Lambda)\|^2] = \mathbb{E}[g_\Lambda(X_\Lambda)\left(\|\boldsymbol{s}_{\Lambda;\theta}(X_\Lambda)\|^2 + 2\nabla_{X_\Lambda} \cdot \boldsymbol{s}_{\Lambda;\theta}(X_\Lambda)\right) + \nabla_{X_\Lambda} g_\Lambda(X_\Lambda)^\top \boldsymbol{s}_{\Lambda;\theta}(X_\Lambda)]$$
$$=: L_{\mathrm{TM}}(\theta)$$

From our conditions we also know that $\theta^*$ is a minimiser of $J(\theta)$ hence by our conditions it is the unique minimiser. Therefore $\theta^*$ is the unique minimiser of $L_{\mathrm{TM}}(\theta)$ $\qquad\square$

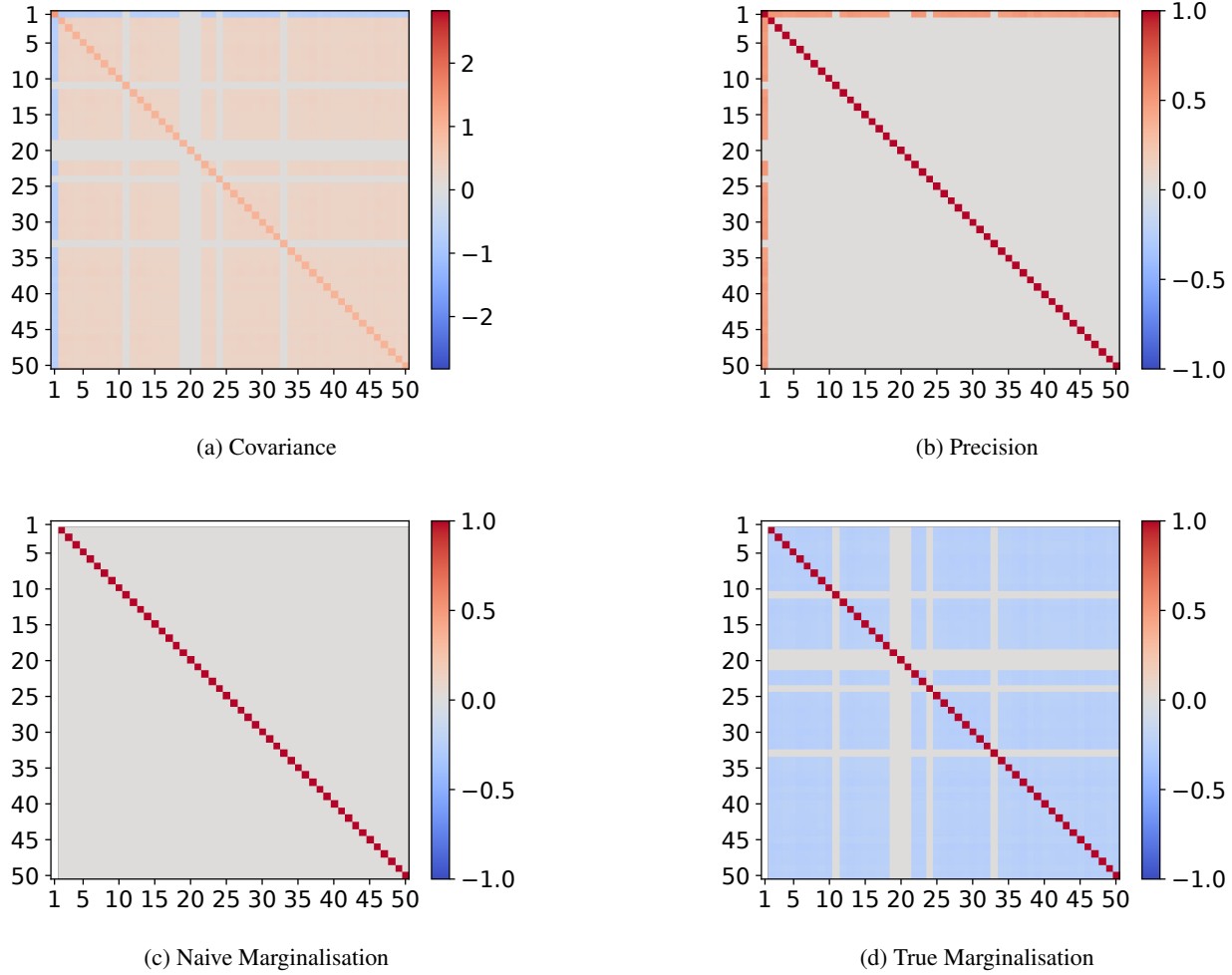

(a) Covariance

(b) Precision

(c) Naive Marginalisation

(d) True Marginalisation

Figure 13: Covariance, precision, and marginalisations of the precisions to remove dimension 1 by the naive approach (i.e. subsetting the precision) and the correct approach. All values cube-rooted for contrastive purposes.

### C.1.2. MNAR PROOFS

*Proof of Proposition A.9.* Firstly we have that

$$\mathbb{E}\left[\|\boldsymbol{s}_{\Lambda,E_\Lambda}(X_\Lambda) - \boldsymbol{s}_{\Lambda,E_\Lambda;\theta}(X_\Lambda)\|^2\right] = \mathbb{E}\left[\|\boldsymbol{s}_{\Lambda,E_\Lambda;\theta}(X_\Lambda)\|^2 - 2\boldsymbol{s}_{\Lambda,E_\Lambda}(X_\Lambda)^\top \boldsymbol{s}_{\Lambda,E_\Lambda;\theta}(X_\Lambda)\right] + C$$

where $C$ does not depend upon $\theta$. Examining the second term closer we see that

$$\sum_{\lambda\in\text{supp}(\Lambda)} \int_{\mathcal{X}_\lambda} \boldsymbol{s}_\lambda(\boldsymbol{x}_\lambda)^\top \boldsymbol{s}_{\lambda;\theta}(\boldsymbol{x}_\lambda)p_\lambda(\boldsymbol{x}_\lambda; E_\lambda)\mathrm{d}\boldsymbol{x}_\lambda$$

$$= \sum_{\lambda\in\text{supp}(\Lambda)} \sum_{j\in\lambda} \int_{\mathcal{X}_\lambda} \boldsymbol{s}_\lambda(\boldsymbol{x}_\lambda)_j \boldsymbol{s}_{\lambda,\theta}(\boldsymbol{x}_\lambda)_j p_\lambda(\boldsymbol{x}_\lambda; E_\lambda)\mathrm{d}\boldsymbol{x}_\lambda$$

$$= \sum_{\lambda\in\text{supp}(\Lambda)} \sum_{j\in\lambda} \int_{\mathcal{X}_\lambda} \nabla_{\boldsymbol{x}_\lambda} p_\lambda(\boldsymbol{x}_\lambda; E_\lambda)_j \boldsymbol{s}_{\lambda;\theta}(\boldsymbol{x}_\lambda)_j \mathrm{d}\boldsymbol{x}_\lambda$$

$$= \sum_{\lambda\in\text{supp}(\Lambda)} \sum_{j\in\lambda} -\int_{\mathcal{X}_\lambda} p_\lambda(\boldsymbol{x}_\lambda; E_\lambda)\frac{\partial}{\partial_j}\boldsymbol{s}_{\lambda;\theta}(\boldsymbol{x}_\lambda)_j \mathrm{d}\boldsymbol{x}_\lambda \quad \text{as a result of integration by parts}$$

$$= \mathbb{E}_{X,\Lambda}\left[\nabla_{X_\Lambda} \cdot \boldsymbol{s}_{\Lambda;\theta}(X_\Lambda)\right]$$

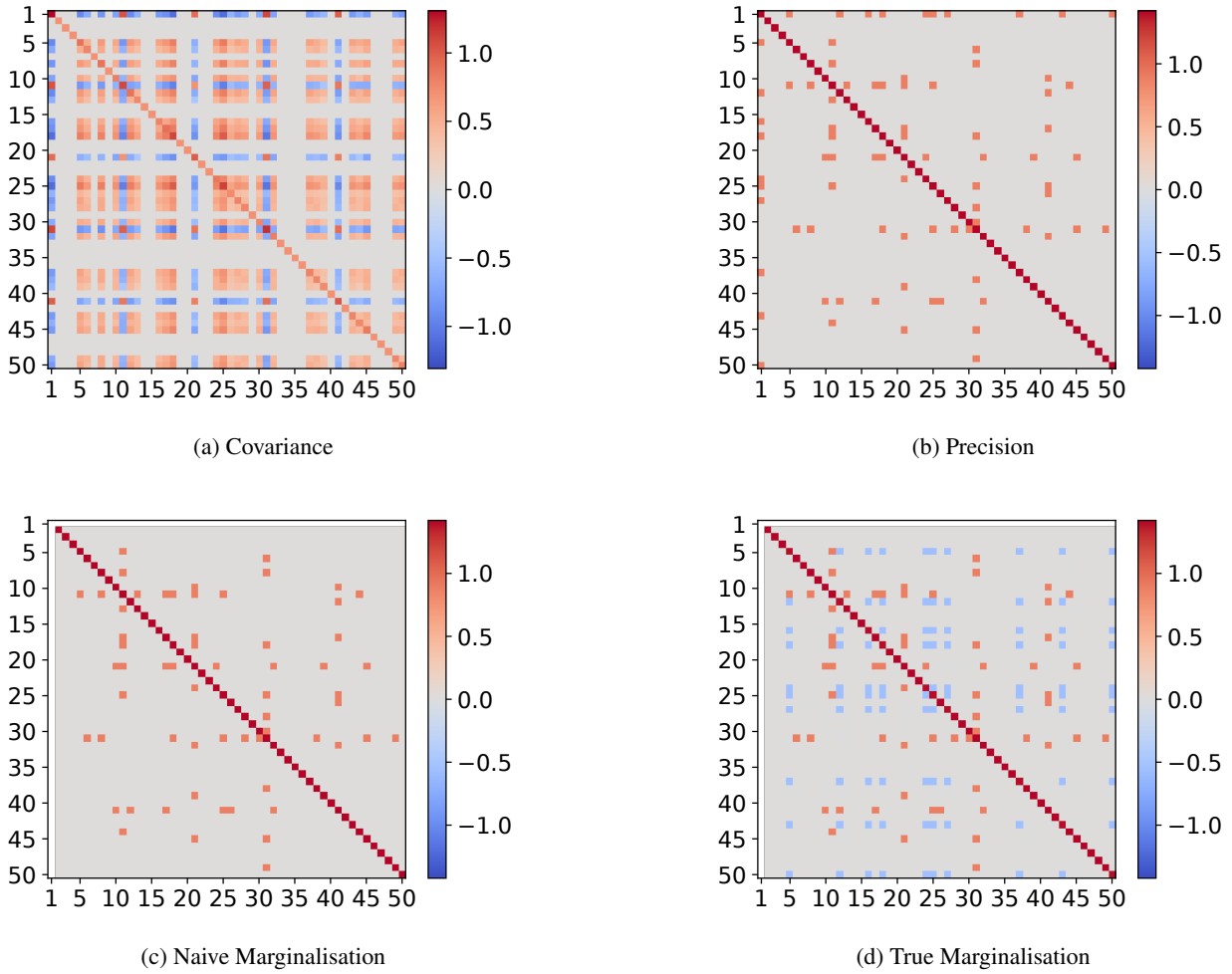

(a) Covariance

(b) Precision

(c) Naive Marginalisation

(d) True Marginalisation

Figure 14: Covariance, precision, and marginalisations of the precisions to remove dimension 1 by the naive approach (i.e. subsetting the precision) and the correct approach. All values cube-rooted for contrastive purposes.

Hence we have

$$\mathbb{E}\left[\|\boldsymbol{s}_\Lambda(X_\Lambda) - \boldsymbol{s}_{\Lambda, E_\Lambda;\theta}(X_\Lambda)\|^2\right] = \mathbb{E}\left[\|\boldsymbol{s}_{\Lambda;\theta}(X_\Lambda)\|^2 + 2\nabla_{X_\Lambda} \cdot \boldsymbol{s}_{\Lambda;\theta}(X_\Lambda)\right] + C$$

Hence, just as in Proposition 4.3 this is minimised when

$$\boldsymbol{s}_\lambda(X_\lambda) = \boldsymbol{s}_{\lambda;\theta}(X_\lambda)$$

a.s. for all $\lambda \in \mathrm{supp}(\Lambda)$. We then have that for any $\lambda \in \mathrm{supp}(\Lambda)$

$$\boldsymbol{s}_\lambda(\boldsymbol{x}_\lambda) = \boldsymbol{s}_{\lambda;\theta}(\boldsymbol{x}_\lambda) \quad \text{for all } x_\lambda \in \mathcal{X}_\lambda$$
$$p_{\lambda;E_\lambda}(\boldsymbol{x}_\lambda) = p_{E_\lambda;\theta}(\boldsymbol{x}_\lambda) \quad \text{for all } x_\lambda \in \mathcal{X}_\lambda$$
$$\Leftrightarrow \mathbb{P}(E_\Lambda|x_\Lambda)p_\lambda(\boldsymbol{x}_\lambda) = p_{\lambda;\theta}(\boldsymbol{x}_\lambda)\mathbb{P}(E_\Lambda|x_\Lambda) \quad \text{for all } x_\lambda \in \mathcal{X}_\lambda$$
$$\Leftrightarrow p_\lambda(\boldsymbol{x}_\lambda) = p_{\lambda;\theta}(\boldsymbol{x}_\lambda) \quad \text{for all } x_\lambda \in \mathcal{X}_\lambda$$

□

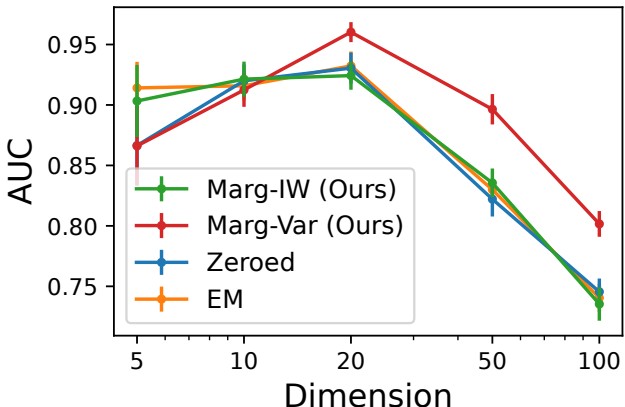

Figure 15: Mean AUC of various methods for edge detection of star-shaped GGM as we increase the dimension presented alongside 95% confidence intervals.

### C.2. Finite Sample Bound Proofs

Here we give some key results alongside their proof to allow us to obtains finite sample bounds for truncated score matching. This first result is what really underpins our approach.

**Proposition C.1.** *Let $(\Theta, d)$ be a compact metric space and for any $\delta > 0$ denote the $\eta$-covering number of $\Theta$ by $N(\eta, \Theta)$. Now define random functions $Z(\theta)$, $Z_r(\theta)\Theta \to \mathbb{R}$ for all $r \in \mathbb{N}$ (with $r$ deterministic) such that for any $\theta \in \Theta$*

$$\mathbb{P}\left(|Z_r(\theta) - Z(\theta)| > \varepsilon\right) \leq \delta(\varepsilon, r)$$

*If we additionally assume that $Z_r, Z$ Lipschitz with constants $C_r, C$ respectively then we have then*

$$\mathbb{P}\left(\sup_{\theta \in \Theta}|Z_r(\theta) - Z(\theta)| > \varepsilon + \eta(C_r + C)\right) \leq N(\eta, \Theta)\delta(\varepsilon, r)$$

*Proof.* Let $\theta_1, \ldots, \theta_{N(\eta, \Theta)}$ be an $\eta$ cover of $\Theta$ then we have that

$$\sup_{\theta \in \Theta}|Z_r(\theta) - Z(\theta)|$$

$$\leq \sup_{\theta \in \Theta}\left\{\min_{j \in [N(\eta, \Theta)]}\left\{|Z_r(\theta) - Z_r(\theta_l))| + |(Z(\theta) - Z(\theta_l)| + |Z_r(\theta_l) - Z(\theta_l)|\right\}\right\}$$

$$\leq \sup_{\theta \in \Theta}\left\{\min_{j \in [N(\eta, \Theta)]}\left\{|Z_r(\theta) - Z_r(\theta_l)| + |Z(\theta) - Z(\theta_l)|\right\}\left\{\max_{j \in [N(\eta, \Theta)]}|Z_r(\theta_l) - Z(\theta|\right\}\right\}$$

$$\leq \sup_{\theta \in \Theta}\left\{\min_{j \in [N(\eta, \Theta)]}\left\{(C_r + C)|\theta - \theta_l|\right\}\right\} + \max_{j \in [N(\eta, \Theta)]}\left\{|Z_r(\theta_l) - Z(\theta_l)|\right\} \quad \text{by our Lipschitz condition}$$

$$\leq (C_r + C)\eta + \max_{j \in [N(\eta, \Theta)]}\left\{|Z_r(\theta_l) - Z(\theta_l)|\right\} \quad \text{by definition of } \theta_1, \ldots, \theta_{N(\eta, \Theta)}$$

Therefore we have the following relationship between events:

$$\left\{\sup_{\theta \in \Theta}|Z_r(\theta) - Z(\theta)| > \varepsilon + 2M\eta\right\} \subseteq \left\{\bigcup_{j \in [N(\eta, \Theta)]}|Z_r(\theta_l) - Z(\theta_l)| > \varepsilon\right\}.$$

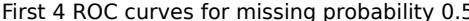

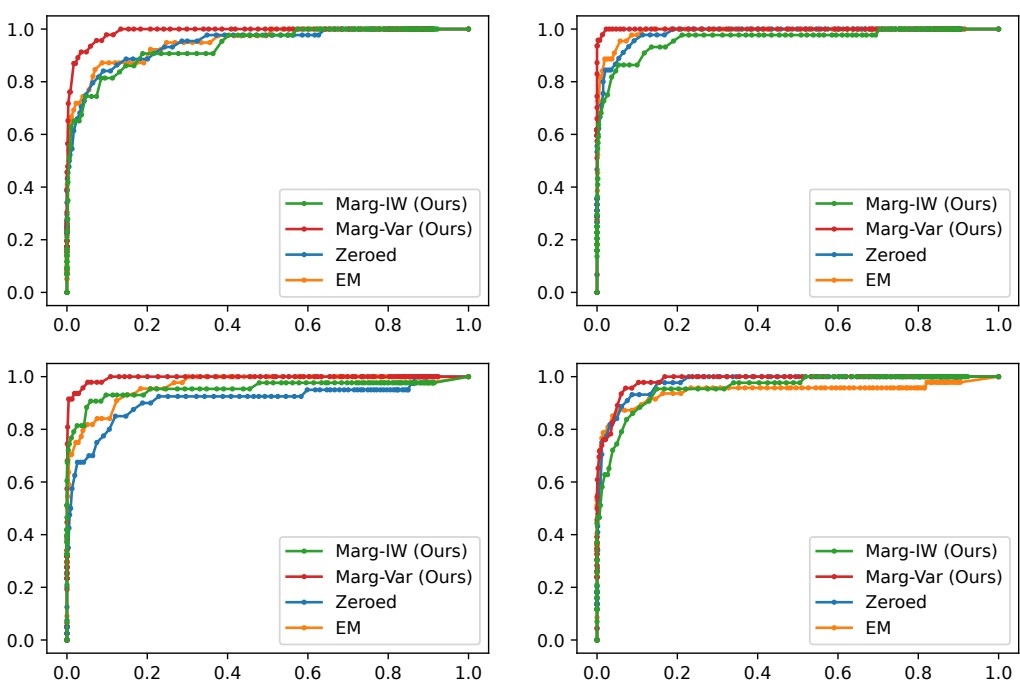

Figure 16: ROC Curves in 4 separate runs for GGM estimation of a model truncated normal distribution with a star-shaped Precision matrix.

This therefore gives

$$\mathbb{P}\left(\sup_{\theta \in \Theta}\left|\frac{1}{n}f(X^{(i)},\theta) - \mathbb{E}[f(X,\theta)]\right| > \varepsilon + \eta(C_r + C)\right)$$

$$\leq \mathbb{P}\left(\bigcup_{j \in [N(\eta,\Theta)]}|Z_r(\theta_l) - Z(\theta_l)| > \varepsilon\right)$$

$$\leq \sum_{j=1}^{N(\eta,\Theta)}\mathbb{P}\left(|Z_r(\theta_l) - Z(\theta_l)| > \varepsilon\right)$$

$$\leq N(\eta,\theta)\delta(\varepsilon,n).$$

$\square$

*Remark* C.2. This result does not require $C_r, C$ too be deterministic. A feature that we will be exploiting later on.

To be able to say meaningful statements about our Lipschitz bounds for our proof, it will also be helpful to make subgaussian statements about nested sums. We give to results to enable this now

**Lemma C.3.** *Let $X, Y$ be independent RVs and define a function $g : \mathcal{X} \times \mathcal{Y} \to \mathbb{R}$. Now suppose that for any $x \in \mathcal{X}$, $g(x, Y)$ is sub-Gaussian with parameter $\sigma$. We then have that $\mathbb{E}[g(X,Y)|Y]$ is sub-Gaussian with parameter $\sigma$*

*Proof.* As $g(x, Y)$ sub-Gaussian we have that for any $\lambda > 0$

$$\mathbb{E}\left[\exp\left\{\lambda\left(g(x,Y) - \mathbb{E}[g(x,Y)]\right)\right\}\right] \leq \exp\left\{\frac{\lambda^2}{\sigma^2}\right\}.$$

Our aim is to then use this to bound

$$\mathbb{E}\left[\exp\left\{\lambda\left(\mathbb{E}[g(X,Y)|Y] - \mathbb{E}[g(X,Y)]\right)\right\}\right].$$

We first have that $\mathbb{E}[g(X,Y)] = \mathbb{E}[\mathbb{E}[g(X,Y)|X]] = \mathbb{E}[\mathbb{E}[g(X,Y)|X]|Y]$ which in turn gives

$$\begin{aligned}
\mathbb{E}\left[\exp\left\{\lambda\left(\mathbb{E}[g(X,Y)|Y] - \mathbb{E}[g(X,Y)]\right)\right\}\right] &= \mathbb{E}\left[\exp\left\{\lambda\left(\mathbb{E}[g(X,Y) - \mathbb{E}[g(X,Y)|X]|Y]\right)\right\}\right] \\
&\leq \mathbb{E}\left[\mathbb{E}\left[\exp\left\{\lambda\left(g(X,Y) - \mathbb{E}[g(X,Y)|X]\right)\right\}|Y\right]\right] \quad \text{by Jensen's inequality} \\
&= \mathbb{E}\left[\mathbb{E}\left[\exp\left\{\lambda\left(g(X,Y) - \mathbb{E}[g(X,Y)|X]\right)\right\}|X\right]\right] \\
&\leq \sup_{x \in \mathcal{X}} E\left[\exp\left\{\lambda\left(g(x,Y) - \mathbb{E}[g(x,Y)]\right)\right\}\right] \quad \text{by independence} \\
&\leq \exp\left\{\frac{\lambda^2}{\sigma^2}\right\}.
\end{aligned}$$

$\square$

**Lemma C.4.** *Let $X, Y$ be independent RVs on spaces $\mathcal{X}, \mathcal{Y}$ and $g : \mathcal{X} \times \mathcal{Y} \to \mathbb{R}$ s.t. for any $x \in \mathcal{X}, y \in \mathcal{Y}$ $g(x, Y)$ and $g(X, y)$ are sub-Gaussian with parameters $\sigma_Y, \sigma_X$ respectively. Let $\left\{X^{(i)}\right\}_{i=1}^n$ and $\left\{Y^{(k)}\right\}_{k=1}^r$ be IID copies of $X, Y$ respectively then*

$$\mathbb{P}\left(\left|\frac{1}{nr}\sum_{i=1}^n\sum_{k=1}^r g(X^{(i)},Y^{(k)}) - \mathbb{E}[g(X,Y)]\right| > \varepsilon\right) \leq n\exp\left\{-\frac{\varepsilon^2\sigma_Y^2 m}{4}\right\} + \exp\left\{-\frac{\varepsilon^2\sigma_X^2 n}{4}\right\}$$

*Proof.* Again let $W^{(i)} := \left|\frac{1}{r}\sum_{i=1}^r g(X^{(i)},Y^{(k)}) - \mathbb{E}[g(X^{(i)},Y)|X^{(i)}]\right|$, We aim to bound $W^{(i)}$ as well as $\left|\frac{1}{n}\sum_{i=1}^n \mathbb{E}[g(X^{(i)},Y)|X^{(i)}] - \mathbb{E}[g(X,Y)]\right|$

For $W^{(i)}$ we have that

$$
\begin{aligned}
\mathbb{P}(W^{(i)} > \varepsilon) &= \mathbb{E}[\mathbb{P}(W^{(i)} > \varepsilon | X^{(i)})] \\
&= \mathbb{E}\left[ \mathbb{P}\left( \left| \frac{1}{r} \sum_{i=1}^{r} g(X^{(i)}, Y^{(k)}) - \mathbb{E}[g(X^{(i)}, Y)|X^{(i)}] \right| < \varepsilon \middle| X^{(i)} \right) \right] \\
&= \mathbb{E}\left[ \exp\left\{ -\frac{\varepsilon^2 \sigma_0 r}{2} \right\} \right] = \exp\left\{ -\frac{\varepsilon^2 \sigma_0 r}{2} \right\}
\end{aligned}
$$

Therefore we have that

$$
\begin{aligned}
\mathbb{P}\left( \frac{1}{n} \sum_{i=1}^{n} W^{(i)} > \varepsilon \right) &\leq \mathbb{P}\left( \bigcup_{i=1}^{n} \left\{ W^{(i)} > \varepsilon \right\} \right) \\
&\leq \sum_{i=1}^{n} \mathbb{P}(W^{(i)} > \varepsilon) \\
&\leq n \exp\left\{ -\frac{\varepsilon^2 \sigma_Y r}{2} \right\}.
\end{aligned}
$$

Additionally from Lemma C.3 we have that $\mathbb{E}[g(X,Y)|X]$ is sub-Gaussian with parameter $\sigma_1$ giving us that

$$
\mathbb{P}\left( \left| \frac{1}{n} \sum_{i=1}^{n} \mathbb{E}[g(X^{(i)}, Y|X^{(i)}] - \mathbb{E}[g(X,Y)] \right| > \varepsilon \right) \leq \exp\left\{ -\frac{\varepsilon^2 \sigma_X^2 n}{2} \right\}
$$

Hence combining these we get

$$
\mathbb{P}\left( \left| \frac{1}{nr} \sum_{i=1}^{n} \sum_{k=1}^{r} g(X^{(i)}, Y^{(k)}) - \mathbb{E}[g(X,Y)] \right| > \varepsilon \right) \leq n \exp\left\{ -\frac{\varepsilon^2 \sigma_Y^2 r}{8} \right\} + \exp\left\{ -\frac{\varepsilon^2 \sigma_X^2 n}{8} \right\}
$$

$\square$

To proceed we define the intermediary step between the population objective and the sample objective this is

$$
L_{\text{TM};n}(\theta) := \frac{1}{n} \sum_{i=1}^{n} \nabla_{X_{\Lambda_i}^{(i)}} \cdot \boldsymbol{s}_{\Lambda_i;\theta}(X^{(i)}) + \frac{1}{2} \| \boldsymbol{s}_{\Lambda_i;\theta}(X^{(i)}) \|^2
$$

**Proposition C.5.** *For $r \in \mathbb{N}$ let $\{X^{'(k)}\}_{k=1}^{r}$ be IID copies of $X' \sim p'$ and assume that $\text{supp}(X) \subseteq \text{supp}(X')$. For $\theta \in \Theta$ and $q_\theta : \mathcal{X} \to [0, \infty)$, with $\|q_\theta\|_1 < \infty$ define RVs $Y_\theta, Y_{\theta,r}$ by*

$$
Z(\theta) := L_{\text{TM};1} = g_\Lambda(X_\Lambda) \left( \nabla_{X_\Lambda} \cdot \boldsymbol{s}_{\Lambda,\theta}(X) + \frac{1}{2} \| \boldsymbol{s}_{\Lambda,\theta}(X) \|^2 \right) + \nabla_{X_\Lambda} g_\Lambda(X_\Lambda)^\top \boldsymbol{s}_{\Lambda;\Theta}(X_\Lambda)
$$

$$
Z_r(\theta) := L_{\text{TM};1,r}(\theta) = g_\Lambda \left( \text{tr}\left( \nabla_{X_\Lambda} \hat{\boldsymbol{s}}_{\Lambda,r;\theta}(X) \right) + \frac{1}{2} \| \hat{\boldsymbol{s}}_{\Lambda,r;\theta}(X_\Lambda) \|^2 \right) + \frac{1}{2} \nabla_{X_\Lambda} g_\Lambda(X_\Lambda)^\top \boldsymbol{s}_{\Lambda;\theta}(X_\Lambda)
$$

*Suppose that the following hold for all $\boldsymbol{x} \in \mathcal{X}$, $\lambda \in \text{supp}(\Lambda)$*

- $0 < a_0 < f_{0,\lambda}(\boldsymbol{x}, \theta) < b_0$
- $\| f_{1,\lambda}(\boldsymbol{x}, \theta) \| < b_1$
- $| \boldsymbol{f}_{2,\lambda}(\boldsymbol{x}, \theta) | < b_2$
- $g_\lambda(\boldsymbol{x}_\lambda) < c_0$

- $\|\nabla_{\boldsymbol{x}_\lambda} g_\lambda(\boldsymbol{x}_\lambda)\| < c_1$

*Then we have that*

$$\mathbb{P}(|Z_r(\theta) - Z(\theta)| > \varepsilon) \leq (4 + 2d) \exp\left\{-\frac{\varepsilon^2 m}{\alpha^2}\right\}$$

*with $\alpha$ depending upon $a_0, b_0, b_1, b_2, c_0, c_1$.*

*Proof.* First define

$$Y_l(\theta) := \mathbb{E}[f_{l,\Lambda}(X_\Lambda, X'_{-\Lambda}, \theta)|X_\Lambda, \Lambda] \qquad\qquad Y_{l,r}(\theta) := \frac{1}{r}\sum_{k=1}^{r} f_{l,\Lambda}(X_\Lambda, X'^{(k)}_{-\Lambda}).$$

Using the definition of the marginal and estimated marginal score we then have that

$$Z(\theta) = g_\Lambda(X_\Lambda)\frac{Y_2(\theta)}{Y_0(\theta)} + \frac{1}{2}\frac{\|Y_1(\theta)\|^2}{Y_0(\theta)^2} + \frac{1}{2}\nabla_{X_\Lambda} g_\Lambda(X_\Lambda)^\top \frac{Y_1(\theta)}{Y_0(\theta)}$$

$$Z_r(\theta) = g_\Lambda(X_\Lambda)\left(\frac{Y_{2,r}(\theta)}{Y_{0,r}(\theta)} + \frac{1}{2}\frac{\|Y_{1,r}(\theta)\|^2}{Y_{0,r}(\theta)^2}\right) + \frac{1}{2}\nabla_{X_\Lambda} g_\Lambda(X_\Lambda)^\top \frac{Y_1(\theta)}{Y_0(\theta)}.$$

We can therefore write write $|Z_r(\theta) - Z(\theta)|$ as

$$|Z_r(\theta) - Z(\theta)| = g_\Lambda(X_\Lambda)\left|\frac{Y_{2,r}(\theta)}{Y_{0,r}(\theta)} + \frac{1}{2}\frac{\|Y_{1,r}(\theta)\|^2}{Y_{0,r}(\theta)^2} - \frac{Y_2\theta)}{Y_0(\theta)} - \frac{1}{2}\frac{\|Y_3(\theta)\|^2}{Y_2(\theta)^2}\right| + \left|\frac{\nabla_{X_\Lambda} g_\Lambda(X_\Lambda)Y_1(\theta)}{Y_0(\theta)} - \frac{\nabla_{X_\Lambda} g_\Lambda(X_\Lambda)Y_{1,r}(\theta)}{Y_{0,r}(\theta)}\right|$$

$$\leq \frac{|Y_{2,r}(\theta) - Y_2(\theta)|}{Y_0(\theta)} + \frac{|Y_{0,r}(\theta) - Y_0(\theta)| \, |Y_2(\theta)|}{Y_0(\theta)Y_{0,r}(\theta)}$$

$$+ \frac{1}{2}\frac{\left|\|Y_{1,r}(\theta)\|^2 - \|Y_1(\theta)\|^2\right|}{Y_0(\theta)^2} + \frac{1}{2}\frac{|Y_{0,r}(\theta)^2 - Y_0(\theta)^2| \, \|Y_{1,r}(\theta)\|^2}{Y_0(\theta)^2 Y_{0,r}(\theta)^2}.$$

$$+ \frac{\left|\nabla_{X_\Lambda} g_\Lambda(X_\Lambda)^\top(Y_{1,r}(\theta) - Y_1(\theta))\right|}{Y_0(\theta)} + \frac{|Y_{0,r}(\theta) - Y_0(\theta)| \left|\nabla_{X_\Lambda} g_\Lambda(X_\Lambda)^\top Y_1(\theta)\right|}{Y_0(\theta)Y_{0,r}(\theta)}$$

$$\leq \frac{|Y_{2,r}(\theta) - Y_2(\theta)|}{a_0} + \frac{|Y_{0,r}(\theta) - Y_0(\theta)| \, b_2}{a_0^2}$$

$$+ \frac{1}{2}\frac{\left|\|Y_{1,r}(\theta)\|^2 - \|Y_1(\theta)\|^2\right|}{a_0^2} + \frac{1}{2}\frac{|Y_{0,r}(\theta)^2 - Y_0(\theta)^2| \, b_1^2}{a_0^4}$$

$$+ \frac{\|Y_{1,r}(\theta) - Y_1(\theta)\| \, c_1}{a_0} + \frac{|Y_{0,r}(\theta) - Y_0(\theta)| \, c_1 b_1}{a_0^2}$$

$$\leq \frac{|Y_{2,r}(\theta) - Y_2(\theta)|}{a_0} + \frac{|Y_{0,r}(\theta) - Y_0(\theta)| \, b_2}{a_0^2}$$

$$+ \frac{1}{2}\frac{\|Y_{1,r}(\theta) - Y_1(\theta)\|^2 \, a_1}{a_0^2} + \frac{1}{2}\frac{|Y_{0,r}(\theta) - Y_0(\theta)| \, b_0 b_1^2}{a_0^4}$$

$$+ \frac{\|Y_{1,r}(\theta) - Y_1(\theta)\| \, c_1}{a_0} + \frac{|Y_{0,r}(\theta) - Y_0(\theta)| \, c_1 b_1}{a_0^2}$$

Now if we define the events

$$E_0(\theta) := \left\{|Y_{0,r}(\theta) - Y_0(\theta)| > \varepsilon\left(\frac{a_0^2}{6b_2} \bigwedge \frac{a_0^4}{3b_0 b_1^2} \bigwedge \frac{a_0^2}{6c_1 b_1}\right)\right\}$$

$$E_1(\theta) := \left\{\|Y_{1,r}(\theta) - Y_1(\theta)\| > \varepsilon\left(\frac{a_0^2}{3b_1} \bigwedge \frac{a_0}{6c_1}\right)\right\}$$

$$E_2(\theta) := \left\{|Y_{2,r}(\theta) - Y_2(\theta)| > \varepsilon\frac{a_0}{6}\right\}$$

then we have that

$$\{|Z_r(\theta) - Z(\theta)| < \varepsilon\} \subseteq \bigcup_{l=1}^{3} E_k(\theta)$$

Using standard Hoeffding bounds for $E_0(\theta), E_2(\theta)$ and union bounds in conjunction with Hoeffding bounds for $E_1(\theta)$ we get that

$$\mathbb{P}(E_0(\theta)) \geq 1 - 2\exp\left\{-\varepsilon^2 r\left(\frac{a_0^4}{36b_0^2 b_2^2} \bigwedge \frac{a_0^4}{9b_0^4 b_1^4} \bigwedge \frac{a_0^4}{36c_1^2 b_1^2 b_0^2}\right)\right\}$$

$$\mathbb{P}(E_1(\theta)) \geq 1 - 2d\exp\left\{-\varepsilon^2 r\left(\frac{a_0^4}{9b_1^4} \bigwedge \frac{a_0^2}{36c_1^2 b_1^2}\right)\right\}$$

$$\mathbb{P}(E_2(\theta)) \geq 1 - 2\exp\left\{-\varepsilon^2 r\frac{a_0^2}{36b_2^2}\right\}.$$

Hence

$$\mathbb{P}(|Z_r(\theta) - Z(\theta)| > \varepsilon) \leq (4 + 2d)\exp\left\{-\frac{\varepsilon^2 m}{\alpha^2}\right\} \quad \text{with}$$

$$\alpha := \max\left\{\frac{6b_0 b_2}{a_0^2}, \frac{3b_0^2 b_1^2}{a_0^4}, \frac{6c_1 b_1 b_0}{a_0^2}, \frac{3b_1^2}{a_0^2}, \frac{6c_1 b_1}{a_0}\frac{6b_2}{a_0}\right\}.$$

$\square$

*proof of Theorem A.15.* Our strategy will be as follows:

- Use our bound on $|Z_r(\theta) - Z(\theta)|$ from Proposition C.5 to bound $|L_{\mathrm{TM};n,r}(\theta) - L_n(\theta)|$.

- Use a covering number argument alongside Lemma C.4 to bound $\sup_\theta |L_{\mathrm{TM};n,r}(\theta) - L_n(\theta)|$.

- Use a similar approach to bound $\sup_\theta |L_n(\theta) - L_{\mathrm{TM}}(\theta)|$.

- Combine these to bound $|L_{\mathrm{TM};n,r}(\theta) - L_{\mathrm{TM}}(\theta)|$

For the first step we have that

$$\mathbb{P}(|L_{\mathrm{TM};n,r}(\theta) - L_{\mathrm{TM};n}(\theta)| > \varepsilon) \leq \mathbb{P}\left(\bigcup_{i=1}^{n} |L_{1,r}^{(i)}(\theta) - L_1^{(i)}(\theta)| > \varepsilon\right)$$

$$\leq \sum_{i=1}^{n} \mathbb{P}(|L_{1,r}(\theta) - L_1(\theta)| > \varepsilon)$$

$$= \sum_{i=1}^{n} \mathbb{P}(|Z_r(\theta) - Z(\theta)| > \varepsilon)$$

$$= n(4 + 2d)\exp\left\{-\frac{\varepsilon^2 m}{\alpha^2}\right\}$$

where

$$L_{1,r}^{(i)}(\theta) = g_{\Lambda_i}(X_{\Lambda_i}^{(i)}) \left( \frac{\frac{1}{r}\sum_{k=1}^{r} f_{2,\Lambda}(X_\Lambda^{(i)}, X_{-\Lambda}^{'(k)}, \theta)}{\frac{1}{r}\sum_{k=1}^{r} f_{0,\Lambda}(X_\Lambda^{(i)}, X_{-\Lambda}^{'(k)}, \theta)} + \frac{1}{2}\frac{\left\| \frac{1}{r}\sum_{k=1}^{r} f_{1,\Lambda}(X_{\Lambda_i}, X_{-\Lambda}^{'(k)}, \theta) \right\|^2}{\left( \frac{1}{r}\sum_{k=1}^{r} f_{0,\Lambda}(X_\Lambda^{(i)}, X_{-\Lambda}^{'(k)}, \theta) \right)^2} \right)$$

$$+ \frac{\frac{1}{n}\sum_{i=1}^{n} g_{\Lambda_i}(X_\Lambda^{(i)})^\top f_1(X_\Lambda^{(i)}, X_{-\Lambda}^{'(k)})}{\frac{1}{n}\sum_{i=1}^{n} f_0(X_\Lambda^{(i)}, X_{-\Lambda}^{'(k)})}$$

$$L_1^{(i)}(\theta) = g_{\Lambda_i}(X_{\Lambda_i}^{(i)}) \left( \frac{\mathbb{E}\left[ f_{2,\Lambda}(X_\Lambda^{(i)}, X_{-\Lambda}', \theta) \Big| X_\Lambda^{(i)} \right]}{\mathbb{E}\left[ f_{0,\Lambda}(X_\Lambda^{(i)}, X_{-\Lambda}', \theta) \Big| X_\Lambda^{(i)} \right]} + \frac{1}{2}\frac{\left\| \mathbb{E}\left[ f_{1,\Lambda}(X_\Lambda^{(i)}, X_{-\Lambda}', \theta) \Big| X_\Lambda^{(i)} \right] \right\|^2}{\mathbb{E}\left[ f_{0,\Lambda}(X_\Lambda^{(i)}, X_{-\Lambda}', \theta) \Big| X_\Lambda^{(i)} \right]^2} \right)$$

$$+ \frac{\mathbb{E}[g_{\Lambda_i}(X_\Lambda^{(i)})^\top f_1(X_\Lambda^{(i)}, X_{-\Lambda}^{'(k)})|X_\Lambda^{(i)}]}{\mathbb{E}[f_0(X_\Lambda^{(i)}, X_{-\Lambda}^{'(k)})|X_\Lambda^{(i)}]}.$$

We now try and derive a Lipschitz constant for both $L_{n,r}$ and $L_n$. Define $g : \mathcal{X} \to \mathbb{R}$ by

$$g(\boldsymbol{x}) = \frac{1}{a_0}M_2(\boldsymbol{x}) + \left( \frac{b_2}{a_0^2} + \frac{b_1^2}{2a_0^4} + \frac{c_1 b_1}{a_0^2} \right)M_0(\boldsymbol{x}) + \left( \frac{a_1}{2a_0^2} + \frac{c_1}{a_0} \right)M_1(\boldsymbol{x})$$

Then we have that For $L_{n,r}$ we have

$$|L_{\mathrm{TM};n,r}(\theta) - L_{n,r}(\theta')| \leq \rho(\theta, \theta')\underbrace{\frac{1}{nr}\sum_{i=1}^{n}\sum_{k=1}^{r} g(X_\Lambda^{(i)}, X_{-\Lambda}^{'(k)})}_{:=C_{n,r}}$$

similarly we have

$$|L_{\mathrm{TM};n}(\theta) - L_n(\theta')| \leq \rho(\theta, \theta')\underbrace{\frac{1}{n}\sum_{i=1}^{n} \mathbb{E}[g(X_\Lambda^{(i)}, X_{-\Lambda}')|X_\Lambda^{(i)}]}_{:=C_n}.$$

Using an identical argument to Proposition C.1 can get the following bound for any $\eta_1 > 0$:

$$\mathbb{P}(\sup_{\theta \in \Theta}|L_{\mathrm{TM}}(\theta) - L_{\mathrm{TM};n,r}(\theta)|> \varepsilon_1 + \eta_1(C_{n,r} + C_n)) > N(\eta_1, \Theta)n(4 + 2d)\exp\left\{ -\frac{\varepsilon^2 m}{\alpha^2} \right\}$$

Hence we now need to bound $C_n$ and $C_{n,r}$. We know that both of these terms converge to $C := \mathbb{E}[g(X_\Lambda, X_\Lambda')]$. To obtain rates on this convergence we make sub-Gaussian statements about $g$. Specifically for $\boldsymbol{x}_{-\lambda} \in \mathcal{X}$ we have that $g(X_\lambda, \boldsymbol{x}_{-\lambda})$ is sub-Gaussian with parameter

$$\sigma_\lambda := \frac{1}{a_0}\sigma_{2,\lambda} + \left( \frac{b_2}{a_0^2} + \frac{b_1^2}{2a_0^4} + \frac{c_1 b_1}{a_0^2} \right)\sigma_{0,\lambda} + \left( \frac{a_1}{2a_0^2} + \frac{c_1}{a_0} \right)\sigma_{1,\lambda}$$

We can therefore immediately bound $C_n - C$ using Lemma C.3 and Hoeffding bounds to get

$$\mathbb{P}(C_n - C > \varepsilon) = \mathbb{E}[\mathbb{P}(C_n - C > \varepsilon|\Lambda)]$$

$$\leq \mathbb{E}\left[ \exp\left\{ -\frac{\varepsilon^2 n}{8\sigma_\Lambda^2} \right\} \right]$$

$$\leq \exp\left\{ -\frac{\varepsilon^2 n}{8\sigma^2} \right\}.$$

where $\sigma := \max_{\Lambda \in \text{supp}(\Lambda)} \sigma_\Lambda$. To bound $C_{n,r} - C$ we can use Lemma C.4 to get

$$\mathbb{E}\left[\mathbb{P}(C_{n,r} - C > \varepsilon)|\Lambda\right] = \mathbb{E}\left[\mathbb{P}\left(\frac{1}{nr}\sum_{i=1}^{n}\sum_{k=1}^{r} g(X^{(i)}, Y^{(k)}) - \mathbb{E}[g(X,Y)] > \varepsilon \middle| \Lambda\right)\right]$$

$$\leq \mathbb{E}\left[n\exp\left\{-\frac{\varepsilon^2 m}{8\sigma'^2_{-\Lambda}}\right\} + \exp\left\{-\frac{\varepsilon^2 n}{8\sigma^2_{\Lambda}}\right\}\right]$$

$$= n\exp\left\{-\frac{\varepsilon^2 m}{8\sigma'^2}\right\} + \exp\left\{-\frac{\varepsilon^2 n}{8\sigma^2}\right\}$$

with $\sigma', \sigma'_{-\lambda}$ define identically to $\sigma, \sigma_\lambda$ with $\sigma_{l,\lambda}$ replaced with $\sigma'_{l,-\lambda}$ and $\sigma_\lambda$ replaced with $\sigma'_{-\lambda}$. As a result we get that

$$\mathbb{P}(\sup_{\theta \in \Theta}|L_{\text{TM};n,r}(\theta) - L_n(\theta)| > \varepsilon_1 + 2\eta_1(C + \varepsilon))$$

$$< N(\eta_1, \Theta)n(4 + 2d)\exp\left\{-\frac{\varepsilon^2 m}{\alpha^2}\right\} + n\exp\left\{-\frac{\varepsilon^2 m}{8\sigma'^2}\right\} + 2\exp\left\{-\frac{\varepsilon^2 n}{8\sigma^2}\right\}$$

Now we have the bound on $\sup_\theta |L_{\text{TM};n,r}(\theta) - L_{\text{TM}}(\theta)|$. We aim to bound $\sup_\theta |L_{\text{TM};n}(\theta) - L_{\text{TM}}(\theta)|$. To that end we have

$$|L_1(\theta)| < \frac{b_2}{a_0} + \frac{1}{2}\frac{b_1^2}{a_0^2} + \frac{c_1 b_1}{a_0} =: \alpha'$$

a.s. and so we can use Hoeffding bounds to get

$$\mathbb{P}(|L_{\text{TM};n}(\theta) - L_{\text{TM}}(\theta)| > \varepsilon) \leq \exp\left\{-\frac{\varepsilon^2 n}{2\alpha'}\right\}.$$

Again using an argument similar to Proposition C.1 we have for any $\eta_2 > 0$

$$\mathbb{P}(\sup_{\theta \in \Theta}|L_n(\theta) - L_{\text{TM}}(\theta)| > \varepsilon_2 + \eta_2(2C + \varepsilon_3) \leq N(\eta_2, \Theta)\exp\left\{-\frac{\varepsilon_2^2 n}{2\alpha'^2}\right\} + \mathbb{P}(C_n - C > \varepsilon_3)$$

Combining these two results gives that for any $\varepsilon > 0$

$$\mathbb{P}(\sup_{\theta \in \Theta}|L_{\text{TM}}(\theta) - L_{\text{TM};n,r}(\theta)| \geq \varepsilon_1 + \varepsilon_2 + \eta_1(2C + 2\varepsilon_3) + \eta_2(2C + \varepsilon_3))$$

$$\leq \mathbb{P}(\sup_{\theta \in \Theta}|L_{\text{TM}}(\theta) - L_n(\theta)| \geq \varepsilon_1 + 2\eta_2(C + \varepsilon_3) \cup \sup_{\theta \in \Theta}|L_{\text{TM};n}(\theta) - L_{\text{TM};n,r}(\theta)| \geq \varepsilon_2 + \eta_2(2C + \varepsilon_3))$$

$$\leq \mathbb{P}(\sup_{\theta \in \Theta}|L_{\text{TM}}(\theta) - L_n(\theta)|\varepsilon_1 + 2\eta_1(C + \varepsilon_3)) + \mathbb{P}(\sup_{\theta \in \Theta}|L_{\text{TM};n}(\theta) - L_{\text{TM};n,r}(\theta)| \geq \varepsilon_2 + \eta_2(2C + \varepsilon_3))$$

$$\leq N(\eta_1, \Theta)n(4 + 2d)\exp\left\{-\varepsilon_1^2 m\alpha^2\right\} + N(\eta_2, \Theta)\left(\exp\left\{-\frac{\varepsilon_2^2 n}{2\alpha'}\right\}\right) + \mathbb{P}(C_n - C > \varepsilon_3) + \mathbb{P}(C_{n,r} - C > \varepsilon_3)$$

$$\leq N(\eta_1, \Theta)n(4 + 2d)\exp\left\{-\varepsilon_1^2 m\alpha^2\right\} + N(\eta_2 \Theta)\left(\exp\left\{-\frac{\varepsilon_2^2 n}{2\alpha'}\right\}\right) + n\exp\left\{-\frac{\varepsilon_3^2 m}{8\sigma'^2}\right\} + 2\exp\left\{-\frac{\varepsilon_3^2 n}{8\sigma^2}\right\} \quad (22)$$

Take $\eta_1 = 1/r$ and $\eta_2 = 1/n$ so that for sufficiently large $n, r$ $N(\eta_1, \Theta) \leq \exp\{p\log((3/2)\text{diam}(\Theta)r)\}$ and $N(\eta_2, \Theta) = \exp\{p\log((3/2)\text{diam}(\Theta)n)\}$. Plugging this into (22) gives

$$\mathbb{P}\left(\sup_{\theta \in \Theta}|L_{\text{TM}}(\theta) - L_{\text{TM};n,r}(\theta)| \geq \varepsilon_1 + \varepsilon_2 + 1/r(2C + 2\varepsilon_3) + 1/n(2C + \varepsilon_3)\right)$$

$$\leq n(4 + 2d)\exp\left\{p\log\frac{3}{2}\text{diam}(\Theta)r - \varepsilon_1^2 n\alpha\right\} + \exp\left\{p\log\frac{3}{2}\text{diam}(\Theta)n - \frac{\varepsilon_2^2 n}{2\alpha'}\right\} + (2 + n)\exp\left\{-\frac{\varepsilon_3^2 n}{8(\sigma \vee \sigma')^2}\right\}$$

Now if we take each of these terms to be equal to $\delta/3$ gives for sufficiently large $n, r$

$$\mathbb{P}\left(\sup_{\theta\in\Theta}|L_{\text{TM}}(\theta) - L_{\text{TM};n,r}(\theta)| \geq \beta_1 \sqrt{\frac{p\log(dnr\,\text{diam}(\Theta)/\delta)}{r}} + \beta_2\sqrt{\frac{p\log(n\,\text{diam}(\Theta)/\delta)}{n}}\right.$$
$$\left. + \beta_3\left(\frac{n+r}{nr}\right)\left(C + \sqrt{\frac{\log(n/\delta)}{n}}\right)\right) < \delta.$$

where $\beta_1 = \frac{27}{\alpha}, \beta_2 = 9\alpha', \beta_3 = 10(\sigma \vee \sigma')$. As there exists $\theta^* \in \Theta$ a minimiser of $L_{\text{TM}}(\theta)$ we now have that

$$|L(\theta_n) - L(\theta^*)| \leq |L(\theta_n) - L_n(\theta_n)| + |L_n(\theta^*) - L(\theta^*)|.$$

Finally we know that $J_{\text{TM}}(\theta) = L_{\text{TM}}(\theta) + C$ and under a correctly specified model $L(\theta^*) = 0$. Therefore we have that

$$\mathbb{P}(|J(\theta_{n,r})| > 2\varepsilon) \leq \mathbb{P}(\sup_{\theta\in\Theta}|L_{\text{TM};n}(\theta) - L_{\text{TM}}(\theta)| > \varepsilon)$$

and so we have our result simply replacing $\beta_k$ with $2\beta_k$. $\qquad\square$

### C.3. Gradient First Proofs

*Proof of Lemma 4.8.* First we have that for any $\lambda$, $\boldsymbol{x}_\lambda$,

$$s_{\theta;\lambda}(\boldsymbol{x}_\lambda) = \nabla_{\boldsymbol{x}_\lambda}\log\int p_\theta(\boldsymbol{x})\mathrm{d}\boldsymbol{x}_{-\lambda}$$
$$= \frac{\nabla_{\boldsymbol{x}_\lambda}\int p_\theta(\boldsymbol{x})\mathrm{d}\boldsymbol{x}_{-\lambda}}{\int p_\theta(\boldsymbol{x})\mathrm{d}\boldsymbol{x}_{-\lambda}}$$
$$= \frac{\int \nabla_{\boldsymbol{x}_\lambda}p_\theta(\boldsymbol{x})\mathrm{d}\boldsymbol{x}_{-\lambda}}{p(\boldsymbol{x}_{-\lambda})}$$
$$= \int \frac{p_\theta(\boldsymbol{x})}{p_\theta(\boldsymbol{x}_{-\lambda})}\nabla_{\boldsymbol{x}_\lambda}\log p_\theta(\boldsymbol{x})\mathrm{d}\boldsymbol{x}_{-\lambda}$$
$$= \mathbb{E}_{\boldsymbol{x}_{-\lambda}|\boldsymbol{x}_\lambda;\theta}[s_\theta(\boldsymbol{x})_\lambda].$$

Taking expectations on both side w.r.t. $(\Lambda, X_\Lambda)$ proves equation (8).

For (9) we have again for any $\lambda$, $\boldsymbol{x}_\lambda$,

$$\nabla\mathbb{E}'_{X'_{-\lambda}\sim p_\theta}[g_\theta(\boldsymbol{x}_\lambda, X'_{-\lambda})] = \int \nabla(p_\theta(\boldsymbol{x}_{-\lambda}|\boldsymbol{x}_\lambda)g_\theta(\boldsymbol{x}))\mathrm{d}\boldsymbol{x}_{-\lambda}$$
$$= \int p_\theta(\boldsymbol{x}_{-\lambda}|\boldsymbol{x}_\lambda)\nabla g_\theta(\boldsymbol{x})\mathrm{d}\boldsymbol{x}_{-\lambda} + \int \nabla p_\theta(\boldsymbol{x}_{-\lambda}|\boldsymbol{x}_\lambda)g_\theta(\boldsymbol{x})\mathrm{d}\boldsymbol{x}_{-\lambda}$$
$$= \int p_\theta(\boldsymbol{x}_{-\lambda}|\boldsymbol{x}_\lambda)\nabla g_\theta(\boldsymbol{x})\mathrm{d}\boldsymbol{x}_{-\lambda} + \int p_\theta(\boldsymbol{x}_{-\lambda}|\boldsymbol{x}_\lambda)\nabla\log p_\theta(\boldsymbol{x}_{-\lambda}|\boldsymbol{x}_\lambda)g_\theta(\boldsymbol{x})\mathrm{d}\boldsymbol{x}_{-\lambda}$$
$$= \mathbb{E}'[\nabla g_\theta(\boldsymbol{x}_\lambda, X'_{-\lambda})] + \mathbb{E}'[\nabla\log p_\theta(\boldsymbol{x}_\lambda, X'_{-\lambda})g_\theta(\boldsymbol{x}_\lambda, X'_{-\lambda})]$$
$$\quad - \mathbb{E}'[\nabla\log p_\theta(\boldsymbol{x}_\lambda)g_\theta(\boldsymbol{x}_\lambda, X'_{-\lambda})]$$
$$= \mathbb{E}'[\nabla g_\theta(\boldsymbol{x}_\lambda, X'_{-\lambda})] + \mathbb{E}'[\nabla\log p_\theta(\boldsymbol{x}_\lambda, X'_{-\lambda})g_\theta(\boldsymbol{x}_\lambda, X'_{-\lambda})]$$
$$\quad - \boldsymbol{s}_{\theta;\lambda}(\boldsymbol{x}_\lambda)\mathbb{E}'[g_\theta(\boldsymbol{x}_\lambda, X'_{-\lambda})]$$
$$= \mathbb{E}'[\nabla g_\theta(\boldsymbol{x}_\lambda, X'_{-\lambda})] + \mathbb{E}'[\nabla\log p_\theta(\boldsymbol{x}_\lambda, X'_{-\lambda})g_\theta(\boldsymbol{x}_\lambda, X'_{-\lambda})]$$
$$\quad - \mathbb{E}'[\nabla\log p_\theta(x_\lambda, X'_{-\lambda})]\mathbb{E}'[g_\theta(\boldsymbol{x}_\lambda, X'_{-\lambda})]$$
$$= \mathbb{E}'[\nabla g_\theta(\boldsymbol{x}_\lambda, X'_{-\lambda})] + \text{Cov}_{p_\theta}(s_\theta(\boldsymbol{x}_\lambda, X'_{-\lambda}), g_\theta(\boldsymbol{x}_\lambda, X'_{-\lambda})).$$

Where again here $\nabla$ represents the gradient w.r.t. either $\boldsymbol{x}_\lambda$ or $\theta$. We can again take expectations on both sides w.r.t. $\Lambda, X_\Lambda$ to get our result. $\qquad\square$

*Proof of Corollary 4.9.* First define

$$L_{\mathrm{M}}(\theta; \boldsymbol{x}_\lambda) := \sum_{j \in \lambda} 2\partial_j \boldsymbol{s}_{\lambda;\theta}(\boldsymbol{x}_\lambda)_j + \boldsymbol{s}_{\lambda;\theta}(\boldsymbol{x}_\lambda)_j^2$$

so that $\mathbb{E}[L_{\mathrm{M}}(\theta; X_\Lambda)] = L_{\mathrm{M}}(\theta)$. Then using our two score identities, (8) & (9) we can re-write $L(\theta; \boldsymbol{x})$ as

$$\begin{aligned}
L_{\mathrm{M}}(\theta; \boldsymbol{x}) &= \sum_{j \in \lambda} \mathbb{E}'[\boldsymbol{s}_\theta(\boldsymbol{x}_\lambda, X'_{-\lambda})_j]^2 + 2\partial_j \mathbb{E}'[\boldsymbol{s}_\theta(\boldsymbol{x}_\lambda, X'_{-\lambda})_j] \quad \text{by (8)}\\
&= \sum_{j \in \lambda} \mathbb{E}'[\boldsymbol{s}_\theta(\boldsymbol{x}_\lambda, X'_{-\lambda})_j]^2 + \mathbb{E}'[\partial_j \boldsymbol{s}_\theta(\boldsymbol{x}_\lambda, X'_{-\lambda})_j] + \mathrm{Var}(s_\theta(\boldsymbol{x}_\lambda, X'_{-\lambda})_j) \quad \text{by (9)}\\
&= \sum_{j \in \lambda} -\mathbb{E}'[s_\theta(\boldsymbol{x}_\lambda, X'_{-\lambda})_j]^2 + 2\mathbb{E}'[s_\theta(\boldsymbol{x}_\lambda, X'_{-\lambda})_j^2] + 2\mathbb{E}'[\partial_i s_\theta(\boldsymbol{x}_\lambda, x_{-\lambda})_j]
\end{aligned}$$

Where all expectations are w.r.t. $X'_{-\lambda}|X_\lambda = \boldsymbol{x}_\lambda; \theta$ Now using the fact that

$$\nabla_\theta \left( \mathbb{E}'[\boldsymbol{s}_\theta(\boldsymbol{x}_\lambda, X'_{-\lambda})_j] \right)^2 = 2E[\boldsymbol{s}_\theta(\boldsymbol{x}_\lambda, X'_{-\lambda})_j]\nabla_\theta \mathbb{E}'[\boldsymbol{s}_\theta(\boldsymbol{x}_\lambda, X'_{-\lambda})_j]$$

and using (9) again on each term of the above gives followed by taking expectations w.r.t. $\Lambda, X_\Lambda$ gives the result. $\qquad\square$

### C.3.1. TRUNCATED SCORE MATCHING
*Proof of Corollary 4.11.* Now using Lemma 4.8, we know that this can be re-written as

$$\begin{aligned}
&\sum_{j \in \lambda} g_\lambda(\boldsymbol{x}_\lambda)_j (\mathbb{E}'[\boldsymbol{s}_\theta(\boldsymbol{x}_\lambda, X'_{-\lambda})_j]^2 + 2\mathrm{Var}'(\boldsymbol{s}_\theta(\boldsymbol{x}_\lambda, X'_{-\lambda})_j) + 2\mathbb{E}'[\partial_j \boldsymbol{s}_\theta(\boldsymbol{x}_\lambda, X'_{-\lambda})_j]) + 2\partial_j g(\boldsymbol{x}_\lambda)_j \mathbb{E}'[\boldsymbol{s}_\theta(\boldsymbol{x}_\lambda, X'_{-\lambda})]\\
&= \sum_{j \in \lambda} g_\lambda(\boldsymbol{x}_\lambda)_j (-\mathbb{E}'[\boldsymbol{s}_\theta(\boldsymbol{x}_\lambda, X'_{-\lambda})_j]^2 + 2\mathbb{E}'[\boldsymbol{s}_\theta(\boldsymbol{x}_\lambda, X'_{-\lambda})_j^2] + 2\mathbb{E}'[\partial_j \boldsymbol{s}_\theta(\boldsymbol{x}_\lambda, X'_{-\lambda})_j]) + 2\partial_j g(\boldsymbol{x}_\lambda)_j \mathbb{E}'[\boldsymbol{s}_\theta(\boldsymbol{x}_\lambda, X'_{-\lambda})]
\end{aligned}$$

Now taking gradient w.r.t. $\theta$ we get

$$\begin{aligned}
\nabla_\theta L(\theta; \boldsymbol{x}_\lambda) = \sum_{j \in \lambda} g_\lambda(\boldsymbol{x}_\lambda)_j \Big\{ &-2\mathbb{E}'[\boldsymbol{s}_\theta(\boldsymbol{x})_j](\mathbb{E}'[\nabla_\theta \boldsymbol{s}_\theta(\boldsymbol{x})_j] + \mathrm{Cov}'(\nabla_\theta \log q_\theta(\boldsymbol{x}), \boldsymbol{s}_\theta(\boldsymbol{x})_j))\\
&+ 2(\mathbb{E}'[\nabla_\theta \boldsymbol{s}_\theta(\boldsymbol{x})_j^2] + \mathrm{Cov}'(\nabla_\theta \log q_\theta(\boldsymbol{x}), \boldsymbol{s}_\theta(\boldsymbol{x})_j^2))\\
&+ 2(\mathbb{E}'[\nabla_\theta \partial_j \boldsymbol{s}_\theta(\boldsymbol{x})_j] + \mathrm{Cov}'(\nabla_\theta \log q_\theta(\boldsymbol{x}), \partial_j \boldsymbol{s}_\theta(\boldsymbol{x})_j)) \Big\}\\
&+ 2\partial_j g(\boldsymbol{x}_\lambda)_j \Big\{ \mathbb{E}'[\nabla_\theta \boldsymbol{s}_\theta(x)_j] + \mathrm{Cov}'(\nabla_\theta \log q_\theta(\boldsymbol{x}), \boldsymbol{s}_\theta(\boldsymbol{x})_j) \Big\}.
\end{aligned}$$

$\qquad\square$

### C.4. IW and Gradient First Relationships
*Proof of Lemma A.18.* We first have that,

$$\begin{aligned}
\hat{\boldsymbol{s}}_{\theta;\lambda}(\boldsymbol{x}_\lambda) &= \nabla_{\boldsymbol{x}_\lambda} \log \frac{1}{r} \sum_{i=1}^r w_i\\
&= \frac{\frac{1}{r}\sum_{i=1}^r \nabla_{\boldsymbol{x}_\lambda} w_i}{\frac{1}{r}\sum_{i=1}^r w_i}\\
&= \frac{1}{r} \sum_{i=1}^r \bar{w}_i \nabla_{\boldsymbol{x}_\lambda} \log q_\theta(\boldsymbol{x}_\lambda, X'^{(i)}_{-\lambda})\\
&= \hat{\mathbb{E}}_{iw}[\nabla_{\boldsymbol{x}_\lambda} log q_\theta(\boldsymbol{x}_\lambda, X'^{(i)}_{-\lambda})] = \hat{\mathbb{E}}_{iw}[\boldsymbol{s}_\theta(\boldsymbol{x}_\lambda, X'^{(i)}_{-\lambda})_\lambda].
\end{aligned}$$

Where the penultimate result uses the fact that $\nabla_{\boldsymbol{x}_\lambda} w_i = w_i \log q_\theta(\boldsymbol{x}_\lambda, X'^{(i)}_{-\lambda})$

Additionally,

$$\nabla_{\boldsymbol{x}_\lambda} hat E_{iw}[g_\theta(\boldsymbol{x}_\lambda, X'_{-\lambda})] = \frac{\nabla_{\boldsymbol{x}_\lambda} \sum_{k=1}^{n} w_k g_\theta(\boldsymbol{x}_\lambda, X'^{(k)}_{-\lambda})}{\sum_{k=1}^{r} w_k} - \frac{\left(\nabla_{\boldsymbol{x}_\lambda} \sum_{k=1}^{r} w_k\right) \left(\sum_{k=1}^{n} w_k g_\theta(\boldsymbol{x}_\lambda, X'^{(k)}_{-\lambda})\right)}{\left(\sum_{k=1}^{r} w_k\right)^2}$$

For the second term we can again use $\nabla_{\boldsymbol{x}_\lambda} w_k = w_k \nabla_{\boldsymbol{x}_\lambda} log p_\theta(\boldsymbol{x}_\lambda, X'^{(k)}_{-\lambda})$ to write this as

$$\hat{\mathbb{E}}_{iw}[\nabla_{\boldsymbol{x}_\lambda} log p_\theta(\boldsymbol{x}_\lambda, X'_{-\lambda})]\mathbb{E}_{iw}[g_\theta(\boldsymbol{x}_\lambda, X'_{-\lambda})]$$

The first term we can write as

$$\frac{\sum_{i=1}^{n} w_k \log p_\theta(\boldsymbol{x}_\lambda, X'_{-\lambda})g_\theta(\boldsymbol{x}_\lambda, X'_{-\lambda})}{\sum_{i=1}^{r} w_k} + \frac{\sum_{i=1}^{n} w_k \nabla_{\boldsymbol{x}_\lambda} g_\theta(\boldsymbol{x}_\lambda, X'_{-\lambda})}{\sum_{i=1}^{r} w_k}$$
$$= \hat{\mathbb{E}}_{iw}[\log p_\theta(\boldsymbol{x}_\lambda, X'_{-\lambda})g_\theta(\boldsymbol{x}_\lambda, X'_{-\lambda})] + \hat{\mathbb{E}}_{iw}[\nabla_{\boldsymbol{x}_\lambda} g_\theta(\boldsymbol{x}_\lambda, X'_{-\lambda})]$$

Combining these 2 gives our desired results. $\qquad\square$

*Proof of Corollary A.19.* The exact same arguments give us the second result but with the importance weighting identities (19) and (20) replacing (8) & (9). $\qquad\square$

# D. Additional Score Matching Details & Extensions

In this section we give some additional details on score matching and introduce some pre-existing score matching extensions and methods. For some of these approach we can adapt our method to work with them while others act as comparison points for our approach.

## D.1. Classical Score Matching

The assumptions for the classical score matching result presented in (1)

**Assumption D.1.** For

(a) The pdf $p(\boldsymbol{x})$ is differentiable w.r.t. $\boldsymbol{x}$;

(b) Our score estimating function $\boldsymbol{s}_\theta$ is differentiable w.r.t. $\boldsymbol{x}$;

(c) $\mathbb{E}[\|\boldsymbol{s}_\theta(X)\|^2], \mathbb{E}[\|\boldsymbol{s}(X)\|^2] < \infty$;

(d) $p(\boldsymbol{x})\boldsymbol{s}_\theta(\boldsymbol{x}) \longrightarrow \boldsymbol{0}$ whenever $\|\boldsymbol{x}\| \longrightarrow \infty$.

## D.2. Zeroed Score Matching and MissDiff

In this section we take $\boldsymbol{x}_\lambda^0$ to be the vector in $\mathbb{R}^d$ with $(\boldsymbol{x}_\lambda^0)_i = x_i$ if $i \in \lambda$ and 0 otherwise and then take $X_\lambda^0$ to be the RV equivalent.

MissDiff is a generative modelling technique that aims to learn generative models from corrupted tabular data using diffusion models with denoising score matching. Throughout their method $\boldsymbol{s}_\theta$ is assumed to be a neural network. The core idea is to replace the standard score matching objective with

$$\mathbb{E}_{t, X_\Lambda(0), X_\Lambda(t)} \left[\mu(t)\|\boldsymbol{s}(X_\Lambda^0(t), t)_\Lambda - \nabla_{X_\Lambda(t)} \log p(X_\Lambda(t)|X_\Lambda(0))\right].$$

Essentially, missing values are zero imputed and then the score is tested only on output dimensions whose input dimensions are non-missing. The idea of this approach is that it trains $\boldsymbol{s}_\theta(^0_\lambda(t), t)_\lambda$ to approximate the true marginal score $\boldsymbol{s}_{\lambda;\theta}(\boldsymbol{x}_\lambda(t), t)$ which in turn encourages $\boldsymbol{s}_\theta(\boldsymbol{x}(t), t)$ to approximate the full score $\boldsymbol{s}_\theta(x(t), t)$.

For comparison purposes, we adapt this in two ways to create Zeroed Score Matching. Firstly we change the objective to standard score matching and secondly we no longer require $\boldsymbol{s}$ to be a neural network. Thus our adapted version of the MissDiff objective is

$$\hat{L}_{\text{Zeroed}}(\theta) = \mathbb{E}\left[\nabla_{(X_\Lambda^0)_\Lambda} \cdot \boldsymbol{s}_\theta(X_\Lambda^0)_\Lambda + \|\boldsymbol{s}_\theta(X_\Lambda^0)_\Lambda\|\right]. \tag{23}$$

The key issue with this approach is that with $s_\theta$ no longer necessarily a neural network it is not reasonable (or indeed even possible) for $s_\theta$ to model both the joint and marginal scores. In other words we cannot expect both $s_\theta(x_\lambda^0) = s_\lambda(x_\lambda)$ and $s_\theta(x) = s(x)$ making it a naive marginalisation method for the score.

We can give a brief example with a multidimensional normal distribution. Supposed that $X \sim N(\mathbf{0}, \Sigma)$ with $P := \Sigma^{-1}$. Then the score is $s(x) = -Px$ which we would model with $s_\theta(x) = -P_\theta x$. Under the Zeroed scheme, we would take the marginal score to be

$$
\begin{aligned}
s_{\lambda,\theta}(x) &= \left(-Px_\lambda^0\right)_\lambda \\
&= -P_{\lambda,\lambda;\theta} x_\lambda.
\end{aligned}
$$

However we know that if $X \sim N(\mathbf{0}, \Sigma_\theta)$ with $\Sigma_\theta = P_\theta^{-1}$ then $X_\lambda \sim N(\mathbf{0}, \Sigma_{\lambda,\lambda;\theta})$ and so we should take

$$
s_{\lambda,\theta}(x) = -(\Sigma_{\lambda,\lambda;\theta})^{-1} x_\lambda = -((P_\theta^{-1})_{\lambda,\lambda})^{-1} x_\lambda
$$

and crucially, $P_{\lambda,\lambda} \neq ((P^{-1})_{\lambda,\lambda})^{-1}$ unless $P = I$.

We illustrate this for our simulated experimental settings in Appendices B.1.2 & B.2.1.

We also explore the implications of this for the marginal Fisher divergence for the normal in Appendix A.4.

### D.3. Sliced Score Matching

One issue with score matching is that $\nabla_x cdot s_\theta(x)$ is computationally expensive to compute for large $d$. A solution to this was proposed in Song et al. (2020) where, instead of testing the full score 1-dimensional projections/slices are tested instead. This is done by introducing another RV $V$ on $(\mathcal{X}, \mathcal{B}_\mathcal{X})$ satisfying $\mathbb{E}[V] = 0$ and $\mathbb{E}[VV^\top]$ positive semi-definite.

The original objective is then

$$
F_S(\theta) := \mathbb{E}\left[\left\{V^\top(s(X) - s_\theta(X))\right\}^2\right]
$$

Which then leads to the following equivalence

$$
\begin{aligned}
L_S(\theta) &:= \mathbb{E}[2\left\{\nabla_X(V^\top s_\theta(X))\right\}^\top V + (V^\top s_\theta(X))^2] \\
&= F_S - C.
\end{aligned}
$$

which is less computationally expensive w.r.t. $d$ and can be approximated with samples of $X$ and $V$. For the proof of this results and the precise conditions see Song et al. (2020).

### D.4. Denoised Score matching

Denoised score matching is another adaptation which removes the need to takes derivatives of the score all together (Vincent, 2011). As we will see later on it is also the method used most prominently in diffusion processes.

In denoising score matching we construct a collection of RVs $\{X(t)\}_{t=0}^T$ with $X(0) = X$ and $X(t)|X(0) \sim N(m(t)X(0), \sigma(t)I_p)$. We assume that the noise $\sigma(t)$ grows sufficiently so that $X(T)$ is approximately an Isotropic Gaussian. The aim is now to estimate $s(x, t) := \nabla_x \log p_t(x)$ where $p_t$ is the PDF of $X(t)$. The denoising score matching objective is then.

$$
\mathbb{E}[\nu(t)\|s_\theta(X(t), t) - \nabla_{X(t)}\nabla_{X(t)}\log p(X(t)|X(0))\|_2^2] = \mathbb{E}\left[\nu(t)\|s_\theta(X(t), t) + p(X(t)|X(0))\|^2\right]
$$

where here $t$ is treated as random and uniformly distributed on $\{1, \ldots, T\}$, $p(x(t)|x(0))$ is the transition kernel from $X(0)$ to $X(t)$ and $\lambda(t)$ is a self-specified weighting function over time. Due to our choice of noising process, $\nabla_{x(t)} \log p(x(t)|x(0)) = \frac{1}{\sigma(t)}x(t) - m(t)x(0)$.

*Remark* D.2. Convention is to up-weight larger values of $t$ as earlier parts of the reverse diffusion process (hence later parts of the original diffusion process) are seen as the most complex and where most of the data's structure is learned.

Our estimate is thus

$$\hat{\theta} = \underset{\theta \in \mathbb{R}^p}{\operatorname{argmin}} \frac{1}{n} \sum_{i \in n} \nu(t_i) \| s_\theta(X^{(i)}(t_i), t_i) + \nabla_{X^{(i)}} \log p(X^{(i)}(t) | X^{(i)})(0)) \|^2$$

where $t_1, \dots, t_n$ are sampled uniformly from $[0, T]$.

*Remark* D.3. Originally denoising score matching was proposed for a single fixed $t$ however for the purpose of generative modelling and annealed Langevin dynamics, multiple noise levels or even a continuous noising processes are used.

# E. Additional Details

## E.1. Objectives

### E.1.1. MARGINAL IW SCORE MATCHING OBJECTIVE

Let $\{X_{\Lambda_i}^{\prime(i,k)}\}_{k=1}^r$ be IID copies of $X_{\Lambda_i}'$ with known PDF $p'(\boldsymbol{x}_{\Lambda_i})$. Our full sample objective is given by

$$\hat{L}_{\mathrm{M};n,r}(\theta) := \frac{1}{n} \sum_{i=1}^n 2 \nabla_{X_{\Lambda_i}^{(i)}} \cdot \left( \frac{\frac{1}{r} \sum_{k=1}^r \frac{\nabla_{X_{\Lambda_i}^{(i)}} q_\theta(X_{\Lambda_i}^{(i)}, X_{-\Lambda_i}^{\prime(i,k)})}{p'(X_{\Lambda_i}^{\prime(i,k)})}}{\frac{1}{r} \sum_{k=1}^r \frac{q_\theta(X_{\Lambda_i}^{(i)})}{p'(X_{-\Lambda_i}^{\prime(i,k)})}} \right) + \left\| \frac{\frac{1}{r} \sum_{k=1}^r \frac{\nabla_{X_{\Lambda_i}^{(i)}} q_\theta(X_{\Lambda_i}^{(i)}, x_{-\Lambda_i}^{\prime(i,k)})}{p'(X_{\Lambda_i}^{\prime(i,k)})}}{\frac{1}{r} \sum_{k=1}^r \frac{q_\theta(X_{\Lambda_i}^{(i)})}{p'(X_{-\Lambda_i}^{\prime(i,k)})}} \right\|^2. \tag{24}$$

### E.1.2. MARGINAL TRUNCATED IW SCORE MATCHING

Our full sample objective for truncated IW missing score matching is given by

$$\hat{L}_{\mathrm{M};n,r}(\theta) := \frac{1}{n} \sum_{i=1}^n \sum_{j \in \Lambda_i} \boldsymbol{g}(X_{\Lambda_i^{(i)}})_j \left\{ 2 \partial_j \left( \frac{\frac{1}{r} \sum_{k=1}^r \frac{\partial_j q_\theta(X_{\Lambda_i}^{(i)}, X_{-\Lambda_i}^{\prime(i,k)})}{p'(X_{\Lambda_i}^{\prime(i,k)})}}{\frac{1}{r} \sum_{k=1}^r \frac{q_\theta(X_{\Lambda_i}^{(i)})}{p'(X_{-\Lambda_i}^{\prime(i,k)})}} \right) + \left( \frac{\frac{1}{r} \sum_{k=1}^r \frac{\partial_j q_\theta(X_{\Lambda_i}^{(i)}, x_{-\Lambda_i}^{\prime(i,k)})}{p'(X_{\Lambda_i}^{\prime(i,k)})}}{\frac{1}{r} \sum_{k=1}^r \frac{q_\theta(X_{\Lambda_i}^{(i)})}{p'(X_{-\Lambda_i}^{\prime(i,k)})}} \right)^2 \right\}$$

$$+ 2 \partial_j \boldsymbol{g}(X_{\Lambda_i})^{(i)} \left( \frac{\frac{1}{r} \sum_{k=1}^r \frac{\partial_j q_\theta(X_{\Lambda_i}^{(i)}, X_{-\Lambda_i}^{\prime(i,k)})}{p'(X_{\Lambda_i}^{\prime(i,k)})}}{\frac{1}{r} \sum_{k=1}^r \frac{q_\theta(X_{\Lambda_i}^{(i)})}{p'(X_{-\Lambda_i}^{\prime(i,k)})}} \right). \tag{25}$$

## E.2. Variational Modelling Details

For our purposes our variational model $p_\phi'$ has to able able to model $p_\theta(\boldsymbol{x}_{-\lambda} | \boldsymbol{x}_\lambda)$ for any value of $\lambda \subseteq d$. For our model we take $X_{-\lambda}' | X_\lambda = \boldsymbol{x}_\lambda \sim N(\mu_\phi(\boldsymbol{x}_\lambda), \sigma_\phi^2 I)$.

Hence we require $\mu_\lambda$ to take in any subset of coordinates and output the complementing coordinates We achieve this by creating $\mu_\phi'$, a $d$-dim in to $d$-dim out Neural Network with 2 hidden layers of 200 nodes as per Burda et al. (2016). For the input we replace $\boldsymbol{x}_\lambda$ with $\boldsymbol{x}_\lambda^0$ the zero filled version inline with the approach taken in MissDiff (Ouyang et al., 2023). That is $(\boldsymbol{x}_\lambda^0)_j = 0$ if $j \notin \lambda$ and is $\boldsymbol{x}_j$ otherwise. As output we then simply take the appropriate coordinates. We can write this more succinctly as

$$\mu_\phi(\boldsymbol{x}_\lambda) = \mu_\phi'(\boldsymbol{x}_\lambda^0)_{-\lambda}$$

We also experimented with making $\mu_\phi'$ a $2d$ dim input NN and taking

$$\mu_\phi(\boldsymbol{x}_\lambda) = \mu_\phi'(\boldsymbol{x}_\lambda^0, m)_{-\lambda}$$

where $m$ is the d-dimensional binary mask for the corruption similarly to GAIN (Yoon et al., 2018) however this did not seem to provide any advantage. We also experimented with making $\sigma_\phi$ depend upon $\boldsymbol{x}_\lambda$ however we found this made training much more unstable.

### E.3. Experiment Implementation Details

#### E.3.1. NORMAL DISTRIBUTION ESTIMATION

The mean is taken fixed at $\mu = (0.5, \ldots, 0.5)^\top$. We randomly construct the covariance by first sampling eigenvalues uniformly on $[0.5, 1.5]$ and then sampling choosing eigenvectors uniformly on the unit hypersphere for the first 9 dimensions. We then construct the the 10th dimension strong dependence on only the first dimension by taking $X_{10} = \frac{1}{2}X_1 + \frac{1}{2}Z$ with $\text{Var}(Z) = \text{Var}(X_1)$. The data is then truncated to be above the 10% quantile or each of the first three dimensions.

In each case batches of 100 samples were taken and a learning rate of 0.01 was used with Adam used as the optimisation algorithm. Our score model was parameterised in terms of the Cholesky decomposition of the precision matrix in order to ensure the Precision estimate stayed positive definite. For our Importance weighting and the EM approach of Uehara et al. (2020), an isotropic Gaussian with mean 0 and coordinatewise variance of 16 was used.

#### E.3.2. GAUSSIAN GRAPHICAL MODEL ESTIMATION

For Gaussian graphical model estimation we add L1 regularisation thereby modifying our objective to be

$$L_{\text{TM}}(\theta) + \gamma \sum_{1 \geq j < j' \leq d} |P_{j,j'}|$$

where $\theta = (\mu, P)$ with $P$ being our precision estimate. We minimise the objective by performing proximal gradient descent on $L_{\text{TM}}(\theta)$. Specifically for a learning rate $\nu$, a current estimate of $\theta$ given by $\theta_t$ and an estimate of the gradient given by $\eta_t$. We take our estimate to be $h_{\gamma,\nu}(\theta_t - \nu\eta_t)$ where

$$h(\beta) := \begin{cases} \beta - \gamma\eta & \text{for } \beta > \gamma\eta \\ 0 & \text{for } -\gamma\eta \leq \beta \geq \gamma\eta \\ \beta + \gamma\eta & \text{for } \beta < -\gamma\eta \end{cases}$$

In our experiments we start with a precision estimate being $P = I$ and with a large value of $\gamma$ and, run 200 iterations, and then decrease the value of $\gamma$ every 10 subsequent iterations for 100 sequentially smaller values of $\gamma$. At the end of each block of 10 iterations the precision matrix is taken and an adjacency matrix is produced by thresholding the entry's values at some small value. The TPR and FPR are then calculated for each of these increasingly dense matrices and then an ROC curves plotted using these values. The AUC of this ROC is then computed which is the statistic reported in the plots.

We took L1 regularisation to ensure that at the highest level the graph had no edges and at the lowest level the graph had all possible edges. For this experiment, this was achieved with $\gamma \in (10^{-1.7}, 10^{-4})$. Throughout we took the threshold for edge presence to be 0.002.

#### E.3.3. REAL WORLD DATA

For these experiments we chose the range of L1 regularisation similarly to ensure a full range of edge densities (here this meant $\gamma \in (10^1, 10^{-4})$ and then constructed a semi-automated procedure for choosing the threshold for edge presence. There is precedent for choosing the detection threshold after L1 regularisation as per Fattahi & Sojoudi (2019). We did this by choosing the non-zero threshold at the smallest value that gave a sufficiently smooth increase in edge density between the snapshots where we sample our estimated adjacency matrices.

Specifically the smoothness we were trying to achieve was avoid sudden decreases in the edge density as our regularisation level decreased. Our rough measure of this was to sum up all the negative jumps between sequential adjacency matrix estimates where the previous jump had been positive. This sum was then taken as our measure of "jumpiness" with larger values representing a larger level. Visually inspecting the change in positive level over time we find that a level of 0.01 for high-dimensional cases and a level 0.05 for low dimensional cases represented a relatively smooth change in edge density. The smallest non-zero threshold that satisfied this was then chosen by iterative shrinking grid search.

To test the performance of our adjacency matrix estimates, we estimated the adjacency matrix using standard score matching on the non-corrupted data. We estimated these adjacency matrices at 5 pre-determined values of edge densities given specifically, $0.05, 0.1, 0.15, 0.20.25$. This lead to 5 different "ground truth" adjacency matrices. For each method, the AUC was calculated for each of these "ground truth" adjacency matrices and these 5 AUCs averaged. For each missing probability, 25 random samples of the corruption were produced and this AUC metric calculated. The average of these was then plotted alongside 95% confidence intervals.

The S&P 100 was taken from the S&P 500 data between 2013 and 2018 given in `https://www.kaggle.com/datasets/camnugent/sandp500` with the 100 stocks that made up the S&P 100 taken from roughly the mid-point of the time period which we obtained from `https://en.wikipedia.org/w/index.php?title=S%26P_100&oldid=666413597`.

The yeast data was obtained from `https://ftp.ncbi.nlm.nih.gov/geo/series/GSE1nnn/GSE1990/matrix/` which was accessed via `https://www.ncbi.nlm.nih.gov/geo/query/acc.cgi?acc=GSE1990` and other subsets have previously been studied in the context of GGM estimation in Yang & Lozano (2015).

All data can also be found in the GitHub repository at `https://github.com/joshgivens/ScoreMatchingwithMissingData`.

