# OpenReview forum: "Score Matching with Missing Data"
_ICML.cc/2025/Conference — ICML 2025 oral_

### Official Review · Reviewer_2txV · 2025-02-21

**Overall Recommendation:** 4

**Summary:**

The paper addresses the problem of parameter estimation from missing data with score matching objective. The authors introduce two general frameworks for estimating the (gradient of) marginal score and demonstrate strong empirical performance on synthetic and real data.

**Claims And Evidence:**

The work provides solid theoretical results to demonstrate the applicability of the proposed frameworks. However, the presented empirical evidence is not sufficiently convincing to me.

In Section 5.1, the authors conduct an experiment on synthetic data where they only report the Fisher divergence between the true and estimate score functions. Since the goal is parameter estimation, it would be more convincing if the authors could conduct experiments on parameter recovery tasks and compare the discrepancy between the true and estimated parameters in various settings of missing data. Then we would know how effectively the score matching estimates perform compared to MLE.

**Essential References Not Discussed:**

Parameter estimation from incomplete data is a fundamental statistical problem, to which maximum likelihood estimation via the EM algorithm has been the dominant approach. MisGAN (Li et al., ICLR'19) is another method that uses GAN for estimation. A recent alternative to MLE is the Wasserstein estimate from the optimal transport framework, wherein one seeks the parameters that minimize the Wasserstein distance between the two distributions. Vo et al. (ICML'24) proposes a framework called OTP for learning parameters from missing data.

*Li, S. C. X., Jiang, B., & Marlin, B. MisGAN: Learning from Incomplete Data with Generative Adversarial Networks. In International Conference on Learning Representations.*

*Vo, V., Le, T., Vuong, L. T., Zhao, H., Bonilla, E. V., & Phung, D. Parameter Estimation in DAGs from Incomplete Data via Optimal Transport. In Forty-first International Conference on Machine Learning.*

**Experimental Designs Or Analyses:**

As discussed above, there is a lack of experiments on how well the proposed method can recover the ground-true parameters, for example the mean and/or variances in the Gaussian case. Furthermore, the current results are limited to Gaussian models. It would be convincing to see how the methods can be applied to various models or data types as the proposed framework is claim to be general.

**Methods And Evaluation Criteria:**

The paper learns the parameters by minimizing the explicit score matching objective over the marginal score function. The objective naturally follows the derivation of the original score matching objective from complete data and achieves convergence under the standard assumptions.

**Other Comments Or Suggestions:**

N/A

**Other Strengths And Weaknesses:**

**Strengths:**

The paper is generally well-written and solid. The extension of explicit score matching to dealing with missing data is a straightfoward approach, yet interesting to me. I also appreciate the authors' attempt to consider MNAR cases, though the assumption that the probability of missingness pattern is given is rather unrealistic, rendering the proposal in Proposition A.9 of little use to me. However, it is acceptable to limit the scope of the current work to M(N)AR cases since it is generally known that without making very specific assumptions, the parameters are non-identifiable from MNAR data.

**Weaknesses:**

Similar to EM, the method requires that the unnormalized density $q$ is tractable for estimation and further developments to compute the marginal scores. Perhaps this is a reason why the experiments are limited to Gaussian models.

At the high level, the proposed algorithm somewhat mimics EM where the Marg-IW and Marg-Var variants can be viewed as two ways to perform inference in the E step: the former assumes the density $p'$ is known, similar to the fact the the posterior is assumed tractable; the latter is similar to doing variational inference. Meanwhile, MisGAN and OTP frameworks can sidestep the intractabilty of the density to be learned, though algorithmically, these frameworks also follow an EM-like style: an inference step for missing value imputation, followed by a step of learning parameters from the pseudo-complete data.

So far it remains unclear to me regarding the theoretical advantages of the score matching estimates compared to MLE and other alternatives namely the Wasserstein estimates, for instance in terms of properties of the estimate, complexity or convergence rate. This is in fact important to understand how the proposed method applies in practice.

**Questions For Authors:**

* For Marg-IW approach, how is p' chosen in practice?
* How is the function $h$ in Corollary 4.7 chosen? Algorithm 2 only discusses how to estimate the conditional densities, without mentioning the function $h$.
* Line 706: What do the notations $B$ and $\mathcal{B}_{\mathcal{X}}$ in  refer to?
* Line 259: Is it supposed to be $\log(n)/n$ convergence rate? Furthermore, in the main text, please define the notations used for Theorem 4.5 such as $\beta_1$, $\delta$ for completeness.

**Relation To Broader Scientific Literature:**

The paper introduces a score matching framework for estimating model parameters from missing data. At a high level, score matching aims to find parameters that minimize the Fisher divergence between the observed data distribution and the model distribution. This approach is part of the broader family of parameter estimation methods based on minimum distance estimation.

**Theoretical Claims:**

There is no issue with the correctness of the theoretical claims that I am aware of.

---

> ### Author Rebuttal · Authors · 2025-04-01
>
> Thank you for your insightful comments and feedback as well as the additional references, we will make sure to include them in our paper.
>
> >Since the goal is parameter estimation, it would be more convincing if the authors could conduct experiments on parameter recovery tasks and compare the discrepancy between the true and estimated parameters in various settings of missing data.
>
> We agree that parameter estimation/recovery experiments would be more aligned with our goals. We link to plots on parameter estimation accuracy for the normal case here: https://ibb.co/kk8LKNV and the truncated case here: https://ibb.co/pjDzCpGQ.
> They show largely the same trends as with the fisher divergence and we will make sure to include them in the paper. These results will be added to the main body of the paper.
>
> Our experiments did initially estimate the parameters however we just reported the performance through the Fisher divergence as this allowed us to present a singular performance metric rather than two numbers (one for the mean and the other for the precision matrix.) Parameter estimation error comparisons will be added to the main paper in the revision.
>
> > It would be convincing to see how the methods can be applied to various models or data types as the proposed framework is claim to be general.
>
> We agree with the reviewer's comments and have ran experiments on an non-Gaussian, ICA inspired model with an intractable likelihood of $p(x)\propto \left(\exp{-\sum_{i,j}\theta_{ij}x_i^2x_j^2}\right)$. The task was estimating $\theta_{i,j}$. We ran experiments with varying sample sizes, missingness proportions, and dimensions and results are presented here: https://ibb.co/7JTJCc6b. The the accuracy is reported in parameter estimation error (Frobenius norm of $\theta$ treating it as a matrix).
>
> > So far it remains unclear to me regarding the theoretical advantages of the score matching estimates compared to MLE and other alternatives namely the Wasserstein estimates, for instance in terms of properties of the estimate, complexity or convergence rate. This is in fact important to understand how the proposed method applies in practice.
>
> The key advantage of score matching estimates over MLE or Wasserstein distance minimizer is that we can work with unnormalisable models such as truncated Gaussians and our aforementioned ICA inspired model. Additionally, even for some normalisable models, if we cannot compute a tractable conditional probability model from joint model, we cannot apply EM (which maximizes the likelihood) easily.
>
> As a direct theoretical comparison between score matching and MLE, [1] was able to show cases where score matching is both computationally efficient and obtains the same optimal rate of convergence as MLE, despite score matching handles unnormalizable models.
>
> ## Questions
> > For Marg-IW approach, how is p' chosen in practice?
>
> In practice kept $p'$ simple, choosing to be an isotropic gaussian with zero mean and standard deviation proportional to the standard deviation of the non-missing data.
>
> >How is the function in Corollary 4.7 chosen? Algorithm 2 only discusses how to estimate the conditional densities, without mentioning the function .
>
> $h$ is just a generic function used to define the functional $\psi$ so $h$ takes the values given in equation (10). In other words this is just notation to try and keep equation 10 as succinct as possible
>
> >Line 706: What do the notations $B$ and $\mathcal{B}_\mathcal{X}$ refer to?
>
> Apologies, did not properly introduce this notation. $\mathcal{B}_{\mathcal{X}}$ is supposed to represent all the possible events on the RV X (i.e. the sigma algebra on our covariate space $\mathcal{X}$) so that $B$ represent any arbitrary event on $\cal X$. We will make sure to properly introduce it in the final paper.
>
> > Line 259: Is it supposed to be $\log(n)/n$ convergence rate? Furthermore, in the main text, please define the notations used for Theorem 4.5 such as $\beta$, $\delta$ for completeness.
>
> Yes sorry it is meant to be $\sqrt{\log(n)/n}$ our aim was to highlight that this a $1/\sqrt{n}$ rate up to log factors which diminish rapidly. Also we will make sure to introduce $\beta$ which is a constant >0 and $\delta$ which is any sufficiently small probability.
>
> ## References
> [1] Pabbaraju, C., Rohatgi, D., Sevekari, A. P., Lee, H., Moitra, A., and Risteski, A. (2023). Provable benefits
> of score matching. In Oh, A., Naumann, T., Globerson, A., Saenko, K., Hardt, M., and Levine, S., editors,
> Advances in neural information processing systems, volume 36, pages 61306–61326. Curran Associates,
> Inc.

---

> > ### Comment · Reviewer_2txV · 2025-04-01
> >
> > Thank you for the additional results. The method is more convincing to me now. I have updated my score accordingly.

---

> > > ### Author Response · Authors · 2025-04-07
> > >
> > > Thank you for taking our feedback into consideration and thank you for updating your review!

---

### Official Review · Reviewer_ftZ3 · 2025-03-02

**Overall Recommendation:** 4

**Summary:**

The paper proposes a principled score-based method for learning jointly-specified probabilistic models in the presence of missing data in the training set. The key idea is to match the scores of marginal distributions by marginalising the missing variables first. Since marginalisation is often computationally intractable, the paper introduces two approaches based on importance sampling and variational inference. The method is validated for truncated Gaussian parameter estimation and Gaussian graphical model estimation.

**Claims And Evidence:**

The proposed methods and their proofs seem to be sound. In fact, the paper is very well-written and the accurately-placed remarks provide well-appreciated observations and caveats.

**Essential References Not Discussed:**

I think the exposition of the missing data scenario is fairly succint but sufficient. However, for the interested readers, references to the classical literature that formalises missing data problems could be useful, e.g. [1].

References:

[1] Little and Rubin (2020). Statistical Analysis with Missing Data: Third Edition. (Section 1.3)

**Experimental Designs Or Analyses:**

The experimental setting focuses on fairly simple Gaussian models, and explores parameter estimation and graphical model estimation problems. While the paper could benefit from at least one experiment on a more "interesting" model, such as an EBM, I believe the number of theoretical results and extensions to various score-matching flavors make up for it.

I've got one small question still. For the VI approach you parametrise the conditional variational distribution using a neural network, which is likely the reason why it performs worse than the other approaches in Figure 1. I wonder if for this simple setting you could have specified the variational distribution _jointly_ as $p_{\phi}'(X) = \text{TruncNorm}(X, \phi)$, fit it via $J_F$ and then sample the conditionals $p_{\phi}'(X'\_{-\Lambda} \mid X\_{\Lambda})$ in equations 10/11.

**Methods And Evaluation Criteria:**

To my knowledge, this is first method score-based method in the context of missing data that may be unbiased subject to the typical conditions for importance sampling and variational inference. The potential unbiasedness will be very appreciated by the practitioners working with missing data.

**Other Comments Or Suggestions:**

Suggestions:

* The first equation in Section 4, regarding model estimation from incomplete data via MLE holds not only for MCAR but also for MAR data, even though you only consider MCAR. I believe that the proposed score-based methods would similarly hold for MAR data, without the need for the additional MNAR results given in the appendices. This would make the paper stronger in general, as MCAR is a fairly strong assumption, rarely true in real scenarios.
* In appendix A.1.4 you assume that $\varphi_{\lambda}$ is known to have a method that is independent of how $\varphi_{\lambda}$ is estimated. However, it would be important to note that when the data is MNAR, $\varphi_{\lambda}$ and, consequently, the density of interest may not be identifiable at all, e.g. [1].
* In Appendix D.4 the denoising score matching is introduced from the diffusion perspective, where instead of learning a score of a single distribution, we learn a score of multiple time-dependent distributions. As such, references to [2] or [3] may be relevant here as [4], that is referenced in the text, did not explicitly cover this scenario (although this is correctly highlighted in the remark at the end of the section).

Typos:
* Line 056: "component [of] Z"
* Line 109: "missing [and] non-missing"
* Line 176: "score matching, [we] can relate"
* Line 242: "results"->"result"
* Line 327: "$\hat J_F, \hat J_F$" -> "$\hat J_{KL}, \hat J_F$"
* Line 437: Last sentence of Section 6 has repeated "score matching"
* Line 687: "begin" -> "being"
* Line 720: "show a provide" -> "provide"
* Line 727: "with" -> "we"
* Line 1761: "\cdot"
* Line 1780: "takes" -> "take"
* Subsection D.4.1 should probably be its own section?

References:

[1] Nabi et al (2020). Full Law Identification In Graphical Models Of Missing Data: Completeness Results

[2] Song and Ermon (2020). Generative Modeling by Estimating Gradients of the Data Distribution

[3] Song et al (2021). Score-Based Generative Modeling through Stochastic Differential Equations

[4] Vincent (2011). A Connection Between Score Matching and Denoising Autoencoders

**Other Strengths And Weaknesses:**

As I have expressed above, the paper solves an important problem, is well-written, and I believe will be found useful by practitioners dealing with missing data.

**Questions For Authors:**

I have no further questions.

**Relation To Broader Scientific Literature:**

The paper correctly compares to the two existing methods in the literature and highlights their weaknesses. The paper also proposes extensions of the proposed method to various flavors of score matching, namely, truncated, sliced, and denoising.

**Theoretical Claims:**

The claims are correct and capture the important nuances as far as I can tell. I have checked the proofs in Appendices A and D.4.1, which seem correct.

One short question:

* In Assumption 4.2 you set $\theta > 0$, is that a typo? I am not sure why the parameters should be positive.

---

> ### Author Rebuttal · Authors · 2025-04-01
>
> Thank you for your helpful comments and all the errata found, we will make sure to correct them. We respond to specific questions below.
>
> >In Assumption 4.2 you set $\theta>0$, is that a typo? I am not sure why the parameters should be positive.
>
> Yes, sorry thank you for spotting this, it is a typo it should be for any $\theta$ in our parameter space.
>
> >For the VI approach you parametrise the conditional variational distribution using a neural network, which is likely the reason why it performs worse than the other approaches in Figure 1. I wonder if for this simple setting you could have specified the variational distribution jointly as, fit it via and then sample the conditionals
> in equations 10/11.
>
> This is a good point, and you're correct that for the truncated normal distribution we can indeed approximate with a normal distribution. We have used a more general neural network model in experiments as in many cases, there may not be a joint distribution from which we can easily draw conditional samples (see our ICA inspired example in our response to reviewer 2txV). We wanted to showcase our method in the most generalised way possible.
>
> Thanks for raising this excellent point. We will highlight this possible variant method for the truncated normal regime in the revision.
>
> >The experimental setting focuses on fairly simple Gaussian models, and explores parameter estimation and graphical model estimation problems. While the paper could benefit from at least one experiment on a more "interesting" model, such as an EBM
>
> We appreciate this feedback and have ran experiments on an non-Gaussian, ICA inspired model with an intractable likelihood of $\left(p(x)\propto\exp{-\sum_{i,j}\theta_{ij}x_i^2x_j^2}\right)$. The task was estimating $\theta_{i,j}$. We ran experiments with varying sample sizes, missingness proportions, and dimensions and results are presented here: https://ibb.co/7JTJCc6b. The the accuracy is reported in parameter estimation error (Frobenius norm of $\theta$ treating it as a matrix).
>
>
> >The first equation in Section 4, regarding model estimation from incomplete data via MLE holds not only for MCAR but also for MAR data, even though you only consider MCAR. I believe that the proposed score-based methods would similarly hold for MAR data, without the need for the additional MNAR results given in the appendices. This would make the paper stronger in general, as MCAR is a fairly strong assumption, rarely true in real scenarios.
>
> Thanks for your comments!
> Our approach may not hold for the MAR case. The MLE approach holding for MAR data is specific to the MLE objective rather than the marginal framework in general. However, one can use our MNAR approach for MAR data and in the MAR case the conditional probabilities of being missing should be identifiable.
>
> One can see that MAR adjusted score splits into the sum of the original score and a term involving the missingness pattern (i.e. for MAR data we have $s_{\lambda}(x_\lambda)=\nabla\log\varphi_\lambda(x_\lambda)+\nabla\log p(x_\lambda)$). However, we cannot apply  score matching since we cannot ignore the term involving the missingness probability. Thanks for pointing this out and we will add this discussion in the revision.
>
> As a small side note, we believe the MAR setting to often be not much less restrictive that MCAR, (especially in cases where each coordinate can be missing). Specifically, any case where any two coordinates can be missing at once is likely to be either MCAR or MNAR at least according to the definition in Little \& Rubin (2020). As an example say $X_2$ being missing depends on $X_1$ and each coordinate being missing is independent while this seems like MAR as $X_2$ being observed directly depends upon only $X_1$, for the case were $X_1,X_2$ are both missing (i.e. $\Lambda=[d]\setminus\\{1,2\\}$), we clearly have $\mathbb{P}(\Lambda=[d]\setminus\\{1,2\\}|X)\neq\mathbb{P}(\Lambda=[d]\setminus\\{1,2\\}|X_{[d]\setminus\\{1,2\\}})$ making it not MAR.
>
> >In appendix A.1.4 you assume that is known to have a method that is independent of how is estimated. However, it would be important to note that when the data is MNAR, and, consequently, the density of interest may not be identifiable at all, e.g. [1].
>
> This is a good point and we should have better highlighted that learning the $\varphi_{\lambda}$ is difficult and often non-identifiable without further assumptions or supplementary data. Indeed learning the joint distribution with MNAR data is in general a non-identifiable task. We will highlight this fact in the appendix and also include the provided citation.
>
> We thank the reviewer for the other citations and will include them in the revision, making sure to highlight that multi-level score matching was introduced specifically for diffusion processes.

---

> > ### Comment · Reviewer_ftZ3 · 2025-04-05
> >
> > Thanks for the response! I am happy with the response and maintain my original recommendation.

---

> > > ### Author Response · Authors · 2025-04-07
> > >
> > > Thank you for taking our feedback into consideration!

---

### Official Review · Reviewer_rz1W · 2025-03-13

**Overall Recommendation:** 4

**Summary:**

The paper explores score matching for missing data, proposing two distinct approaches: importance-weighted score matching and a variational approach. The importance-weighted method relies on reweighting score matching objectives using an auxiliary distribution over missing variables, while the variational method frames score estimation as an optimization problem using a parametric variational model. The authors provide theoretical justifications for both methods and evaluate them empirically on synthetic Gaussian data, as well as two real-world datasets: S&P 100 stock data and yeast gene expression data. The results indicate that their methods improve Fisher divergence estimates and AUC scores for missing data imputation, showing modest but consistent improvements over baselines such as Expectation Maximization (EM) and a modified version of MissDiff, a recent diffusion-based model for missing data.

**Claims And Evidence:**

The paper presents two main methodological contributions—importance-weighted score matching and a variational approach for marginal score estimation—and claims that these methods improve score estimation and missing data imputation. The theoretical justifications for both methods are sound, with well-formulated derivations that clearly support the proposed objectives.

However, the empirical evidence is somewhat limited in scope. The experiments primarily focus on Gaussian data, and only two real-world datasets (S&P 100 and yeast gene expression) are used for evaluation. While the results demonstrate modest improvements over baselines in terms of Fisher divergence and AUC scores, the gains are not substantial. Additionally, the choice of baseline for comparison raises concerns—MissDiff is used in a modified form without its original neural network component, making it unclear how the proposed method would compare against a full diffusion-based model. Furthermore, the importance-weighted approach assumes knowledge of an auxiliary distribution  $p{\prime}$  over missing variables, which may not be available in practical applications, potentially limiting its real-world applicability.

Overall, while the theoretical claims are well-supported, the empirical evaluation could be more comprehensive, particularly in terms of dataset diversity and baseline fairness.

**Essential References Not Discussed:**

I would suggest that the authors discuss the following related works, which explore diffusion-based approaches for missing data imputation. Their discussion could help better contextualize the contributions of the current paper.

[1] Zhang, Hengrui, Liancheng Fang, and S. Yu Philip. "Unleashing the Potential of Diffusion Models for Incomplete Data Imputation." CoRR (2024).

[2] Chen, Zhichao, et al. "Rethinking the diffusion models for missing data imputation: A gradient flow perspective." Advances in Neural Information Processing Systems 37 (2024): 112050-112103.

[3] Zheng, Shuhan, and Nontawat Charoenphakdee. "Diffusion models for missing value imputation in tabular data." NeurIPS 2022 First Table Representation Workshop.

**Experimental Designs Or Analyses:**

The empirical soundness is limited due to the concerns previously discussed. The experiments primarily focus on Gaussian data, and while two real-world datasets are included, they may not fully capture the complexity of missing-data scenarios. Additionally, the MissDiff baseline is simplified, making the performance comparisons less conclusive. A broader evaluation with non-Gaussian data and stronger baselines would improve the robustness of the findings.

**Methods And Evaluation Criteria:**

The proposed methods—importance-weighted score matching and a variational approach for marginal score estimation—are well-motivated and theoretically sound. The evaluation metrics, including Fisher divergence and AUC scores, are appropriate for assessing score estimation and imputation performance.

However, the scope of the experiments is limited. The reliance on Gaussian data raises concerns about the generalizability of the method to more complex distributions. Additionally, the MissDiff baseline is simplified, making it unclear whether the proposed methods would outperform a full diffusion-based model. Expanding the evaluation to non-Gaussian data and stronger baselines would make the results more conclusive.

**Other Comments Or Suggestions:**

I have only found one potential typo:
- In line 263 (left column): "Setting it at $r=10$ in our experiments." could be better phrased as "We set it at $r=10$ in our experiments."

**Other Strengths And Weaknesses:**

All relevant strengths and weaknesses have been pointed out in the review.

**Questions For Authors:**

I don't have further questions at this point.

**Relation To Broader Scientific Literature:**

The paper builds on prior work in score-based generative modeling and missing data imputation, particularly leveraging score matching techniques for handling incomplete data. The importance-weighted approach follows principles from importance sampling and prior work on weighted score matching, while the variational approach aligns with existing variational methods used in generative modeling. Additionally, the paper compares its methods to MissDiff, a recent diffusion-based approach for missing data, though the baseline is simplified in the experiments. While the paper is well-positioned within the literature, a more thorough discussion of how the proposed methods compare to alternative generative approaches could strengthen its positioning.

**Theoretical Claims:**

The paper provides clear and strong theoretical justifications that support the proposed importance-weighted score matching and variational marginal score estimation approaches. The derivations are well-structured, and the key results appear correct.

---

> ### Author Rebuttal · Authors · 2025-04-01
>
> Thank you for your useful comments and feedback as well as the additional references which we will make sure to include. We address some specific points below
>
> >However, the scope of the experiments is limited. The reliance on Gaussian data raises concerns about the generalizability of the method to more complex distributions ... Expanding the evaluation to non-Gaussian data and stronger baselines would make the results more conclusive.
>
> Thanks for your comments! To further enhance our experimental section, we have included a new experiment on parameter estimation for an intractable ICA inspired model where $p(x)\propto\exp\{-\sum_{ij}\theta_{ij}x_i^2x_j^2\}$. Links to experimental results on estimate $\theta$ under varying sample size, missingness proportions, and dimensions here: https://ibb.co/7JTJCc6b. As we can see in higher dimensional settings, Marg-Var clearly outperforms other methods while in lower dimensional settings, both Marg-Var and Marg-IW perform best (with EM performing similarly to Marg-IW).
>
> > The MissDiff baseline is simplified, making it unclear whether the proposed methods would outperform a full diffusion-based model.
>
> In this paper, we primarily focus on the parameter estimation tasks. MissDiff was originally designed for estimating the score function in a diffusion process, not the parameterized density model. As such, Missdiff, in its original formulation could not be directly applied to our tasks as it does not provide parameter estimates. This also holds for any other diffusion based imputation approaches such as CoRR which again do not give us a direct parameter estimation procedure.
>
> We include the MissDiff baseline more as an example of what naive marginalisation of the scores would look like and to demonstrate that the idea behind MissDiff can't naturally adapted to the parameter estimation problem. We will make sure to clarify this further in the paper and change the name of the MissDiff-Param approach in our results to make it clear that it is a variant of MissDiff that has been adapted to our problem.

---

> > ### Comment · Reviewer_rz1W · 2025-04-09
> >
> > Dear authors,
> >
> > I mistakenly posted the following as an “Official Comment,” not realizing it wouldn’t be visible to you. I’m re-posting it here to keep you informed:
> >
> > > Thank you for addressing my concerns. The new experiments substantially strengthen the justification of the contributions, and I have adjusted my score accordingly.
> >
> > Thank you again for your thoughtful response.

---

### Official Review · Reviewer_iocR · 2025-03-14

**Overall Recommendation:** 3

**Summary:**

The given paper presents novel framework for Score Matching (SM) in missing data framework. Particularly, the setting assumes that the (multi-dimensional) random variable under consideration has missing coordinates. The authors solve this problem by using an auxiliary variable that denotes the masking random vector. Using this, the authors introduce 'Marginal Score Matching' where the score matching is performed only on the visible coordinates of the random variable. The authors then provide several (tractable) variants of marginal score matching - truncated SM, denoising SM,  sliced SM. However, these methods are still intractable due to marginalization integral present in marginal score definition. To this end, the authors provide two ways to estimate these loss objective - (a) using importance weighting and, (b) using variational approximation. The authors present empirical results on synthetic datasets, PGMs and real world tabular datasets like Stock prices and Yeast data. They compare their method against MissDiff and EM to show the efficacy of their method.

**Claims And Evidence:**

Most of the claims made in the paper are Theoretical which are rigorously justified via mathematical proofs.

The empirical claims seem a little unclear. While the authors point out problems with EM-based methods in Related works, it seems that their method doesn't provide significant advantage performance wise (E.g., Fig. 1).

**Essential References Not Discussed:**

N/A

**Experimental Designs Or Analyses:**

The experiments are primarily performed on synthetic datasets. While it helps in understanding the validity of the method in simple settings, it is not clear how it would perform in complex settings (such as image generation/inpainting, etc). However, since the paper has theoretical inclination, I won't count it as a negative.

Performance wise, it looks like the proposed method, particularly Marg-Var has very slight advantage over EM whereas Marg-IW seems to be performing as well as EM. Can the authors provide an explanation for this?

**Methods And Evaluation Criteria:**

The proposed method makes sense, however, I am not sure about the evaluations. Primarily because of the limited metrics (AUC) used to demonstrate the efficacy of proposed method.

**Other Comments Or Suggestions:**

1. Line 102: \del_j -> \del_1 ?

**Other Strengths And Weaknesses:**

#### Strengths
1. The paper is well written and presented. Although heavily theoretical, I was able to most of the claims and proofs.
2. The proposed setting has practical relevance, specially for tabular data.
3. Going through the proofs, it seems most of the SM objective can be adapted to missing data scenario.

#### Weaknesses
1. The experimental section looks a little weak. Perhaps including tabular data benchmarks could help with this.
2. Marg-IW seems to have similar performance as EM. Is there any insight into this?
3. I am confused as to why these objectives cannot be extended to neural networks? Can't one optimize neural network parameters using these objectives?

**Questions For Authors:**

N/A

**Relation To Broader Scientific Literature:**

The work essentially extends the various score matching objectives to the missing data setting. While there are methods like Ambient Diffusion or Cold Diffusion that operate in more generic setting, the current work has theoretical flavor to it which extends the existing approaches to missing data settings.

**Theoretical Claims:**

I went through the proofs of most of the results. The authors have provided solid mathematical grounding to the proposed method. I am happy and satisfied with the claims and proofs.

---

> ### Author Rebuttal · Authors · 2025-04-01
>
> Thank you for your useful comments and feedback, we address some specific points below.
>
> >The proposed method makes sense, however, I am not sure about the evaluations. Primarily
> because of the limited metrics (AUC) used to demonstrate the efficacy of proposed method.
>
> Thanks for pointing this out!
> Since we focus on the parameter estimation problem,
> we have now added new metrics specifically designed for the parameter estimation problem in standard and truncated normal estimation: https://ibb.co/kk8LKNV and here: https://ibb.co/pjDzCpGQ respectively.
> Additionally we have included a new ICA inspired parameter estimation experiment (more details in 2nd paragraph of response to reviewer 2txV) which can be found here: https://ibb.co/7JTJCc6b .
>
> In Gaussian Graphical Model tasks, we focussed on AUC as we were treating it as an edge detection problem and wanted to demonstrate the efficacy over a variety of sparsity settings. The use of classification metric in graphical model learning tasks through the False Positive Rate (FPR) and True Positive Rate (TPR) was used in previous works [2] and [3], thus, we are following the convention, using the AUC to summarize the overall performance of the graphical model learning.
> For further illustration, we have included a plot of the ROC curves for the first 4 runs of the star based precision with a missing probability of 0.5 here: https://ibb.co/35dgkVnK and will be sure to include them in the appendix of the final paper. In these we can see that the proposed Marg-Var seems to consistently outperform the other approaches.
>
> >Marg-IW seems to have similar performance as EM. Is there any insight into this?
>
> Thanks for raising this interesting question.
> We believe this is because the both methods rely on importance weighting to approximate marginal/conditional expectations. Additionally they both aim to maximize the marginal score (EM can be seen as maximizing the marginal likelihood iteratively.)
>
> However, the detailed reason of their performance similarity is definitely an interesting observation which is worthy of further exploration and we will highlight this future direction in the revision.
>
> >I am confused as to why these objectives cannot be extended to neural networks? Can't one optimize neural network parameters using these objectives?
>
> Yes, You are correct that this approach can be extended to estimating neural network models (such as energy-based model) from missing observations, similar to how its variant for handling non-missing observations [4].
>
> Since most neural network models are also unnormalizable,
> this extension to our method would further expand the usage of such models. We see this flexibility of handling various unnormalizable models as one of the core strengths of our method.
> We decided to focus on analysing the performance of score matching through simpler and more interpretable models such Gaussian Graphical Models as this was something which we felt was under-explored in existing missing data literature. Moreover, estimating truncated graphical model using score matching has been higlighted as an application of score matching [2,3].
>
> Application of our methods to various neural network architectures and its use in downstream tasks such as learning a diffusion model from missing data is definitely a strong direction for future research, and we will make sure to highlight this in our conclusion.
>
> ## References
> [2] Lin, L., Drton, M., and Shojaie, A. (2016). Estimation of high-dimensional graphical models using regularized
> score matching. Electronic Journal of Statistics, 10(1):806 – 854. Publisher: Institute of Mathematical
> Statistics and Bernoulli Society.
>
> [3] Shiqing Yu, Mathias Drton, Ali Shojaie (2022). Generalized score matching for general domains, Information and Inference: A Journal of the IMA.
>
> [4] Song, Yang, and Diederik P. Kingma. "How to train your energy-based models." arXiv preprint arXiv:2101.03288 (2021).

---

### Decision · Program_Chairs · 2025-05-01

**Decision:**

Accept (oral)

**Comment:**

The authors consider statistical estimation with score matching, in the presence of missing data.

They focus on the case where the model of interest is a model with intractable normalising constant (the case of score-based models, where the score is directly modelled is less emphasised in the paper). They propose an objective to maximise, which is the expected value of the norm between the observed scores (the missing values are marginalised) and modelled ones ($F_M(\theta)$, defined in Eqn. (3)). When no values are missing, this reduces to the standard expected score-matching loss. They then use the standard clever tricks for the score-matching literature to obtain empirical versions of $F_M(\theta)$, both when the data is compactly supported and not. They then attack this empirical loss using two approaches to deal with the missing values: one based on importance sampling, and one based on variational inference. They perform experiments on toy datasets, Gaussian graphical models, and added new experiments on independent component analysis (ICA) during the discussion period.

All in all, this is an excellent paper, that nicely extends several blends of score matching to the missing data setting. During the discussion period, **experiments on ICA significantly strengthened the paper, and it is crucial that these experiments are included in the final version of the paper**.

Among the other things that should be clarified in the final version are the advantages/drawbacks of the proposed methods against the EM approach and other methods to estimate models with intractable normalising constants with missing values, and whether or not the work on denoising score matching could be applied to diffusion models.

Finally, two things that I wondered while reading the paper, and that were not really discussed during the discussion: what are the theoretical properties of the obtained parameters estimates (consistency, asymptotic normality...)? Could this be extended to score matching for data that lives on a manifold? These two points could be optionally discussed in the final version.